# Dual carbon sequestration with photosynthetic living materials

Dalia Dranseike [1,7], Yifan Cui[1,7], Andrea S. Ling[2], Felix Donat [3], Stéphane Bernhard[1], Margherita Bernero [4], Akhil Areeckal[1], Marco Lazic[1], Xiao-Hua Qin [4], John S. Oakey [5], Benjamin Dillenburger [2], André R. Studart [6] & Mark W. Tibbitt [1] ✉

Natural ecosystems efficiently sequester $CO_2$ but containing and controlling living systems remains challenging. Here, we engineer a photosynthetic living material for dual $CO_2$ sequestration that leverages biomass production and insoluble carbonate formation via microbially induced carbonate precipitation (MICP). To achieve this, we immobilize photosynthetic microorganisms within a printable polymeric network. Digital design and fabrication of the living structures ensure sufficient light access and nutrient supply to encapsulated cyanobacteria, enabling long-term culture for over a year. We showcase that photosynthetic living materials are able to sequester $2.2 \pm 0.9$ mg of $CO_2$ per gram of hydrogel material over 30 days and $26 \pm 7$ mg of $CO_2$ over 400 days. These findings highlight the potential of photosynthetic living materials for scalable, low-maintenance carbon sequestration with applications in carbon-neutral infrastructure and $CO_2$ mitigation.

Biological ecosystems, such as forests, aquatic systems, and wetlands, offer efficient pathways for carbon sequestration (storing carbon in a carbon pool) and conversion into carbon-based materials[1]. Natural systems operate under ambient conditions with sunlight and commonly available small molecules as their sole inputs. Further, living systems can sense, self-repair, and respond to their surroundings, making them resilient to environmental changes[1,2]. Biological carbon sequestration, for example via afforestation or the growth of marine phytoplankton and algae, is also cost-efficient and environmentally friendly[3]. In this context, natural carbon sequestration can serve as a passive, low-impact complement to industrial carbon sequestration, which normally requires specific, extreme, and energy-intensive conditions[1,4] and proximity to large emission sources[5]. However, natural carbon sequestration is typically slower than industrial carbon sequestration, and the control of living systems outside of their native environments is often challenging[5,6].

Strategies to engineer living systems for active $CO_2$ sequestration would provide an additional approach to mitigate the accumulation of human-generated $CO_2$ in the atmosphere. The $CO_2$ concentrating mechanism of many photosynthetic microorganisms accumulates $CO_2$ within the cell body up to 1000-fold above ambient levels[6,7]. Subsequently, concentrated carbon can be fixed in the form of biomass generated during growth[8,9]. In addition to biomass production, microbially-induced carbonate precipitation (MICP) in certain species can sequester $CO_2$ irreversibly in the form of inorganic carbonate precipitates. MICP proceeds via multiple metabolic pathways, including ureolysis, sulfate reduction, and denitrification[10,11]. In some organisms, MICP can occur as a direct by-product of photosynthesis, whereby the inorganic precipitates effectively act as an additional carbon sink[12], enabling dual carbon sequestration. In this context, immobilizing photosynthetic microorganisms, such as algae and cyanobacteria, within a support matrix may provide an approach to drive

[1]Macromolecular Engineering Laboratory, Department of Mechanical and Process Engineering, ETH Zurich, Zurich, Switzerland. [2]Digital Building Technologies, Institute of Technology and Architecture, ETH Zurich, Zurich, Switzerland. [3]Laboratory of Energy Science and Engineering, Department of Mechanical and Process Engineering, ETH Zurich, Zurich, Switzerland. [4]Institute for Biomechanics, Department of Health Sciences and Technology, ETH Zurich, Zurich, Switzerland. [5]Department of Chemical and Biomedical Engineering, University of Wyoming, Laramie, WY, USA. [6]Complex Materials, Department of Materials, ETH Zurich, Zurich, Switzerland. [7]These authors contributed equally: Dalia Dranseike, Yifan Cui. ✉e-mail: mtibbitt@ethz.ch

biological $CO_2$ sequestration in the form of engineered photosynthetic living materials via dual carbon sequestration.

To date, engineered living materials have primarily been used for applications in biomedicine, sustainable materials production, and as living building materials[13–17]. For example, MICP has been exploited, primarily via ureolysis, to mechanically reinforce living materials based on the in situ formation of a stiff mineral phase[18]. Robust composites were produced via biomineralization using ureolytic MICP within a cellulose matrix[19,20]. Similarly, precipitates deposited in porous materials filled cracks and improved mechanical properties of composite building structures as well as consolidated soils[21]. Ureolytic MICP is attractive due to its short incubation period (typically 1–4 days), resistance to contamination, and rapid biomineralization; however, it poses substantial environmental concerns due to the associated production of large amounts (1–2 equimolar) of ammonia[22]. Further, ureolytic MICP requires a constant supply of urea and only proceeds in a narrow range of environmental conditions[23–25]. These challenges constrain the use of ureolytic MICP for long-term $CO_2$ sequestration[26]. Many of these limitations can be addressed with photosynthetic MICP, which requires no additional feedstocks and produces no toxic by-products[18,25]. Recently, photosynthetic MICP was used to design living building materials that were reinforced by mineralization over time[17]. While photosynthetic living materials have been explored for carbon sequestration via reversible biomass accumulation[9], they have not been explored for $CO_2$ sequestration via biomass accumulation *and* irreversible MICP using atmospheric $CO_2$ as the main carbon source and light as the sole source of energy.

In this work, we engineer photosynthetic living materials for dual $CO_2$ sequestration by immobilizing MICP-capable photosynthetic Cyanobacterium *Synechococcus* sp. strain PCC 7002 within a printable Pluronic F-127 (F127)-based polymeric network. Dual carbon sequestration via biomass generation and insoluble carbonate formation proceeds over the lifecycle (beyond one year) of the bio-printed structures. The mineral phase mechanically reinforces the living materials and stores sequestered carbon in a more stable form. This work provides a strategy to engineer photosynthetic living materials as a scalable $CO_2$ sequestration method to complement ongoing strategies to mitigate atmospheric $CO_2$ accumulation.

## Results

### Engineering living materials for dual carbon sequestration

We engineered photosynthetic living materials for dual carbon sequestration by exploiting the carbon concentrating mechanism of cyanobacterium strain PCC 7002 (Fig. 1a). Dual $CO_2$ sequestration proceeds via both reversible biomass accumulation and irreversible mineral precipitation. For carbon concentration to occur, $CO_2$ from the surrounding environment dissolves in aqueous solutions and forms bicarbonate ions ($HCO_3^-$) that are then transported to the marine β-cyanobacteria 7002 carboxysome[27]. Within the carboxysome, carbonic anhydrase (CA) catalyzes the conversion of bicarbonate into $CO_2$, which is then fixed by ribulose-1,5-bisphosphate carboxylase/oxygenase (RuBisCo) during photosynthesis into two molecules of 3-phosphoglycerate. This product is enzymatically converted into sugars that support cell growth and biomass production, representing the reversible part of $CO_2$ sequestration.

Irreversible $CO_2$ sequestration as mineral precipitation occurs outside the cells, near the cyanobacteria. Negatively charged extracellular polysaccharides on the bacterial membrane combined with a suitable extracellular environment (alkaline pH, presence of divalent cations), facilitates the nucleation and formation of insoluble carbonates[27–31]. Thus, by culturing photosynthetic living materials in simulated seawater that contains $Ca^{2+}$ and $Mg^{2+}$, an increase in medium pH, which occurred in cyanobacteria-containing samples (Supplementary Fig. 1), allows $CO_3^{2-}$ in the medium to be fixed irreversibly into calcium or magnesium carbonates. Chemical equilibrium then favors

the dissolution of additional atmospheric $CO_2$ into the culture medium, continuously driving the dual carbon sink.

Cyanobacteria PCC 7002 were encapsulated in a hydrogel matrix to fabricate photosynthetic living materials capable of dual carbon sequestration (Fig. 1b, Supplementary Fig. 2). A synthetic polymeric hydrogel based upon Pluronic F-127 (F127) was chosen for the encapsulation of cyanobacteria due to its bioinert nature and its processing versatility. F127-based hydrogels have been used in the design of engineered living materials due to the facile diffusion of most small molecules through it[32]. Functionalized F127-bis urethane methacrylate (F127-BUM) can be photo-cross-linked either post-printing or processed directly via light-based additive manufacturing for long-term structural stability (Fig. 1c). In order to structure the photosynthetic living materials using digital fabrication, we designed a bioink (13.2 wt% F127 and 7.3 wt% F127-BUM) that maintained high viability of encapsulated PCC 7002 (Supplementary Fig. 3) and printability via direct ink writing and light-based additive manufacturing. Rationally designed hydrogel network allows the access of light into the scaffold, thereby enabling homogeneous bacteria growth throughout the entire structure.

Bioink transparency is required for efficient light transmission to drive photosynthesis[28]. Light attenuation due to absorption and scattering within the bioink is one of the main challenges to overcome during cultivation of encapsulated photosynthetic species[33]. The F127-based hydrogel transmitted light ($76 \pm 3\%$) over the entire visible wavelength range (400–750 nm); the BG11–ASNIII medium used to culture PCC 7002 had $95 \pm 4\%$ transmittance in the same range (Fig. 1d). The addition of the photoinitiator lithium phenyl-2,4,6-tri-methylbenzoylphosphinate (LAP), which absorbs light below 420 nm, reduced transmittance only at low wavelengths that are not relevant for PCC 7002 photosynthesis (Supplementary Fig. 4)[34]. The encapsulation of cyanobacteria decreased the transmittance of the final construct to $28 \pm 8\%$ and $31 \pm 9\%$ before and after cross-linking, respectively, due to scattering and productive absorption of light by the photosynthetic bacteria (Supplementary Fig. 5). Overall, the optical properties of the photosynthetic living material were suitable to enable cyanobacteria viability and growth when encapsulated within F127-based hydrogel.

To structure the photosynthetic living materials for enhanced carbon sequestration and longevity, we designed the F127-based hydrogel for both direct ink writing and light-based additive manufacturing (Supplementary Figs. 6 and 7). The ink exhibited shear-thinning (shear-thinning index $n = 0.09$, $n < 1$, Supplementary Table 1) and elastic recovery (-90%) after high shear, demonstrating its feasibly for extrusion-based printing (Supplementary Figs. 8 and 9). To stabilize printed structures for long-term use, the hydrogel mixture was photo-cross-linked ($\lambda = 405$ nm; $I = 8$ mW cm$^{-2}$; $t = 60$ s) to increase the final storage modulus ($G'$) of the construct (Fig. 1c, e, Supplementary Figs. 10 and 11).

### Dual $CO_2$ sequestration capacity of photosynthetic living materials

Dual $CO_2$ sequestration in the photosynthetic living materials was provided by PCC 7002 biomass growth and insoluble carbonate precipitate formation through MICP. To further investigate both aspects of carbon sequestration, uniform circular samples ($V_{sample} = 40$ μL, $d = 10$ mm, thickness $l = 0.5$ mm) were printed using 22 G conical nozzles ($\varnothing = 0.41$ mm) via direct ink writing and incubated for 30 days (Fig. 2a). During the incubation, BG11–ASNIII medium was changed every 5 days, and the dissolved inorganic carbon (DIC) was quantified in the collected samples (Supplementary Method 1). From day 5, the concentration of $Ca^{2+}$ in the medium was set to 8.65 mM via $CaCl_2$ addition to simulate natural seawater conditions[35] that promote MICP. 40.4 mg L$^{-1}$ (0.38 mM) of $Na_2CO_3$ was dissolved in the medium to simulate natural seawater. Analysis of the total DIC under equilibrium

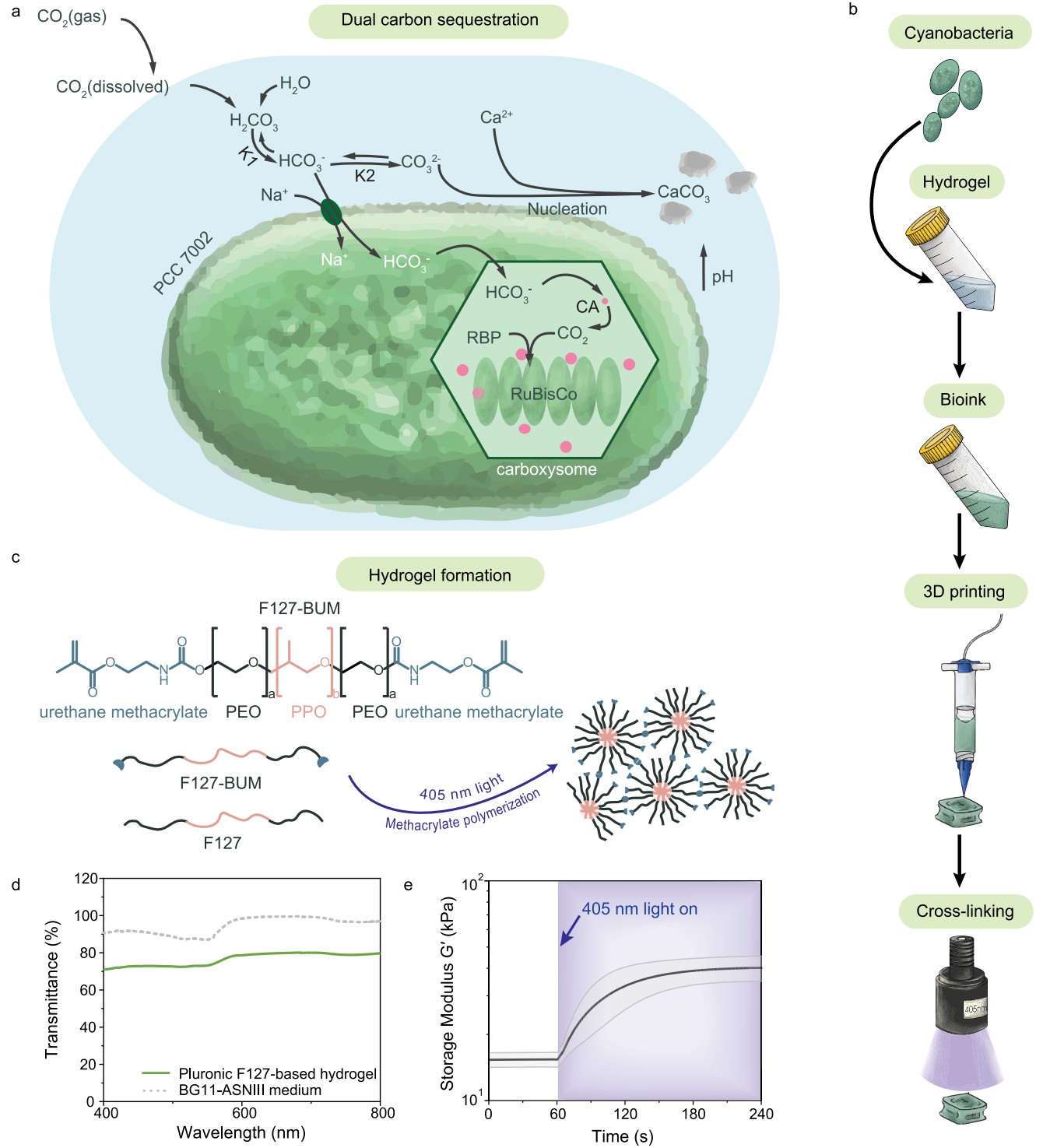

**Fig. 1 | Preparation of photosynthetic living materials for dual carbon sequestration. a** PCC 7002 are capable of $CO_2$ sequestration via biomass accumulation and microbially induced carbonate precipitation (MICP) during photosynthesis using dissolved $CO_2$ as the major carbon source. Cartoon elements used in the schematic diagram were created using Adobe Illustrator 2023. **b** Photosynthetic living materials were fabricated by combining PCC 7002 with a hydrogel bioink, that was structured via 3D printing and stabilized via photopolymerization. Cartoon elements used in the schematic diagram were created using Procreate and Adobe Illustrator 2024. **c** The support hydrogel was composed of a blend of F127 and F127-BUM (bis-urethane methacrylate), which was photocross-linked to form the final stable constructs. Cartoon elements used in the schematic diagram were created using Adobe Illustrator 2023. **d** The hydrogel permitted visible light penetration to support the photosynthesis of encapsulated PCC 7002. **e** The storage modulus ($G'$) increased upon photo-cross-linking under 405 nm (intensity $I = 8 \, \text{mW cm}^{-1}$) light rendering the stable final materials. The gray shaded region indicates standard deviation from independent measurements ($n = 4$). Data are mean ± SD. Source data are provided as a Source Data file.

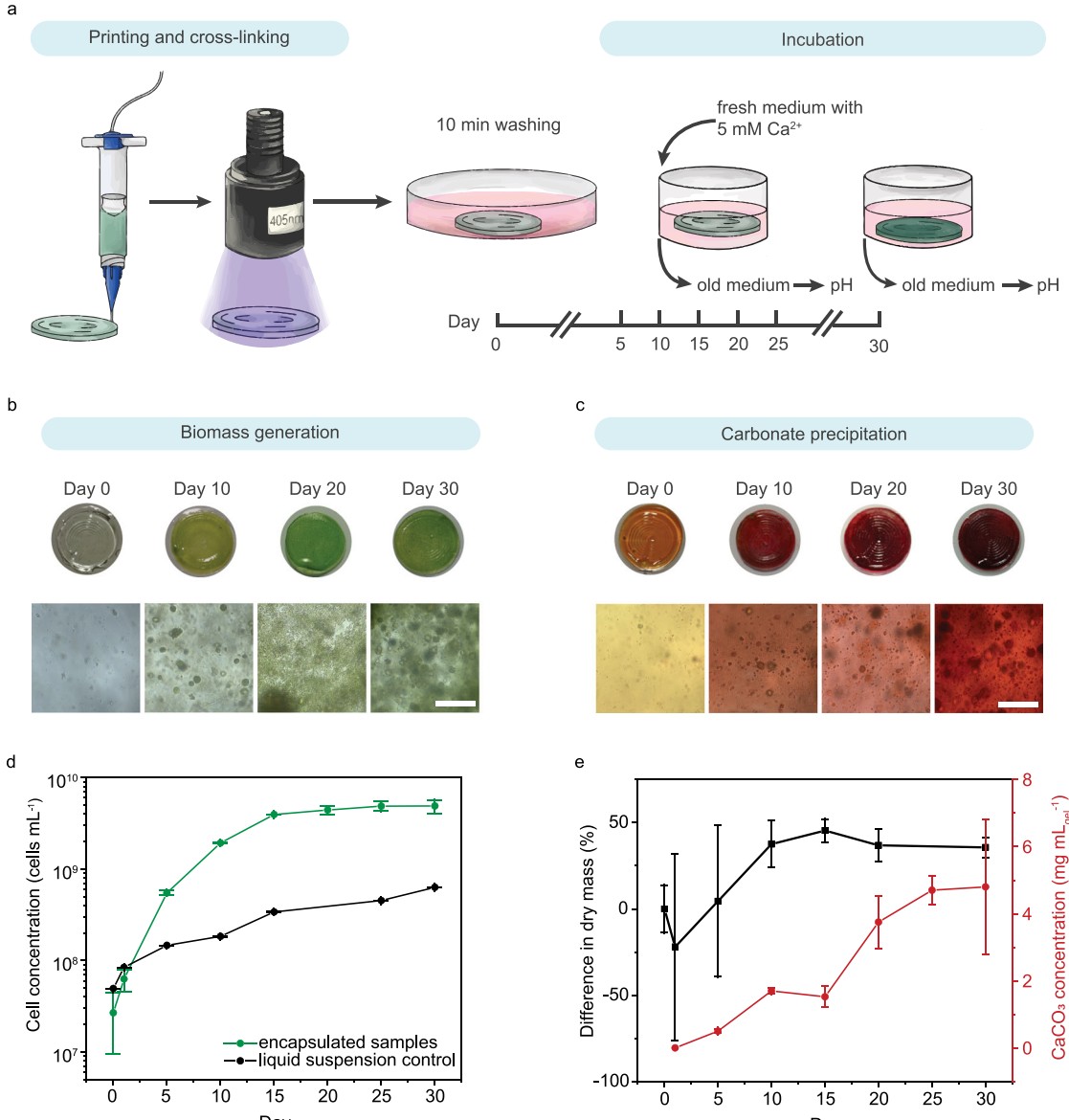

**Fig. 2 | Dual carbon sequestration in photosynthetic living materials.**
**a** Standardized discs of the photosynthetic living material were printed and cross-linked prior to incubation in PCC 7002 culture medium for 30 days. Samples were exposed to light (wavelength $\lambda = 405$ nm, intensity $I = 8$ mW cm$^{-2}$) and the medium was exchanged every 5 days throughout the incubation. Cartoon elements used in the schematic diagram were created using Procreate and Adobe Illustrator 2024. **b** Biotic samples containing PCC 7002 generated biomass during the 30-day incubation, evidenced by the increasingly green color of the discs and expansion of cell clusters. Biomass accumulation was enabled by carbon utilization via carbonic anhydrase (CA) catalysis of $HCO_3^-$. Representative micrograph from $n = 5$ independent samples with similar results. Scale bar, 100 μm. **c** In addition to biomass accumulation, carbon sequestration in the form of $CaCO_3$ precipitates was visualized by Alizarin Red S staining of the biotic samples over 30 days. Representative micrograph from $n = 5$ independent samples with similar results. Scale bar, 100 μm.

**d** An increase in cell concentration was observed throughout the 30-day incubation in the randomly selected encapsulated samples ($n = 3$, biological replicates), with an initial rapid growth rate up to day 15 followed by a slower growth rate from day 15 to day 30 (green). Liquid suspension with the same starting cell concentration was used as a control of cell growth (black). Data are mean ± SD. **e** Both biomass growth and $CaCO_3$ precipitation contributed to the increase in dry mass of the biotic samples. The cumulative $CO_2$ sequestration by biotic samples was reflected by the difference in normalized dry mass between biotic and abiotic samples (black). Over the incubation period of 30 days, biotic samples accumulated 36% more dry mass than the abiotic samples. Alizarin Red S staining on the biotic samples indicated a total of 4.8 mg $CaCO_3$ precipitation per milliliter of gel (red). Error bars indicate the propagated standard deviation of the difference between two groups. Source data are provided as a Source Data file.

in the culture medium indicated that the DIC content was above or at the quantity of inorganic carbon initially dissolved in the form of $CO_3^{2-}$ (Supplementary Fig. 1). This increase in the DIC confirmed the dissolution of atmospheric $CO_2$, which together with dissolved $Na_2CO_3$ in the simulated seawater medium, served as the carbon sources for $CO_2$ sequestration.

PCC 7002 biomass growth within the living material for $CO_2$ capture was evaluated by fluorescence microscopy (cell counts) and

chlorophyll extraction (Fig. 2b). After 10 days of incubation samples exhibited visible clusters of PCC 7002 cells as well as many individual cells throughout the volume of the hydrogel matrix. The number of clusters and individual cells per unit volume of living material (written as $mL_{gel}$) increased over time with cell concentrations reaching $5 \times 10^9$ cells $mL_{gel}^{-1}$ after 30 days (Fig. 2d, Supplementary Fig. 12). Further, the choice of initial $OD_{730nm}$, within the range of 0.3–1.6, did not impact cell growth (Supplementary Fig. 13a).

Moreover, the majority of the encapsulated bacteria remained within the hydrogel throughout the incubation in biotic samples (Supplementary Fig. 13b). Beyond 30 days, we expect the biomass accumulation to reach a pseudo-steady state (with similar cell death and growth rates) based on the plateau observed in the cell concentration beyond day 25.

To confirm $CO_2$ sequestration via the production of insoluble carbonate precipitates via MICP, precipitates were visualized using calcium staining. In the presence of divalent ions such as $Ca^{2+}$ and $Mg^{2+}$, cyanobacteria go through the biomineralization process to form gypsum, calcite, and magnesite[36]. We confirmed the accumulation of $Ca^{2+}$ in the hydrogels using Alizarin Red S staining (Fig. 2c). On day 0, abiotic and biotic samples were mainly orange after staining with green PCC 7002 cells visible in the biotic samples, indicating minimal $Ca^{2+}$ accumulation initially. The abiotic samples remained orange throughout the 30-day incubation (Supplementary Fig. 14). Biotic samples stained red by day 10, indicating $Ca^{2+}$ accumulation in the whole volume of the hydrogel and the dark red color was observed around growing bacteria clusters over time. From the analysis of Alizarin Red S staining, we observed that the amount of precipitates was equivalent to >1 mg of $CaCO_3$ $mL_{gel}^{-1}$ already after 10 days (Fig. 2e). It reached $4.8 \pm 2$ mg of $CaCO_3$ $mL_{gel}^{-1}$ by the end of the 30-day period. Precipitates were also observed in the bottom of the incubation wells, and all analyses were carried out in fresh wells to avoid any influence of precipitation outside of the living material on our quantification (Supplementary Fig. 15). To assess the amount of biomass and precipitate formation, the mass of abiotic and biotic samples was measured and compared (Fig. 2e). The initial decrease in sample mass was attributed to the diffusion of unfunctionalized F127 polymer out of the printed discs, which also caused high variability in the data of day 1 and day 5. Over the incubation period of 30 days, the biotic samples had approximately 36% more dry mass than control abiotic samples with significant increase ($P = 0.001$) in dry mass observed from day 10 of incubation. In total, biomass and carbonate precipitates in biotic samples accounted for approximately 45% of the final sample mass (Supplementary Method 2).

To quantify the extent of precipitate generation, the organic biomass and polymer matrix were removed by thermal decomposition. As the temperature ($T = 600\,°C$) was above the decomposition temperature of organic matter (polymer and biomass) and below the decomposition temperature of carbonate compounds[37], only the insoluble carbonate precipitates remained after thermal decomposition. The mass of the inorganic precipitates corresponded to 50 μmol ($2.2 \pm 0.9$ mg) of $CO_2$ sequestered via MICP per gram of hydrogel (Supplementary Fig. 16 and Supplementary Method 2). $CO_2$ sequestered via MICP has been stored in a more stable mineral form.

**Composite materials form during the life cycle**

Having demonstrated that photosynthetic living materials are capable of $CO_2$ sequestration via both biomass accumulation and carbonate formation, we further investigated the composition of the mineral phase produced during incubation. MICP results in the formation of crystalline carbonates that vary in stability and solubility with calcite being the most stable polymorph and, therefore, desirable for carbon sinking purposes (Fig. 3a)[38]. TGA analysis confirmed carbonate presence in the precipitates (Fig. 3b), which was additionally confirmed by FTIR (Supplementary Fig. 17). XRD analysis of precipitates in biotic samples revealed a highly crystalline structure with the main peaks corresponding to calcium–magnesium carbonates indicated by the position of the main diffraction peak representing the (104) plane (Supplementary Fig. 18). The observed shift likely indicated compositional variation due to the incorporation of magnesium in the carbonate structure as previously observed in marine environments and, in this case, due to the use of simulated seawater (ASNIII medium)[39]. The XRD was obtained after thermal decomposition ($T = 600\,°C$, which is below the decomposition temperature of $CaCO_3$)[37] as the signal from the F127-based matrix was much stronger than the one of the precipitates prior to decomposition (Supplementary Fig. 19).

Precipitates were distributed throughout the polymeric matrix (Fig. 3c). Energy dispersive X-ray analysis (EDS) indicated mainly carbon and oxygen in the abiotic matrix, whereas peaks of calcium and magnesium were prominent in the areas with precipitates in biotic samples. Elemental mapping corroborated these data with calcium, magnesium, and oxygen enrichment in overlapping regions near clusters of cyanobacteria, suggesting the pericellular formation of carbonates (Fig. 3d and Supplementary Fig. 20). EDS measurements also indicated that Ca to Mg atom % ratio in the area of the precipitates was $78 \pm 5\%$ to $22 \pm 5\%$ (number of analyzed samples $n = 10$), which confirmed that both elements were embedded in the precipitates.

The formation of precipitates in hydrogel matrices via MICP can reinforce the mechanical properties of the material[17,40,41]. The storage moduli ($G'$) of the printed biotic samples on day 2 of the experiment were slightly lower compared with abiotic samples ($5.4 \pm 2$ kPa and $6.7 \pm 2$ kPa, respectively). A significant increase ($P = 0.05$) in storage moduli ($G'$) in the case of biotic samples was observed already after 10 days (Supplementary Fig. 21). Over the course of 30 days, the modulus increased significantly to $10.1 \pm 1$ kPa ($P = 0.01$) in the biotic case while it did not change for abiotic samples ($6.5 \pm 1$ kPa) (Fig. 3e). Comparable results were obtained for the Young's moduli ($E$) of the samples using tensile tests of cast samples (Supplementary Fig. 22). We associated the increase in modulus with the formation of reinforcing inorganic precipitates[42], though increases in biomass could also contribute to the change in Young's moduli[43,44]. A similar trend was observed for the material toughness. After 30 days, the toughness of the biotic samples increased from $960 \pm 110$ J m$^{-3}$ on day 2 to $1120 \pm 200$ J m$^{-3}$ on day 30 (Fig. 3f, Supplementary Fig. 23).

**3D-printed photosynthetic living structures for dual carbon sequestration**

While simple discs served as a standardized shape for the systemic characterization of photosynthetic living materials, the geometry of the living structures was tailored via digital design and fabrication to improve the $CO_2$ sequestration and long-term viability of the photosynthetic living materials. We designed 3D lattice structures with strut sizes between 0.15 mm and 0.70 mm to facilitate gas and nutrient transport within the printed constructs. The transport of gases and liquid medium as well as access to light are essential for the efficient functioning of biologic processes (including carbon sinking) in the photosynthetic living materials[9,45]. The decrease in light penetration over the course of the life cycle is expected due to increased light absorption and scattering with increased biomass, resulting in reduced proliferation in the volume fraction further from the surface (Supplementary Fig. 24). Drawing inspiration from cellular fluidics[46], we employed a lattice design to engineer structures whereby growth medium was passively transported vertically through the construct via capillary forces (Fig. 4a, Supplementary Fig. 25). In this manner, the structure did not need to be fully immersed in the simulated seawater, minimizing medium use, and facilitating gas transport from the surrounding air to the living material. To achieve high resolution and defined internal porosity, light-based volumetric 3D printing was used to fabricate the lattice structures. Centimeter-scale objects were printed with complex geometries and an optical resolution of 28 × 28 μm within tens of seconds (Fig. 4b). The 3D-printed photosynthetic living structure was cultured for over a year, during which it continuously produced chlorophyll and performed dual carbon sequestration (Supplementary Fig. 26). After 30 days of incubation, the structure was able to stand vertically on a flat surface and liquid was actively drawn up via the construct's internal structure. The printed sample stiffened further over prolonged incubation due to the accumulation of carbonate precipitates. This mechanical enhancement was

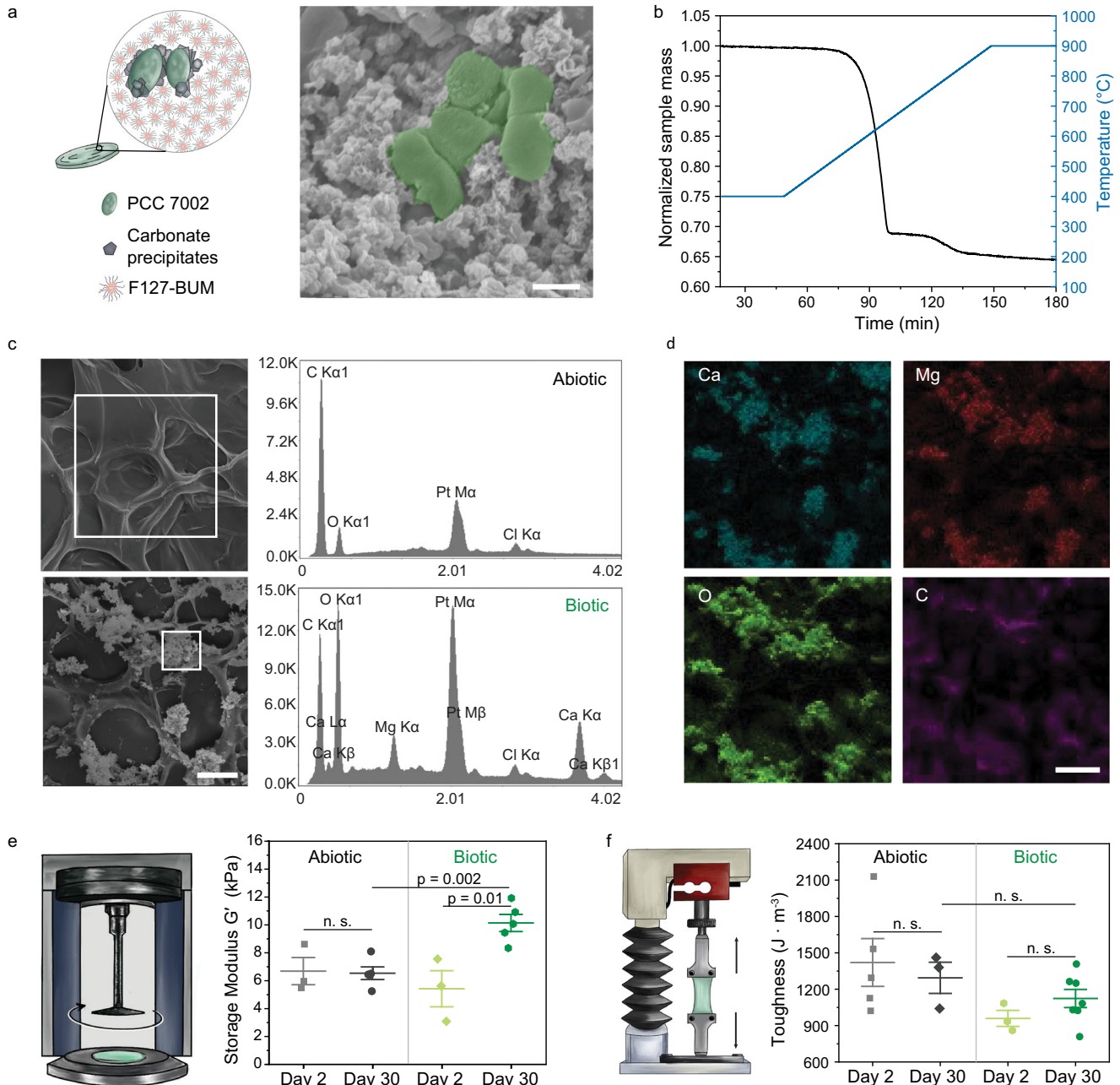

**Fig. 3 | Carbonate formation within photosynthetic living materials. a** During incubation, carbonate precipitates formed around PCC 7002 cells as visualized by scanning electron microscopy (SEM). Scale bar, 1 μm. Cartoon elements used in the schematic diagram were created using Procreate and Adobe Illustrator 2024. **b** Thermogravimetric analysis on the obtained mineral precipitates under nitrogen environment of the remaining precipitates, following removal of the polymer content and bacterial biomass by thermal decomposition, indicated the presence of carbonates in the biotic samples[37,69]. **c** Energy-dispersive X-ray spectroscopy (EDS) analysis on the selected region indicated in the SEM images of abiotic and biotic samples corroborated the presence of calcium and magnesium-containing carbonates only in the biotic samples. SEM imaging was performed on $n = 3$ independent samples, all of which showed comparable results. A representative image is shown. Scale bar, 20 μm. **d** Representative EDS elemental mapping indicated co-localization of Ca, Mg, O, and C atoms in the mineralized regions of the biotic samples. Scale bar, 20 μm. **e** The storage modulus ($G'$), measured by shear

rheometry, increased during the 30-day incubation in the biotic samples ($n = 3$, biological replicates on day 2; $n = 5$, biological replicates on day 30). The modulus of abiotic samples ($n = 3$, biological replicates) remained constant. Cartoon elements used in the schematic diagram were created using Procreate and Adobe Illustrator 2024. n.s. indicates not significant. *P* values were calculated using a two-sided two-sample *t*-test, no adjustments were made for multiple comparisons. Data are mean ± SD. **f** Correspondingly, the toughness of the abiotic samples ($n = 4$, biological replicates on day 2; $n = 3$, biological replicates on day 30) did not change substantially between day 2 and day 30, while the toughness of biotic samples ($n = 3$, biological replicates on day 2; $n = 7$, biological replicates on day 30) increased during the 30-day incubation. Cartoon elements used in the schematic diagram were created using Procreate and Adobe Illustrator 2023. n.s. indicates not significant. *P* values were calculated using a two-sided two-sample *t*-test, no adjustments were made for multiple comparisons. Data are mean ± SD. Source data are provided as a Source Data file.

observable through the structure's increased capacity to maintain an upright position at day 30, 60, 120, and 365 and a final modulus of $111 \pm 7$ kPa (Supplementary Table 2). Over the incubation period of 400 days the living structures sequestered $26 \pm 7$ mg of $CO_2$ per gram

of hydrogel material in the form of carbonate precipitates (Supplementary Table 3). This prolonged culture of the photosynthetic living materials showed that dual carbon sequestration can take place beyond 30 days, especially within rationally designed 3D structures.

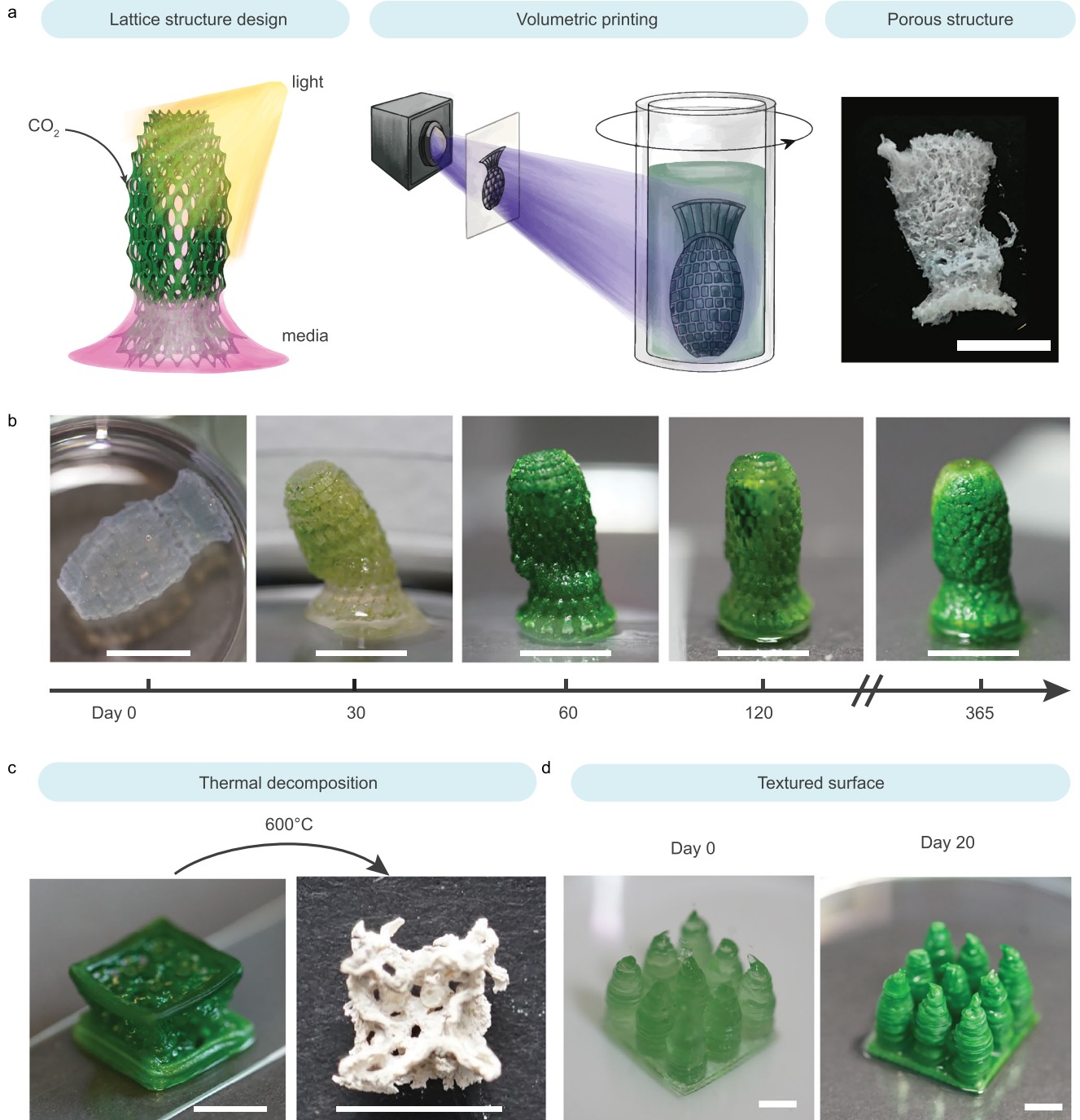

**Fig. 4 | Digital fabrication of photosynthetic living structures for dual carbon sequestration. a** Lattice structures were digitally designed to facilitate light exposure and medium transport via passive capillary wetting, enabling growth of the photosynthetic living materials. The structures were fabricated using volumetric printing. The porosity of the printed structure can be observed in the lyophilized sample. Cartoon elements used in the schematic diagram were created using Procreate and Adobe Illustrator 2024. **b** The 3D-printed lattice evolved during the incubation from a free-floating structure on day 0 to a self-standing and partially mineralized construct at day 30. The photosynthetic living material continued to evolve over time and maintained a vibrant green color up to 365 days, indicative of continuous chlorophyll production. Scale bar, 1 cm. **c** A larger cubic lattice structure was printed using direct ink writing and incubated for 60 days. The sample was thermally decomposed (temperature $T = 600\,°C$) to remove the biomass and polymer matrix, revealing a templated mineral construct. Scale bar, 1 cm. **d** To increase the volume of viable material per unit of surface area, a coral-inspired structure was printed via direct ink writing (day 0), which demonstrated improved growth (day 20) as compared with a flat construct of the same surface coverage (footprint) and volume owing to enhanced light exposure. Scale bar, 0.5 cm.

To demonstrate the scalable production of photosynthetic living materials, direct ink writing was used to print similar lattice structures at a larger scale. A body-centered cubic (BCC) lattice structure (unit cell length $l = 5\,mm$, strut diameter $d = 0.12\,mm$) was printed. The structure passively perfused medium to the top surface while only partially immersed in BG11–ASNIII medium (Supplementary Fig. 27a). After 60 days of incubation, the lattice was thermally decomposed ($T = 600\,°C$). The remaining carbonate precipitates after thermal decomposition retained the shape of the porous lattice structure (Fig. 4c), indicating that carbon sequestration was

homogeneous throughout the structure despite being partially immersed in medium.

This living material also has the potential to be applied on existing surfaces as coatings for dual carbon sequestration. Starting from bulk material, we observed 5 mm to be an optimal material thickness to maintain PCC 7002 viability ($0.5\,g_{material}\,cm^{-2}$) (Supplementary Fig. 27b)[9]. To maximize the viable volume of the material per surface area, textured surface was designed as $3 \times 3$ pillar array on a $2 \times 2$ cm base to minimize self-shielding of light similarly to coral reef structures[47]. With this design we increased the printed gel volume by 150% while maintaining the same $1\,cm^2$ footprint (to $0.75\,g_{material}\,cm^{-2}$) and without compromising bacteria viability compared with a bulk hydrogel (Fig. 4d, Supplementary Fig. 28). This result further highlights how the synergy between living materials and the design of living structures can increase the efficiency of dual carbon sequestration.

## Discussion

The design of our photosynthetic living materials builds upon many recent advances in engineered living materials[9,13,48,49]. The diverse functionality of microorganisms has enabled the design and application of engineered living materials in various fields. For instance, bacteria-based biosensors are used to detect target molecules, such as glucose or specific pathogens[50,51]. Living skin patches adopt skin microbiomes to control bacterial outgrowth on the wounded area[52]. Notably, bacterial mineralization processes have been exploited to design microorganism-based living building materials that can self-heal, strengthen over time, and regenerate[16,17,53], demonstrating the potential to combine biomass accumulation and mineralization within a single living material. Living materials have also been used to replace conventionally pollution-intensive processes, such as applications in the energy industry, the production of textiles or structural materials, due to their inherent bioremediation abilities, especially through the carbon sequestration capabilities of certain microorganisms[54-58].

Building on these concepts, we engineered printable photosynthetic living materials that were capable of carbon sequestration through biomass accumulation and inorganic carbonate precipitation. To enhance the carbon sequestration potential of our photosynthetic living material, we used different additive manufacturing approaches to design biomimetic porous structures that drew upon principles outlined by cellular fluidics[46]. Tailoring construct geometry via additive manufacturing enabled our photosynthetic living structures to continuously perform dual carbon sequestration beyond one year. Throughout the incubation period of 400 days, our photosynthetic living material continuously performed dual carbon sequestration, with most of the sequestered carbon stored in the stable mineral form. The total amount of $CO_2$ sequestered through mineral formation after 400 days was $26 \pm 7$ mg per gram of photosynthetic living material. This amounts to approximately 12 times more than the $CO_2$ sequestered through mineral formation during our 30-day experiments ($2.2 \pm 0.9$ mg per gram of living material). With minimum requirements of sunlight and atmospheric $CO_2$, our photosynthetic living materials showed a consistent carbon sequestration efficiency throughout the entire incubation period, and demonstrated the potential of using natural carbon sequestration processes at scale[59]. We expect the system to reach its peak photosynthesis rate early as the cells go through an initial exponential growth phase and later operate at a pseudo-steady state (Supplementary Fig. 29). Future work should include a more quantitative evaluation of $CO_2$ sequestration through biomass accumulation to better understand the photosynthetic rate of the cyanobacteria. Direct measurements of $O_2$ evolution using an oxygen electrode would provide a more accurate quantification of photosynthesis. Additionally, established computational models could be applied to further explore the relationship between cytosolic pH of the cyanobacteria and $CO_2$ assimilation into biomass[60].

As a comparison of the efficiency of this living approach, similar $CO_2$ capture initiatives via chemical mineralization, such as carbonating recycled concrete aggregates, have been able to sequester $CO_2$ at a quantity of 6.7 mg per gram of recycled concrete aggregates[61]. Thus, our first-generation photosynthetic living material ($26 \pm 7$ mg of $CO_2$ sequestration per gram of living material) may be competitive with industrial chemical mineralization process. Moreover, the carbon sequestration capacity of our photosynthetic living materials is not limited by the availability of essential chemicals, such as the presence of $Ca(OH)_2$ in recycled concrete aggregates. Comparatively, $CO_2$ sequestration via chemical and biological mineralization is less efficient than carbon capture and storage (CCS); however, CCS requires a concentrated $CO_2$ source as well as controlled temperature and pressure to operate. Our photosynthetic living materials function at ambient conditions with atmospheric $CO_2$ and simulated seawater as carbon sources, which highlights the ability to use photosynthetic living materials as a complementary approach in a more distributed manner.

Implementing photosynthetic living materials in broad application still requires improved usability and upscaling of the material production. This can be achieved by taking advantage of recent advances in biofabrication for the creation of larger scale porous[62] or granular[63] structures. Upscaling would provide the means to do more robust $CO_2$ sequestration analysis using gas composition monitoring (Supplementary Fig. 30, Supplementary Method 3) as well as control and tracking over carbon sources via media and atmospheric supply. In addition, methods to engineer optical structures at scale for efficient light harnessing, may further improve efficiency[47]. Photosynthetic living materials also hold promise for future applications as surface coatings for green building materials or bioreactors in commercial-scale sequestration plants, enabling bioremediation of $CO_2$ emissions and could potentially support the reduction of $CO_2$ emission in the built environment. However, a comprehensive life cycle assessment would be necessary to quantify the net carbon impact of such systems. Their simple requirements and easy maintenance also enable possible installation in various environments, ranging from urban to rural landscapes, for long-term and sustainable $CO_2$ sequestration. To further enhance the efficiency of the system, we foresee the possibilities to genetically modify or select microorganisms or microorganism consortia with higher photosynthetic rates.

## Methods

### Bacteria culture and encapsulation
The cyanobacteria *Synechococcus* sp. PCC 7002 (Pasteur Institute, France) were cultured in BG11–ASNIII medium mixture (Supplementary Method 4, and Supplementary Tables 4 and 5) in Erlenmeyer flasks on a shaker plate with constant shaking (150 rpm) under full-spectrum white light (180 μmol photons $m^{-2}\,s^{-1}$) with a 12-h day (on)/night (off) cycle. Optical density of the bacteria suspension at 730 nm ($OD_{730nm}$) was measured using UV-visible light spectrophotometer (Lambda 35, Perkin Elmer) to monitor bacterial culture growth phases. For encapsulation, cyanobacteria were collected at the early exponential growth phase.

### Bioink preparation
Stock solutions of 18 wt% F127 in culturing medium, 30 wt% of F127-BUM (Supplementary Method 5 and Supplementary Fig. 31) in Milli-Q water, and 5 wt% LAP (Supplementary Method 6 and Supplementary Fig. 32) in Milli-Q water were mixed at 3:1:0.082 ratio to reach final concentrations of 13.2 wt% F127, 7.3 wt% F127-BUM, and 0.1 wt% LAP. This formulation was optimized to have the lowest polymer content and stable hydrogel after photo-cross-linking (Supplementary Figs. 6 and 10). LAP concentration was chosen to have a high cross-linking rate without compromising cell viability (Supplementary

Figs. 33 and 34). For the bioink preparation, cyanobacteria suspension from the early exponential phase was centrifuged at $3500 \times g$ for 5 min, the supernatant was discarded, and the cell pellet was resuspended in the prepared bioink mixture (liquefied on ice) via pipetting to reach a homogeneous solution with an equivalent optical density $OD_{730nm} = 0.8$, as this was the value measured in our liquid culture at the start of the exponential growth phase. For volumetric printing, an equivalent optical density $OD_{730nm} = 0.3$ was used to allow sufficient light transmittance for the fabrication. Choice of initial OD did not impact cell growth (Supplementary Fig. 13a). Biotic bioink $OD_{730nm}$ calculation is described in Supplementary Method 7 and Supplementary Fig. 35.

## Bioink characterization

The bioink was characterized using strain-controlled shear rheometer (MCR 502; Anton-Paar). The samples were loaded on a glass bottom plate with a light source ($\lambda = 405$ nm, $I = 8$ mW cm$^{-2}$; M405L3, Thorlabs) underneath and measured using 20 mm plate-plate probe with 0.8 mm gap size at 25 °C. Cross-linking kinetics of the abiotic gels was evaluated by using oscillatory time sweep ($\omega = 10$ rad s$^{-1}$; $\gamma = 0.1\%$). The samples were irradiated after 60 s of initial oscillation to induce photopolymerization of F127-BUM in the gel mixture until the plateau storage modulus $G'$ was reached. Bioink printability characterization is described in Supplementary Method 8.

UV-vis spectra of the medium and hydrogel samples were measured using a plastic cuvette (1 cm light path) with a UV-visible light spectrophotometer (Lambda 35, Perkin Elmer). The transmittance was measured over the range of 800-400 nm.

## Biofabrication

3D samples for the measurements were fabricated using a pneumatic-driven direct ink writing 3D printer (BioX, Cellink). The bioink was loaded into 3 mL cartridges with a 22 G conical nozzle ($\varnothing = 0.41$ mm) and the cartridge was heated to 37 °C for 5 min prior to printing. The printing was performed at a pressure range of $P = 40-60$ Pa and a speed of $v = 5-10$ mm s$^{-1}$. Single layer circular disc-shaped samples ($\varnothing = 1$ cm, $h = 150-350$ μm) were modeled in Rhinoceros 3D and G-code generation was done via Slic3r. The more complex lattice structures were designed with Rhinocerous 3D (Rhino7) and Grasshopper software. The printed disc-shaped samples were cross-linked for 1 min using an LED light source ($\lambda = 405$ nm light, $I = 8$ mW cm$^{-2}$, Thorlabs GmbH, Germany), followed by a 10 min washing step in BG11–ASNIII medium. The washing medium was discarded, and fresh medium was added to the samples for bacteria growth.

For volumetric printing, a Tomolite tomographic bioprinter (Readily3D, Switzerland) was used[64,65]. All STL models were generated with Rhino7 and Grasshopper software. The size constraints for the Tomolite vials were up to 25 mm in height and 14 mm in diameter. The lattice shown in Fig. 4a, b, is 21.85 mm tall and 11.85 mm in diameter, with a volume of 101.87 mm$^3$ and a surface area of 2145.84 mm$^2$. The lattice is radially arrayed into 16 sectors, in 3 concentric rings, and 20 vertical segments. This generated a lattice with 960 pores, with pore diameter from 0.5–1.0 mm, and strut diameters from 0.15–0.70 mm (median diameter 0.20–0.24 mm). The lattices were sliced into tomographical projection planes using a commercial software (Apparite, Readily3D, Switzerland). Bioink (4 °C) was loaded into autoclaved glass printing vials (outer diameter $\varnothing = 20.0$ mm, inner diameter $\varnothing = 15.0$ mm) and solidified at 25 °C for 5 min before printing. Cross-linking of the bioink was induced by a light source with a 405 nm wavelength with a pre-calibrated light dosage of 94 mJ cm$^{-2}$ (Supplementary Method 9). The average light intensity during the printing process was set to be 8 mW cm$^{-2}$. After printing, the vial was cooled to 4 °C to liquify the unpolymerized bioink. The retrieved polymerized structure was immersed in culturing medium to allow removal of excess LAP and

residual bioink. The sample was incubated beyond one year with the above-mentioned culturing conditions.

## Living material incubation conditions

The printed circular disk-shaped samples were cross-linked for 1 min using an LED light source ($\lambda = 405$ nm, $I = 8$ mW cm$^{-2}$, Thorlabs GmbH, Germany), followed by a 10 min washing step in BG11–ASNIII medium. After that, printed samples were incubated under the same light conditions as liquid culture. The samples were incubated in BG11–ASNIII medium for 5 days with a change to fresh medium on day 2. The old medium was collected at each change for pH measurements. Fresh BG11–ASNIII medium with an additional 5 mM CaCl$_2$ was added as the calcium-rich culturing medium. The medium was changed and collected every 5 days (on day 5, 10, 15, 20, and 25) until day 30. The collected medium pooled from individual samples ($n = 20$ samples, biological replicates) was used for pH measurements to evaluate total dissolved inorganic carbon as fully described in the Supplementary Method 1. The samples were inspected using optical microscope (Panthera Classic, Motic) equipped with Plan UC 10x/0.25, 40x/0.65 Ph2 and 100×/1.25 oil objectives and camera (Moticam S, Motic). The images were captured with Motic Images Plus 3.090 software.

## Biomass quantification and cell viability characterization

To measure the sample mass change, 5 of the freshly printed (day 0) abiotic and biotic disc samples were washed in Milli-Q water for 10 min to remove salt deposited on the surface from the culturing medium. The samples were then dried under a constant laminar flow in a 1.5 mL microcentrifuge tube of known mass to evaporate the water entrapped within the material. The dried samples together with the microcentrifuge tube were weighed and the mass was used as the initial benchmark for characterization. On day 5, 10, 20 and 30, abiotic and biotic samples ($n = 5$; biological replicates) were randomly selected and washed with the same procedure. The samples were subsequently dried under the same condition before weighing. The mass was normalized to the average mass of the abiotic or biotic discs on day 0.

Cell viability and growth within the hydrogels were evaluated using confocal microscopy for day 1 to 5 and chlorophyll extraction for day 10 to 30 (Supplementary Fig. 36). Dead cells within the printed disc samples were stained using SYTOX Blue dead cell staining. Disc samples were first washed 3 times in 1 mL of MilliQ water then stained using 0.2 μM staining solution diluted with 1x ASNIII medium. The samples were then incubated in the dark for 15 min, washed 3 times with 1x ASNIII medium for 5 min, and stored in the same medium until imaging. The images of SYTOX Blue dead cell staining and cell autofluorescence were acquired with confocal microscopy (TCS SP8, Leica with a 506357 ∞ /0.14 - 0.18/OFN25/E 40x, water objective, 26 slices, 2 μm each). A 405 nm laser was used to excite SYTOX Blue and emission was collected at 450–510 nm (HyD detector). Chlorophyll autofluorescence was excited with a 488 nm laser and was collected at 660–700 nm (HyD detector). Z-stacks of 50.4 μm were used for maximum intensity Z-projections using LAS X (Leica). Separate images ($n = 4$ for day 0, $n = 5$ for day 1, $n = 3$ for day 5–30) of dead cell and autofluorescence were imported in CellProfiler 4.2.5, (https://cellprofiler.org/). Gaussian filter was applied to the images followed by primary object identification using a built-in module (Adaptive Otsu threshold algorithm was applied with 3 classes where middle intensity was considered as background), resulting in a binary image and object count.

Chlorophyll extraction was done on randomly chosen disc samples ($n = 5$ randomly selected samples) every 5 days starting from day 10. The samples were fully immersed in 0.5–1 mL of isopropyl alcohol and homogenized (IKA T25 Ultra Turrax with S25N-8G-ST inset). The slurry was then sonicated for 85 min followed by centrifugation at $19,000 \times g$ for 10 min. Technical triplicates of 100 μL supernatant were transferred into a 96-well plate and the absorption at 663 nm was

measured by UV–Vis plate reader (HIDEX Sense). Suspension samples of different optical densities were used to calculate an equivalent cell concentration from the chlorophyll amount using exponential fitting (Supplementary Fig. 29). A non-linear fitting was chosen as we observed an increase in encapsulated cell concentration with chlorophyll extraction due to the interference of other cyanobacterial pigments, such as phycobiliprotein[66–68].

For suspension and medium samples from the well plate, samples were collected every 5 days and the cell number was measured on a Multisizer 4e (Beckman Coulter) equipped with a 30 μm aperture tube (gain: 4, current: 600 μA). For each solution, 25 μL of media were diluted in 10 mL of 0.2-μm-filtered Isoton II solution within a 25 mL Accuvette (Beckman Coulter).

PCC 7002 encapsulation efficiency was quantified using the cell count and chlorophyll extraction results. The encapsulation efficiency was calculated as the percentage of encapsulated cell number to the total number of encapsulated cells and cells in the culture medium. Leakage was defined as the percentage of cells found in the culture medium to the total cell number of encapsulated cells and cells in the culture medium.

### Inorganic precipitate characterization

Calcium-containing precipitate deposits in circular disc-shaped hydrogels were stained using 40 mM Alizarin Red S solution (pH = 4.2) in Milli-Q water. Each sample was fully immersed in 2 mL of Alizarin Red S solution for 20 s. The sample was then washed in 200 mL of Milli-Q water 3 times for 5 min and imaged using an optical microscope. To quantify the extent of precipitation, we extracted the Alizarin Red S stain from the samples using 10% acetic acid (0.5 mL per sample, $n = 3$, biological replicates) with vortexing for at least 1 h. The solutions were then neutralized using 1 M NaOH and 100 μL of the neutralized extraction solution was used to measure the absorbance at 405 nm using a UV-vis plate reader (HIDEX Sense). A calibration curve was generated using pure $CaCO_3$ following the same method to quantify the amount of $CaCO_3$ in the samples based on the Alizarin Red S absorbance (Supplementary Fig. 15a). Alizarin Red S staining was also performed on randomly chosen recovered wells ($n = 3$, biological replicates) of 24-well plate every 5 days. 0.5 mL of 40 mM Alizarin Red S solution was added to each well for 20 s. Each well was then washed with 0.5 mL Milli-Q water 3 times to collect the precipitates, which were then recovered by centrifugation at $12,300 \times g$ for 10 minutes. The supernatant was then discarded and the remaining precipitate pellet was dissolved in 0.5 mL of 10% acetic acid by pipetting. The solution was neutralized and the absorbance was measured under the same conditions as those of the disc-shaped hydrogel samples.

The presence of carbonates in the precipitates was confirmed by thermogravimetric analysis using a TGA/DSC 3+ (Mettler Toledo). The sample was placed inside a 70 μL alumina crucible and heated from 25 to 900 °C in pure $N_2$ under a constant flow rate of 150 mL min⁻¹. Buoyancy effects were accounted for by performing measurements with an empty crucible. The sample was first heated to 400 °C and held at this temperature for 30 min to remove any bound water, before continuing heating to 900 °C at a heating rate of 5 °C min⁻¹. The same measurement conducted with an empty crucible was taken as the baseline and was subtracted from the measured data.

The extent of carbonate precipitate formation within abiotic and biotic samples was further characterized by thermal decomposition at atmospheric pressure. 5 dry abiotic or biotic samples were loaded into a porcelain crucible and heated to 600 °C (ramp rate = 2 °C min⁻¹) for 2 hours. The samples were then cooled to room temperature and the remaining mass was measured. In addition, the remaining precipitates were collected for attenuated total reflectance Fourier transform infrared spectroscopy (ATR-FTIR, Spectrum Two, PerkinElmer; 1 cm⁻¹ resolution, 32 scans) as well as X-ray diffraction (XRD) analysis with a

PANalytical Empyrean diffractometer (Cu Kα radiation, 45 kV and 40 mA) with an X'Celerator Scientic ultrafast line detector and Bragg-Brentano HD incident beam optics. XRD samples were measured over the $2\theta$ range 10–90° for 1 h with a step increment of 0.016°. Results were compared with reference patterns from the ICDD database. XRD pattern baseline removal for analysis is described in the Supplementary Method 10.

Lyophilized samples (Supplementary Method 11) were used for Scanning Electron Microscope (SEM) characterization. Samples were sputtered with 10 nm of Pt/Pd 80/20 with a metal sputter coater (CCU-010, Safematic, Zizers, Switzerland). SEM imaging was done using a secondary electron detector with a scanning electron microscope JEOL JSM-7100F (accelerating voltage $V = 12$ kV, working distance $WD = 10$ mm) equipped with an Ametek-EDAX EDS detector (Si(Li) 20 mm). Element analysis was performed with the EDAX Team software.

### Living material mechanical properties

Viscoelastic properties of the printed hydrogels at different time points ($n = 3$, biological replicates) during incubation were evaluated using the same rheometer as for the bioink characterization equipped with a Peltier stage and 8 mm sandblasted probe (gap size $h = 0.15$–$0.35$ mm, normal force $F_N = 0.1$ N) at 25 °C. Samples were cut from disc-shaped samples using a metal punch ($\varnothing = 8$ mm) to match the rheometer probe geometry. Dynamic oscillatory frequency sweep measurements were performed in the range of 100 to 0.1 rad s⁻¹ ($\gamma = 0.3\%$). Young's modulus was calculated as

$$E = 2G(1 + \nu) \tag{1}$$

using storage value $G'$ at $\omega = 10$ rad s⁻¹ and an assumption for Poisson ratio of $\nu = 0.5$.

To obtain homogeneous samples for tensile testing, abiotic and biotic bioink were molded into a rectangular-shaped mold (12 mm × 50 mm × 1 mm, width × length × thickness) and cross-linked for 2 min ($\lambda = 405$ nm light, $I = 8$ mW cm⁻²). Samples were then washed and incubated under the same conditions as the disc-shaped samples. Uniaxial tensile testing was performed on the samples ($n = 5$, biological replicates) incubated for 2 and 30 days using a tensile testing machine (Stable Micro Systems) equipped with two parallel metal screw-clamps. A 2 mm notch was cut with a razor blade on one side of the sample prior to the stress-strain measurement. The sample was placed securely between the two parallel clamps with an initial grip-to-grip distance of 300 mm. Stress-strain measurement was performed with a constant stretch speed of 0.5 mm s⁻¹ until failure. The sample toughness was calculated by measuring the area under the stress–strain curve and Young's modulus of the sample was calculated as the slope of the linear regime of the curve (Supplementary Fig. 12).

Compressive tests were conducted on the same tensile testing machine (Stable Micro Systems) equipped with a steel probe ($\varnothing = 0.2$ mm). The compression test was done at a constant deformation rate of 0.05 mm min⁻¹ on a cylinder-shaped photosynthetic living material sample after 400 days of incubation ($\varnothing = 10$ mm, $h = 3$ mm). The compressive modulus was obtained by measuring the slope of the linear regime of the stress–strain curve (Supplementary Table 2).

Statistical analysis of the samples at different time points was performed using two-sided two-sample $t$-tests (OriginPro 2019 software). All values are shown as average with standard deviation or root mean square of standard deviations in cumulative cases.

### Reporting summary

Further information on research design is available in the Nature Portfolio Reporting Summary linked to this article.

## Data availability

Data supporting the findings of this work are available within the paper and its Supplementary Information files. A reporting summary for this article is available as a Supplementary Information file. Source data are provided with this paper.

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

## Acknowledgements

This work was done within the framework of the ALIVE initiative (Advanced Engineering with Living Materials) and funded by the SFA-AM program (Strategic Focus Area – Advanced Manufacturing) by the ETH Board. This work was also funded by the Swiss National Science Foundation (SNSF 200021_184697; MWT) and the NIH-funded Wyoming IDeA Networks of Biomedical Research Excellence Program (P20GM103432; JSO). The authors thank Ines Oberhuber and the Medical Microsystems Laboratory (ETH Zurich, Prof. Simone Schürle) for providing access to their dynamic light scattering instrument and their Multisizer. The authors acknowledge the ScopeM facility and Dr. Anne Greet Bittermann for the help with SEM–EDX imaging.

## Author contributions

D.D., Y.C., and M.W.T. conceived the project, designed the experiments, and wrote the manuscript. D.D. and Y.C. performed the experiments, analyzed the data, and interpreted the results. A.S.L. and B.D. conducted the digital design of the living structures, while A.A. and A.R.S. contributed to the living surface design. F.D. assisted with TGA and XRD experiments and data interpretation. S.B. supported polymer synthesis and characterization. M.B. and X.H.Q. provided support for volumetric printing experiments. Y.C. and M.L. carried out sequestration experiments in a closed chamber. J.S.O. contributed to the design of biomass evaluation. All authors reviewed and edited the manuscript.

## Competing interests

The authors declare no competing interests.
