## [Peer Review file · Nature Communications]

Dual carbon sequestration with photosynthetic living materials

Corresponding Author: Professor Mark Tibbitt

Version 1:

Reviewer comments:

Reviewer #1

(Remarks to the Author)

The authors have performed several additional experiments. My comments are mostly related to the additional data:

1. Figure S11a: The BG11-ASNIII medium was changed every 5 days, so additional carbon source is made available (potentially) each time that the medium is refreshed. How is that taken into consideration?
2. Figure S11a: Why are the values for the DIC in the abiotic medium varying like this with a peak at 20 days? Please also include the number of replicates and error bars.
3. The amount of CO₂ sequestered by precipitation is less than 0.1% of the whole mass of the materials. Based on Figure S16 (biotic culture) and Supplementary text on "CO₂ sequestration by biomass growth (pH change) in living materials", the precipitates' weight is fairly low, so the dual carbon sequestration does not appear really efficient.
4. Concerning the carbon sequestration, is it possible to provide a quantitative analysis of the carbon sequestration contribution from each factor? (e.g., biomass accumulation, carbonate precipitation without the source of sodium bicarbonate in ASNIII-BG11).
5. Figure S30: The CO₂ consumption rate data is quite noisy, and a very low R₂ value (0.15) is shown on panel b. Also, in panel b, is there a reason why the x-axis is from 0.5 to 1.4h? Please also include the number of replicates and error bars.

Minor points:

- The authors use expressions like carbon-negative and carbon-neutral infrastructure in the manuscript. To use such wording, it would require a quantitative cradle-to-grave CO₂ emission analysis (LCA).
- Caption of Figure S11a: Shouldn't that be BG11-ASNIII medium?
- Typo in Figure S11: medium (twice)
- Typo in Figure 2a: medum

Reviewer #2

(Remarks to the Author)

After reading the authors' Response to Reviewers, I appreciate the addition of cell counts and chlorophyll measurements to support the conclusions regarding biomass growth. However, I remain unconvinced that measuring extracellular pH is a valid way to quantify uptake of inorganic carbon into biomass in this system, which is open to the atmosphere. The standard way to quantify photosynthesis in cyanobacteria is to measure O₂ evolution using an oxygen electrode or by membrane inlet mass spectrometry.

I understand that there are several references (cited in the Response to Reviewers) that describe the release of OH⁻ ions by carbonic anhydrase located in the carboxysome as part of a model for the cyanobacterial CCM and MICP, but this is an oversimplification that is not biochemically correct. A paper on quantitative modeling of the CCM in beta-cyanobacteria (Mangan et al., 2016. PNAS 113: E5354-E5362) notes that the combined action of CA and Rubisco in the carboxysome actually produces a net H⁺, not an OH⁻ ion.

The statements on p. 4 ("Here, carbonic anhydrase (CA) within the carboxysome converts the bicarbonate species into CO₂,

releasing a hydroxyl ion (OH-) into the surrounding media^{7,33–35}. The OH- is excreted to the exterior of the bacterium...”) and p. 7 (“Using HCO₃⁻ as a substrate, carbonic anhydrase generates equimolar amounts of OH- ions, which are excreted into the surrounding medium, and CO₂ molecules, which are integrated into bacterial metabolism.^{17,27}”) and the depiction in Fig. 1 are incorrect.

Reviewer #3

(Remarks to the Author)
Please see attached comments.

Version 2:

Reviewer comments:

Reviewer #1

(Remarks to the Author)
The authors have made improvements in the manuscript. However, the authors have still not addressed two critical comments.

Major comments:

1. It's still questionable whether atmospheric CO₂ is truly the primary carbon source for sequestration (lines 183-184). It's not sure whether we can regard these materials as meaningful sequestration system if the authors take initial input [CO₃²⁻] into consideration as “cost” carbon sources.

Maybe I miss something, but when I look at Figure S1a, I find that the amount of carbonate refreshed every 5 days is not an insignificant amount. For example, let's consider the decrease in DIC from day 5 to day 10: Right after the medium refreshment at day 5, [DIC] should be similar to the one at day 0 (i.e. ~900 μmol/L with ~380 μmol/L coming from Na₂CO₃ in the medium). At day 10, [DIC] have reduced to ~600 μmol/L, corresponding to a reduction of 300 μmol/L used for biomass and carbonate precipitates. Why could these 300 μmol/L not come (in significant part) from Na₂CO₃ in the medium? And then from day 15 to 20, it goes from ~900 μmol/L to ~400 μmol/L, and why could these 500 μmol/L “used” not come in large part from Na₂CO₃ in the medium?

The authors should make two things clear in the manuscript or supplementary information: 1] How much CO₃ of carbonate precipitates or biomass originally come from the atmospheric CO₂, at least with clear simulation or calculations. The [CO₃²⁻] supplied every 5 days (refreshed) can be used for both clearly. 2] The contents of Figure S1a should include more details to make it clearer. It would help a lot if this graph also included the datapoints for [DIC] right after the refreshing of the medium. These are critical points, which are essential to convincingly demonstrate the claims of “dual sequestration”. So, once this has been demonstrated, I recommend the data to be included as a main Figure or Table rather than in the S.I.

2. Figure S30: The rate of CO₂ change in a closed chamber is indeed a way to directly demonstrate CO₂ sequestration. However, the uncertainty on this measurement is very high, with quite low R². The addition of 2 controls is needed to validate this result: A control in dark conditions where CO₂ should increase because of respiration of the cells (this data should already be available to the authors since the materials were monitored for 10 days, so I guess also while in dark condition). A control without cells (but in the same chamber and with an initial CO₂ concentration higher than atmospheric) to check whether the sensor reports a constant amount of CO₂ in such conditions, without any leakage, etc. It would be nice to show these controls and calibration data together. The authors can refer to the following paper for further information: Banerjee, S., Siemianowski, O., Liu, M., Lind, K. R., Tian, X., Nettleton, D., & Cademartiri, L. (2019). Stress response to CO₂ deprivation by *Arabidopsis thaliana* in plant cultures. *PLoS One*, 14(3), e0212462.

Also, the authors should display the full data from their CO₂ measurements including the 30min equilibration time and also the data after 1.4h, and explain clearly why the period between 0.5h-1.4h was specifically used to calculate the rate in change of [CO₂].

If these are not possible, I would recommend to just remove figure S30 from the Supplementary data.

Minor comment:

1. In the revised manuscript (Line 173), the authors mentioned “uniform circular samples (V_{sample} = 40 μL, d = 10 mm)”. Is the volume (40 μL) correct here?

Reviewer #2

(Remarks to the Author)
In this revised manuscript, the authors have satisfactorily addressed the comments from my previous review.

Reviewer #3

(Remarks to the Author)
The authors have satisfactorily addressed all of my comments.

Version 3:

Reviewer comments:

Reviewer #1

(Remarks to the Author)

All my comments are properly addressed. I recommend publication.

RESPONSE TO REVIEWERS: All responses are written in blue font and all changes to the main text or supplementary information have been highlighted in yellow.

Reviewer #1 (Remarks to the Author):

The manuscript of Dranseike and co-authors reports on a photosynthetic living material that sequesters CO₂ via two routes: biomass accumulation and MICP. The living material is based on cyanobacteria encapsulated into a Pluronic F127-based hydrogel matrix. The use of Pluronic F127 enables them to 3D-print their living material. Over the course of 30 days of growth, the dry mass of the material is shown to increase. Such increase is then associated biomass growth and MICP. My most major concerns on this study relate to the novelty of this “dual sequestration” and to some technical aspects.

We appreciate the Reviewer’s consideration of our work. By addressing specifically the comments below, we believe that a revised version of our manuscript will articulate more clearly the ‘dual sequestration’ nature of our approach as well as clarify the technical aspects that were not sufficiently clear in the original manuscript.

1. From what I understand, the main novel aspect is this dual sequestration obtained in a living material, i.e. combination of CO₂ sequestration via biomass accumulation and photosynthetic MICP. This manuscript claims that such dual sequestration has not yet been explored in photosynthetic living materials. I am not sure that I agree with this. For example, the work of Reinhardt et al. *Front. Bioeng. Biotechnol.* 2023, which the authors also cite, reports on a living material that is using CO₂ sequestration for biomass accumulation (cell growth) and photosynthetic MICP.

We appreciate the Reviewer noting the main novel aspect of the work, which is to tailor a living material specifically for dual carbon sequestration. We apologize that the original writing gave the impression that systems with both biomass generation and precipitation have not yet been explored. However, we maintain that, to our knowledge, no system has been engineered specifically for efficient dual carbon sequestration using photosynthetic living materials. The work of Reinhardt et al., and others, explored photosynthetic cyanobacteria primarily to reinforce materials via microbial-induced carbonate precipitation (MICP). This process also inherently involves biomass accumulation, but the focus of the materials design has been on the use of precipitation to augment the mechanical properties of the structure and the design was not tailored for efficient dual carbon sequestration.

More specifically, the manuscript by Reinhardt et al. encapsulated photosynthetic cyanobacteria within a printable mixture of alginate, methylcellulose, and sand. Their work focused on the change in mechanical properties of these living building materials with the aim of fabricating an environmentally benign alternative to the carbon emission intensive cement sector. The authors carried out a thorough analysis of the mechanical properties and cell viability of the living building materials and the materials were not engineered for efficient carbon sequestration specifically.

In the present work, we tailored the material design specifically to enable efficient dual carbon sequestration (via both biomass and precipitate generation) of the encapsulated cyanobacteria. This required rational material selection and 3D design that were combined with rigorous analysis of the developed photosynthetic living materials. That said, the work of Reinhardt et al. and others

motivated the feasibility of our approach, and we have reframed the Introduction and Discussion to articulate these points more clearly.

Text in the original version of the Introduction:

“Recently, photosynthetic MICP was used to design living building materials that mineralized over time¹⁷.”

“Drawing inspiration from natural systems, we engineered photosynthetic living materials for dual CO₂ sequestration by immobilizing MICP-capable photosynthetic microorganisms within a printable polymeric network.”

Updated text in the revised version of the Introduction (page 3, 4):

*“Recently, photosynthetic MICP was used to design living building materials that **were reinforced by mineralization over time¹⁷**.”*

*“Drawing inspiration from natural systems **and recent advances in engineered living materials¹⁷**, we engineered photosynthetic living materials for dual CO₂ sequestration by immobilizing MICP-capable photosynthetic microorganisms within a printable polymeric network.”*

Text in the original version of the Discussion:

“The design of our photosynthetic living materials builds upon many recent advances in engineered living materials^{9,13,50,51}. The diverse functionality of microorganisms has enabled the design and application of engineered living materials in various fields. For instance, bacteria-based biosensors are used to detect target molecules, such as glucose or specific pathogens^{52,53}. Living skin patches adopt skin microbiomes to control bacterial outgrowth on the wounded area⁵⁴. Utilizing the bacterial-mineralization process, microorganism-based living building materials can self-heal and regenerate^{16,17,27}. Living materials have also been used to replace conventionally pollution-intensive processes, such as applications in the energy industry, the production of textiles or structural materials, due to their inherent bioremediation abilities, especially through the carbon sequestration capabilities of certain microorganisms⁵⁵⁻⁵⁹.”

Updated text in the revised version of the Discussion (page 14):

*“The design of our photosynthetic living materials builds upon many recent advances in engineered living materials^{9,13,50,51}. The diverse functionality of microorganisms has enabled the design and application of engineered living materials in various fields. For instance, bacteria-based biosensors are used to detect target molecules, such as glucose or specific pathogens^{52,53}. Living skin patches adopt skin microbiomes to control bacterial outgrowth on the wounded area⁵⁴. **Notably, bacterial mineralization processes have been exploited to design microorganism-based living building materials that can self-heal, strengthen over time, and regenerate^{16,17,27}, demonstrating the potential to combine biomass accumulation and mineralization within a single living material.** Living materials have also been used to replace conventionally pollution-intensive processes, such as applications in the energy industry, the production of textiles or structural materials, due to their inherent bioremediation abilities, especially through the carbon sequestration capabilities of certain microorganisms⁵⁵⁻⁵⁹.”*

2. The authors write that, in their living materials MICP is using atmospheric CO₂ as the sole carbon source. I do not agree with this since they use the BG11 medium, which contains sodium carbonate.

We appreciate the Reviewer's comment and agree that identifying the precise carbon source is important and tracking this would require further investigation, such as isotope labeling. Therefore, we revised the manuscript text to state that atmospheric carbon was the main source and include additional experimental data on the carbon balance and equilibrium in the culture medium throughout the sample incubation to support the claim that atmospheric CO₂ is the main source of carbon (see more details in the response to the next comment R1.3). We have also updated the composition table to include the full list of ions from ASN II and BG11 media in the Supplementary Information.

Text in original version (page 13):

"Our photosynthetic living materials function at ambient conditions with atmospheric CO₂ as the sole carbon source, which highlights the ability to use photosynthetic living materials as a complementary approach in a more distributed manner."

Text in the revised version (page 14):

*"Our photosynthetic living materials function at ambient conditions with atmospheric CO₂ as the **main** carbon source, [...]"*

3. The amount of sodium carbonate in BG11 should be considered in their calculations of CO₂ sequestration, also considering that the medium is changed regularly, so more sodium carbonate is accessible to the living material.

To address the comment of the Reviewer, we further elaborated our analysis of the dissolved inorganic carbon equilibria in the Results section of the manuscript and clarified the information about the carbon source to avoid confusion for the reader. In total, our findings demonstrate that the dissolved inorganic carbon in the medium was always higher than the amount supplied by the sodium carbonate in the medium. Therefore, the majority of the CO₂ taken up by the system arose from the atmospheric carbon that equilibrates the system. As it is not possible to track each molecule specifically, we have softened the text in the results to state that the CO₂ sequestration was achieved "with atmospheric carbon as a primary carbon source" (page 7).

We analyzed the culture medium according to the established protocols for inorganic carbon equilibria used in seawater analysis, known as CO₂SYN [ref. 1]. We also added the corresponding data and an extended version of the CO₂SYN calculations in the Supplementary Information (page 7) and a brief version below. As can be seen in **Figure S8**, the dissolved inorganic carbon (both from the culture medium and atmosphere) is in equilibrium. The equilibria depend on medium conditions such as salinity, alkalinity, pH, and temperature. At the end of each 5-day incubation period, we recollected and pooled the media, measured its pH, total alkalinity, and salinity. From that, we were able to calculate the total amount of dissolved inorganic carbon under equilibrium. We can show that in the case of biotic samples, the total amount of dissolved inorganic carbon was greater than the total amount of CO₃²⁻ coming from the medium components despite:

1. the simultaneous occurrence of MICP and;

2. the increase in pH as biomass grows, which decreases the solubility of carbon species overall.

Due to the mentioned equilibria of carbon species, total dissolved inorganic carbon in abiotic control samples was substantially higher than the CO_3^{2-} coming from BG11–ASNIII medium, which did not decrease as much as in the biotic case because the pH is lower (due to the absence of biomass growth and bacterial metabolic activities). This highlights that substantial amounts of atmospheric carbon dissolve in the medium in our open culture system even with the moderate addition of dissolved carbonates in the BG11–ASNIII medium.

Overall, as the total amount of dissolved inorganic carbon in the culture medium throughout the entire incubation period was more than the amount of carbon we supplemented through the addition of sodium carbonate, we are confident that a substantial portion of the carbon in the final system came from atmospheric carbon. Based on our data, the medium components define the initial amount of dissolved inorganic carbon in the medium and either remain or are replenished by atmospheric carbon. On day 25, the total dissolved inorganic carbon was slightly below (343 $\mu\text{mol/kg}$) the amount of inorganic carbon added to the medium in the form of Na_2CO_3 (381 $\mu\text{mol/kg}$), which we associated with the high pH of the media and continuous MICP.

1. Lewis, E. R., & D. W. R. Wallace (1998). Program developed for CO₂ system calculations. No. cdiac: CDIAC-105. Environmental System Science Data Infrastructure for a Virtual Ecosystem (ESS-DIVE)(United States). <https://www.osti.gov/biblio/1464255>.

Previous text in Results section (page 7):

“During the incubation, BG11–ASNIII medium was changed every 5 days. From day 5, the concentration of Ca^{2+} in the medium was set to 8.65 mM via CaCl_2 addition to simulate natural seawater conditions³⁷ and to promote MICP. To highlight the CO_2 sequestration abilities of the living material, no additional HCO_3^- was dissolved in the medium. As such, all HCO_3^- essential for cell metabolism came from the dissolution of gaseous atmospheric CO_2 .”

Updated text in Results section (page 7):

*“During the incubation, BG11–ASNIII medium was changed every 5 days and the dissolved inorganic carbon (DIC) was quantified in the collected samples (see **Supplementary Information, Calculation of dissolved inorganic carbon in BG11-ASNIII medium with CO2SYS program**). From day 5, the concentration of Ca^{2+} in the medium was set to 8.65 mM via CaCl_2 addition to simulate natural seawater conditions³⁷ and to promote MICP. To highlight the CO_2 sequestration abilities of the living material, no additional HCO_3^- was dissolved in the medium. 40.4 mg L^{-1} (0.38 mM) of Na_2CO_3 was dissolved in the medium to simulate natural seawater; however, CO_3^{2-} ions are not directly taken up by cyanobacteria as source of inorganic carbon. Analysis on the total amount of DIC under equilibrium in the culture medium also showed similar or even higher value than the amount of carbon initially dissolved in the form of CO_3^{2-} (**Figure S11a**). The higher DIC capacity of the culture medium confirms the dissolution of atmospheric CO_2 and, therefore, the CO_2 sequestration ability of the living materials using atmospheric CO_2 as a primary carbon source.”*

Figure S11. a) Total amount of dissolved inorganic carbon (DIC) under equilibrium in the collected abiotic (black) and biotic (green) medium. Gray dashed line indicates the amount of dissolved inorganic carbon in the form of Na_2CO_3 that was used in simulated seawater medium (ASNIII). **b)** pH changes of pooled biotic (green) and abiotic (black) medium every 5 days.

4. The pH change was measured in the culture medium around the living material. Have the authors checked that the culture medium is free of cyanobacteria during the whole period of incubation? Cyanobacteria are likely to also grow in this culture solution and to affect the pH. Hence what is measured is (at least in part) the CO_2 sequestration capability of the cyanobacteria in the liquid and not only in the hydrogel.

We appreciate the Reviewer's comment and have added data on cell concentration in hydrogels and surrounding medium to the Supplementary Information. The ratio between encapsulated and escaped cells shows that less than 20% of the total number of the cells escaped. The retention efficiency, calculated based on 1 disc sample immersed in 2 mL of culture medium, was more than 90% for the first 25 days and more than 80% on Day 30 (**Figure S13**). Increase in bacteria leakage/growth on day 30 (18%) might be attributed to the space limitation within the gel as we also observed a slower increase in the number of encapsulated cells potentially reaching a steady state. Additionally, we observed that while the number of escaped cells was relatively constant, the increase in pH of the surrounding culture medium every 5 days was increasing. Furthermore, we exchanged the whole medium every 5 days to limit growth and accumulation of cyanobacteria outside of the hydrogel. Any contribution to the media's pH change from escaped cyanobacteria would be marginal and we thus conclude that the major contribution to the increase in pH came from the encapsulated cells. Therefore, we do not anticipate that this contributes substantially to the overall analysis. The contribution would potentially be even lower in the samples that are only partially immersed in the medium.

Figure S13. Cell count in the printed disc sample quantified with chlorophyll extraction (green) and cell count in the surrounding culture medium (pink) measured via Coulter Counter. The right axis indicates the percentage of cells that leaked into the surrounding culture medium (2 mL) normalized to the total number of cells in 1 printed disc sample (gray), as well as the encapsulation efficiency calculated as 100% – leakage % (brown).

Updated text in Results section (page 7):

“The pH did not change in abiotic control samples throughout the incubation (Figure S11b) and the majority of the encapsulated bacteria remained within the hydrogel throughout the incubation in biotic samples (Figure S13).”

5. Since there is a strong focus on carbon sequestration, I recommend a more rigorous analysis on CO₂ sequestration values. For example, would it be possible to directly measure the evolution of CO₂ used in the atmosphere during material growth? Given the previous point, it is preferable to do this without liquid at the base of the material, or only when there is freshly added culture media (without cyanobacteria).

We thank the Reviewer for the recommendation, and we think this is a great idea. We performed CO₂ sequestration measurements in a closed system. Over the measurement period, we observed a decrease in atmospheric CO₂ level. In brief, a 0.25 mL sample with approximately 10 mL of culture medium was placed in a glass jar in an air-tight closed container (250 mL) with an embedded CO₂ sensor (Sensirion, SEK-SCD41 Multiple Function Sensor) for 10 days. The container was opened every day and was allowed to equilibrate with atmospheric air to replenish the CO₂ within the closed system. The rate of change of CO₂ concentrations were measured during the 12-hour day cycle (Figure S30a). Starting from measurement day 3, the rate of change in CO₂ concentrations within the closed container was below zero, indicative of CO₂

sequestration. The rate of change in CO₂ concentration reached a value of approximately -30 ppm per hour beyond day 5. We also collected a representative snapshot of the CO₂ concentration within the closed container for one of the samples over the course of 1 hour (**Figure S30b**). Average atmospheric CO₂ concentration decreased from approximately 947 ppm to 940 ppm during the measurement time.

The new **Figure S30** is appended below:

Figure S30. a) Rate of change of CO₂ concentration in a 250 mL closed container over a measurement period of 10 days with a 0.25 mL sample and 10 mL culture medium. CO₂ sequestration was observed from day 3 and plateaued beyond day 5. **b)** Representative snapshot of CO₂ evolution (black) in a closed container and linear interpolation of the measurement (blue).

In our future research, we plan to build a continuous measurement system to obtain more accurate read-outs of the rate of CO₂ sequestration of the living materials. We believe that further development on the experimental design will be a valuable area of research for the future work. More accurate analysis on direct CO₂ measurement would require a tailored measurement setup considering all the environmental parameters under dynamic equilibrium. In future work for upscaling the use of photosynthetic living materials, we plan to engineer controlled bioreactors that quantitatively control and monitor CO₂ and O₂ input/output, temperature, humidity, and the supply of culture media. In this manner, we would be able to more quantitatively analyze the CO₂ sequestration directly during growth. We agree that this is an important area of research for future work, especially for the scale-up and translation of these concepts. We would like to emphasize again that in the present work, we relied on extensive post mortem analyses of our photosynthetic living materials to ensure robust quantification of the CO₂ uptake and mineralization of the material, which convincingly demonstrate the activity and importance of the present system.

To address the continued need for more robust quantification of CO₂ uptake during the life cycle of the photosynthetic living materials, we have added the following text to the Discussion.

Text in original version of the Discussion (page 13):

“Implementing photosynthetic living materials in broad application still requires improved usability and upscaling of the material production. This can be achieved by taken advantage of recent advances in biofabrication for the creation of larger scale porous⁶² or granular⁶³ structures.”

Updated text in the revised version of the Discussion (page 15):

“Implementing photosynthetic living materials in broad application still requires improved usability and upscaling of the material production. This can be achieved by taking advantage of recent advances in biofabrication for the creation of larger scale porous⁶² or granular⁶³ structures. Upscaling would provide the means to do more robust CO₂ sequestration analysis using gas composition monitoring (Figure S30) as well as control over carbon sources via media supply.”

6. The pH reaches values above 9 in the growth medium. Was there also calcium precipitation observed in the dish containing the growth medium (outside the hydrogel)?

We observed calcium precipitates in the culture dish (24-well plates). However, the calcium precipitation in the well plates did not affect our measurements of the quantification of the CO₂ sequestration ability of the living materials as all samples were transferred to fresh, unused wells prior to any analysis.

While there were minimal cyanobacteria observed in the growth medium outside of the hydrogel during the incubation period of 30 days (see R1.4), we did observe calcium precipitation in the well plates of biotic samples as the well plate surface was negatively charged and could serve as additional nucleation sites. We also quantified the difference in the amount of precipitates on the well plates (samples transferred every 5 days to new wells) between biotic and abiotic samples throughout the incubation period (Figure S15). Again, it should be noted that these additional precipitates were not counted in the quantification of the CO₂ sequestration ability of the living materials.

Figure S15. a) Calibration curve to calculate equivalent CaCO₃ amount from the absorbance of Alizarin Red S staining. **b)** Equivalent CaCO₃ amounts calculated as difference in Alizarin Red S staining of the precipitates in the biotic and abiotic wells normalized per milliliter of hydrogel. The printed disc samples were transferred into new well plates every 5 days and Alizarin Red S quantification was done on the recovered well plates.

To quantify the CO₂ sequestration, we carried out extensive post mortem analyses on all hydrogel samples and that the calcium carbonate precipitates within the well plate or the growth medium

did not affect the analysis of carbon sequestration within the photosynthetic living materials, as our samples were analyzed only after transfer to a new well thorough washing to remove any residual salts on the surface from the growth medium prior to burning the construct itself and measuring quantitatively the mineral phase within the material. The presence of additional carbonate precipitates in the surrounding environment of biotic samples, that were not accounted for in our calculations, suggests that the CO₂ sequestration ability of the photosynthetic living materials may even go beyond the amount of CO₂ sequestered within the living material itself depending on the local growth environment.

Sentence added to the revised version of the Results section of the manuscript (page 8):

“Biotic samples stained red by day 10, indicating Ca²⁺ accumulation in the whole volume of the hydrogel and the dark red color was observed around growing bacteria clusters over time. From the analysis of Alizarin Red S staining, we observed that the amount of precipitates was equivalent to > 1 mg of CaCO₃ (mL⁻¹ of hydrogel) already after 10 days (Figure 2e). It reached 4.8 ± 2 mg of CaCO₃ (mL⁻¹ of hydrogel) by the end of the 30-day period. Precipitates were also observed in the bottom of the incubation wells, and all analyses were carried out in fresh wells to avoid any influence on precipitation outside of the living material on our quantification (Figure S15).”

Updated Methods section about Alizarin Red S staining quantification (page 18):

“Calcium-containing precipitate deposits in circular disc-shaped hydrogels were stained using 40 mM Alizarin Red S solution (pH = 4.2) in Milli-Q water. Each sample was fully immersed in 2 mL of Alizarin red S solution for 20 s. The sample was then washed in 200 mL of Milli-Q water 3 times for 5 min and imaged using an optical microscope. To quantify the extent of precipitation, we extracted the Alizarin Red S stain from the samples using 10% acetic acid (0.5 mL per sample, n = 3) with vortexing for at least 1 hour. The solutions were then neutralized using 1 M NaOH and 100 μL of the neutralized extraction solution was used to measure the absorbance at 405 nm using a UV-vis plate reader (HIDEX Sense). A calibration curve was generated using pure CaCO₃ following the same method to quantify the amount of CaCO₃ in the samples based on the Alizarin Red S absorbance (Figure S15a). Alizarin Red S staining was also performed on randomly chosen recovered wells (n = 3) of 24-well plate every 5 days. 0.5 mL of 40 mM Alizarin Red S solution was added to each well for 20 s. Each well was then washed with 0.5 mL Milli-Q water 3 times to collect the precipitates, which were then recovered by centrifugation at 12,300 rcf for 10 minutes. The supernatant was then discarded and the remaining precipitate pellet was dissolved in 0.5 mL of 10% acetic acid by pipetting. The solution was neutralized and the absorbance was measured under the same conditions as those of the disc-shaped hydrogel samples.”

7. What is the viability of bacterial cells within gels after post-processing (e.g., printing, curing at 405 nm)? The authors use a 22G conical nozzle, which could have high shear stress exerted on cells. And the cells are exposed to UV during the curing.

We agree that this is a valuable set of data to include in the manuscript to guide the design of photosynthetic living materials, and therefore, we added data on cell viability after the printing

process to the Supplementary Information. To confirm that the bacterial cells survive the printing process (extrusion and photo-cross-linking), we include viability data for day 0 and day 1 (**Figure S2**). The cells were viable ($> 50\%$) after printing and the population recovered to $> 85\%$ viability by day 1. Given this, we can tailor the initial concentration of cyanobacteria to ensure a robust population of the sample at day 1 that remains viable and grows for over 1 year.

We also agree that the photo-cross-linking step using 405 nm light is likely one factor that impacts cell viability during printing, and therefore added the information on the impact of different light intensities on the reaction rate and cell viability to the Supplementary Information. We observed substantially compromised cell viability when using light intensity $I = 12 \text{ mW cm}^{-2}$ (**Figure S10a**). Therefore, we chose to use a light intensity of $I = 8 \text{ mW cm}^{-2}$ for our experiments to minimize photo-cross-linking time while achieving sufficient cell viability (**Figure S10b**).

The Supplementary Figures are appended below:

Figure S2. a) Cell viability on day 0 and day 1 after bioprinting. **b)** Representative maximum intensity Z-projection images of dead cell staining (cyan) and cell autofluorescence (green). Scale bar, $100 \mu\text{m}$.

Figure S10. a) Cell viability on day 5 after photo-cross-linking the bioink at different light intensities (SYTOX Blue dead cell staining, cyan; cell autofluorescence, green; maximum intensity Z-projection). Scale bar, $100 \mu\text{m}$. **b)** Bioink photo-cross-linking at different light intensities of 4, 8, and 12 mW cm^{-2} ($\lambda = 405 \text{ nm}$; light turned on at $t = 60 \text{ s}$).

Previous section in the Methods (page 16):

“Biomass characterization

5 of the freshly printed (day 0) abiotic and biotic disc samples were washed in Milli-Q water for 10 min to remove salt deposited on the surface from the culturing medium. The samples were then dried under a constant laminar flow in a 1.5 mL microcentrifuge tube of known mass to evaporate the water entrapped within the material. The dried samples together with the microcentrifuge tube were weighed and the mass was used as the initial benchmark for characterization.

On day 5, 10, 20 and 30, five of the abiotic and biotic samples were randomly selected and washed with the same procedure. The samples were subsequently dried under the same condition before weighing. The mass was normalized to the average mass of the abiotic or biotic discs on day 0.”

Updated Methods section (pages 17–18):

“Biomass *quantification and cell viability* characterization

To measure the sample mass change, 5 of the freshly printed (day 0) abiotic and biotic disc samples were washed in Milli-Q water for 10 min to remove salt deposited on the surface from the culturing medium. The samples were then dried under a constant laminar flow in a 1.5 mL microcentrifuge tube of known mass to evaporate the water entrapped within the material. The dried samples together with the microcentrifuge tube were weighed and the mass was used as the initial benchmark for characterization. On day 5, 10, 20 and 30, 5 of the abiotic and biotic samples were randomly selected and washed with the same procedure. The samples were subsequently dried under the same condition before weighing. The mass was normalized to the average mass of the abiotic or biotic discs on day 0.

Cell viability and growth within the hydrogels were evaluated using confocal microscopy and chlorophyll extraction (Figure S34). Dead cells within the printed disc samples were stained using SYTOX Blue dead cell staining. Disc samples were first washed 3 times in 1 mL of MilliQ water then stained using 0.2 μ M staining solution diluted with 1x ASNIII medium. The samples were then incubated in the dark for 15 min, washed 3 times with 1x ASNIII medium for 5 min, and stored in the same medium until imaging. The images of SYTOX Blue dead cell staining and cell autofluorescence were acquired with confocal microscopy (TCS SP8, Leica with a 506357 \times /0.14 - 0.18/OFN25/E 40x, water objective, 26 slices, 2 μ m each). A 405 nm laser was used to excite SYTOX Blue and emission was collected at 450–510 nm (HyD detector). Chlorophyll autofluorescence was excited with a 488 nm laser and was collected at 682–720 nm (HyD detector). Z-stacks of 50.4 μ m were used for maximum intensity Z-projections using LAS X (Leica). Separate images (n = 3–5) of dead cell and autofluorescence were imported in CellProfiler 4.2.5, (<https://cellprofiler.org/>). Gaussian filter was applied to the images followed by primary object identification using a built-in module (Adaptive Otsu threshold algorithm was applied with 3 classes where

middle intensity was considered as background), resulting in a binary image and object count.

Chlorophyll extraction was done on randomly chosen disc samples ($n = 5$) every 5 days starting from day 10. The samples were fully immersed in 0.5–1 mL of isopropyl alcohol and homogenized (IKA T25 Ultra Turrax with S25N-8G-ST inset). The slurry was then sonicated for 85 min followed by centrifugation at 19000 rcf for 10 min. Technical triplicates of 100 μ L supernatant were transferred into a 96-well plate and the absorption at 663 nm was measured by UV-vis plate reader (HIDEX Sense). Suspension samples of different optical densities were used to calculate an equivalent cell concentration from the chlorophyll amount (Figure S25).[†]

Figure S25. **a)** Calibration curve fitting to relate chlorophyll absorption at OD_{663nm} to cell suspension OD_{730nm} (black) and calibration curve fitting to relate cell suspension OD_{730nm} and cell number measured via particle counting (black). **b)** Cell concentration in liquid suspension (particle counting, black) compared to average cell concentration in encapsulated samples obtained via confocal microscopy max projection (light green) and chlorophyll extraction (dark green). **c)** Comparison of equivalent OD_{730nm} of encapsulated cells (green) and actual OD_{730nm} of liquid suspension (black) from day 10.

8. The modulus and the toughness of the material increased after 30 days, but the values are considerably lower than wood-based materials for example (another photosynthetic living materials with CO₂ sequestration). It would be useful if the authors compare the values to other materials, including those based on ureolytic MICP for comparison.

We thank the Reviewer for this suggestion; however, we feel that a comparison of the mechanics of these materials to wood-based alternatives or any other material would distract from the central line of the paper. Our aim was solely to design a photosynthetic living material for long-term CO₂ sequestration, and the increase in mechanical properties was a “by-product” of the calcium carbonate precipitation pathway and further proof that the material was filling with a substantial mineral phase. This photosynthetic living material was not designed for load-bearing but specifically for efficient (compared with chemical processes), permanent (compared with wood materials), and environmentally-friendly (compared with ureolytic bacteria) CO₂ sequestration method. The work of Reinhardt et al., described above, nicely demonstrated how mineralization joining sand particles can be used to enhance mechanical properties and we have referred to this nice demonstration in several places. In order to focus our work on dual carbon sequestration and distinguish it from Reinhardt et al. and other studies in the field, we prefer to de-emphasize the mechanical reinforcement aspects of our work and further highlight the carbon sequestration.

9. A ratio of 70% CaCO₃ and 30% MgCO₃ is obtained from the shift of the main diffraction peak representing the calcite (104) plane. To which level of accuracy can this approach measure ratios in carbonate precipitates?

Ca to Mg atom % ratios of EDX analysis (number of analyzed samples n = 10) was 78 ± 5 % to 22 ± 5 %, which indicates lower Mg amount in the mixture. Therefore, the 70% CaCO₃ and 30% MgCO₃ ratio is potentially an underestimation of CaCO₃ amount in the mixture. However, if we compare the peak position of (104) plane to carbonates with increasing Mg content, the peak position is also similar to the pattern of dolomite, which has 50% Ca and 50% Mg. We used a ratio of 70% Ca and 30% Mg as a conservative estimate of the CaCO₃ content given the EDX quantification. We compared the difference in sequestered CO₂ calculations between pure CaCO₃ and carbonate with 70% Ca and 30% Mg in the Supplementary Information, resulting in a calculated difference of 5 %, indicating that the precise ratio of Ca:Mg did not have a substantial effect on the quantification.

In this work, we aimed to produce crystalline precipitates that are less soluble than amorphous ones to provide more stable carbon storage. We rephrased the Results section to emphasize this aspect instead of identifying a particular crystalline structure. We also include an extended comparison to potential components of the precipitates in **Figure S18** also showing aragonite as another CaCO₃ form that could be in the precipitates.

Figure S18. XRD pattern of the precipitates after sample thermal decomposition at 600 °C and reference XRD patterns of aragonite, magnesite, dolomite and calcite⁷.

Text in the Results section of the original manuscript (page 9):

“Having demonstrated that photosynthetic living materials are capable of CO₂ sequestration via both biomass accumulation and carbonate formation, we further investigated the composition of the mineral phase produced during incubation. MICP results in the formation of calcium carbonates that vary in stability and solubility with calcite being the most stable polymorph and, therefore, desirable for carbon sinking purposes (**Figure 3a**)⁴¹. XRD analysis of precipitates in biotic samples revealed a highly crystalline structure with the main peaks corresponding to calcite structure with a noticeable shift of the main diffraction peak representing the (104) plane (**Figure 3b**). The observed shift likely indicated compositional variation due to the incorporation of magnesium in the carbonate structure as previously observed in marine environments and, in this case, due to the use of simulated seawater (ASNIII medium)⁴². The XRD was obtained after thermal decomposition (T = 600 °C, which is below the decomposition temperature of CaCO₃⁴⁰) as the signal from the F127-based matrix was much stronger than the one of the precipitates (**Figure S10**).

Precipitates distributed throughout the polymeric matrix (**Figure 3c**). Energy dispersive X-Ray analysis (EDS) indicated mainly carbon and oxygen in the abiotic matrix whereas peaks of calcium and magnesium were prominent in the areas with precipitates in biotic samples. Elemental mapping corroborated these data with calcium, magnesium, carbon, and oxygen in overlapping regions near clusters of cyanobacteria, suggesting the pericellular formation of carbonates (**Figure 3d**).”

Text in the revised version of the Results section (page 10):

“Having demonstrated that photosynthetic living materials are capable of CO₂ sequestration via both biomass accumulation and carbonate formation, we further investigated the composition of the mineral phase produced during incubation. MICP results in the formation of **crystalline** carbonates that vary in stability and solubility with calcite being the most stable polymorph and, therefore, desirable for carbon sinking purposes (**Figure 3a**)⁴¹. **TGA analysis confirmed carbonate presence in the precipitates (Figure 3b), which was additionally confirmed by FTIR (Figure S17).** XRD analysis of precipitates in biotic samples revealed a highly crystalline structure with the main peaks corresponding to **calcium–magnesium carbonates indicated by the position of the main** diffraction peak representing the (104) plane (**Figure S18**). The observed shift likely indicated compositional variation due to the incorporation of magnesium in the carbonate structure as previously observed in marine environments and, in this case, due to the use of simulated seawater (ASNIII medium)⁴². The XRD was obtained after thermal decomposition (T = 600 °C, which is below the decomposition temperature of CaCO₃⁴⁰) as the signal from the F127-based matrix was much stronger than the one of the precipitates (**Figure S19**).

Precipitates distributed throughout the polymeric matrix (**Figure 3c**). Energy dispersive X-Ray analysis (EDS) indicated mainly carbon and oxygen in the abiotic

matrix whereas peaks of calcium and magnesium were prominent in the areas with precipitates in biotic samples. Elemental mapping corroborated these data with calcium, magnesium, and oxygen enrichment in overlapping regions near clusters of cyanobacteria, suggesting the pericellular formation of carbonates (**Figure 3d and S20**). EDX measurements also indicated that Ca to Mg atom % ratio in the area of the precipitates was $78 \pm 5 \%$ to $22 \pm 5 \%$ (number of analyzed samples $n = 10$), which confirmed that both elements were embedded in the precipitates.”

Reviewer #2 (Remarks to the Author):

This manuscript describes a system in which *Synechococcus* sp. strain PCC 7002 cells are encapsulated in a hydrogel matrix, where they evidently grow via photosynthetic uptake of CO₂ and simultaneously promote formation of insoluble carbonates (e.g. calcium carbonate). Although this is an interesting system, I am not convinced by the indirect measurements of CO₂ sequestration via measurement of extracellular pH.

We are grateful for the Reviewer's remarks about our living material system for CO₂ uptake. We now provide more detailed information about the biomass growth as one of the carbon sequestration mechanisms based on imaging and cell viability data in the updated version of the manuscript, to support the measurements based on extracellular pH.

Major comments:

1. On p. 4, the authors state “carbonic anhydrase (CA) within the carboxysome converts the bicarbonate species into a hydroxyl ion (OH⁻) and CO₂.” On p. 7, this is restated as “Using HCO₃⁻ as a substrate, carbonic anhydrase generates equimolar amounts of OH⁻ ions, which are excreted into the surrounding medium.” These statements are incorrect. CA catalyzes the conversion of bicarbonate (HCO₃⁻) and H⁺ to CO₂ and H₂O. OH⁻ is not a product of the reaction. Overall, there should be an uptake of H⁺ through CA activity, which will increase the pH, but this is a very indirect way to measure photosynthetic CO₂ assimilation, and there are many other microbial processes that could affect the pH of the extracellular medium.

We thank the Reviewer for raising this point and we apologize for having not explicitly described the function of carbonic anhydrase in the context of marine β-cyanobacteria PCC 7002 clearly in the original version of the manuscript. The function of carbonic anhydrase (CA) within the carboxysome of PCC 7002 is indeed to convert HCO₃⁻ to CO₂^{1,2}. However, according to the body of literature describing carbon concentration mechanisms of cyanobacteria¹⁻⁷, the species that perform MICP do it in this way:

- CO₂ uptake as HCO₃⁻ into the cell occurs via active transport where CA enzymes convert CO₂ into HCO₃⁻ at a uniport and then HCO₃⁻ diffuses into the carboxysome. The equilibrium of this reaction is, as described by the Reviewer, $\text{CO}_2 + \text{H}_2\text{O} \rightleftharpoons \text{H}^+ + \text{HCO}_3^-$.
- However in the carboxysome, CA converts HCO₃⁻ into CO₂, liberating OH⁻ ions that are then released from the cell.

As a result, the pH increases in the sheath, which shifts the dissolved carbon equilibrium towards HCO₃⁻ conversion to CO₃²⁻, favoring CaCO₃ nucleation within the sheath.

Due to this mechanism in the carboxysome of PCC 7002—corroborated by the references below—we believe that it is reasonable to do a first order estimation of the amount of carbon integrated into biomass during growth based on the amount of hydroxyl ions released. To corroborate this approach, we conducted cell count with confocal microscopy as well as chlorophyll extraction to benchmark the increase in biomass throughout the 30 days. Cell count and chlorophyll extraction further supports the increase in biomass previously quantified by the increase in the pH due to biomass growth, which was positively correlated with the amount of CO₂ sequestered via biomass accumulation.

1. Riding, R. (2006). Cyanobacterial calcification, carbon dioxide concentrating mechanisms, and Proterozoic-Cambrian changes in atmospheric composition. *Geobiology* 4(4), 299-316.
2. Kamennaya, N., Ajo-Franklin, C., Northen, T. & Jansson, C. (2012). Cyanobacteria as biocatalysts for carbonate mineralization. *Minerals* 2, 338–364.
3. Reinhardt, O., Ihmann, S., Ahlhelm, M., & Gelinsky, M. (2023). 3D bioprinting of mineralizing cyanobacteria as novel approach for the fabrication of living building materials. *Frontiers in Bioengineering and Biotechnology*, 11, 1145177.
4. Garg, R., Garg, R., & Eddy, N. O. (2023). Microbial induced calcite precipitation for self-healing of concrete: a review. *Journal of Sustainable Cement-Based Materials*, 12(3), 317-330.
5. Achal, V., Mukherjee, A., Kumari, D., & Zhang, Q. (2015). Biomineralization for sustainable construction—A review of processes and applications. *Earth-science reviews*, 148, 1-17.
6. Dittrich, M., & Obst, M. (2004). Are picoplankton responsible for calcite precipitation in lakes?. *AMBIO: A Journal of the Human Environment*, 33(8), 559-564.
7. Merz, M. U. E. (1992) The biology of carbonate precipitation by cyanobacteria. *Facies* 26, 81–101.

We adjusted the manuscript text to be more specific about the function of carbonic anhydrase in cyanobacteria PCC 7002 and release of OH⁻ ions that contributes to the MICP process. In addition, we removed the illustrations of chemical reactions from **Figure 2** to avoid confusion for the reader.

Text in the original manuscript:

“For carbon concentration to occur, CO₂ from the surrounding environment dissolves in aqueous solutions and forms bicarbonate ions (HCO₃⁻) that are then transported to the bacterial carboxysome³³. Here, carbonic anhydrase (CA) within the carboxysome converts the bicarbonate species into a hydroxyl ion (OH⁻) and CO₂.”

Updated text in the revised version of the manuscript (page 4):

“For carbon concentration to occur, CO₂ from the surrounding environment dissolves in aqueous solutions and forms bicarbonate ions (HCO₃⁻) that are then transported to the marine β-cyanobacteria 7002 carboxysome³³. Here, carbonic anhydrase (CA) within the carboxysome converts the bicarbonate species into CO₂, releasing a hydroxyl ion (OH⁻) into the surrounding media^{7,33–35}.”

Updated Figure 2:

Figure 2. Dual carbon sequestration in photosynthetic living materials.

2. One example of a microbial process that affects pH is the calcification reaction itself, which releases 2 H⁺ for each CaCO₃ formed.

We thank the reviewer for this comment. We agree that any release of H⁺ into the growth medium from precipitation or other biological processes would affect the measurements based on pH change in the medium. However, in this context, release of H⁺ ions into the growth medium should be compensated by the release of OH⁻ ions during the bacterial metabolic activities and we observed pH increase after each media change every 5 days. Therefore, the surrogate pH measurement conducted in the original manuscript is more likely to be an underestimation of the actual amount of biomass produced. To further support our findings on biomass growth, we have corroborated the original biomass quantification via pH measurement with additional quantification of biomass accumulation based on cell count with confocal microscopy and

chlorophyll extraction in the revised manuscript (**Figure S29**). With these, we also observed an increase in cell number and high viability, which positively correlated with the amount of biomass production and CO₂ sequestration. We still include the data of cumulative CO₂ sequestration as biomass calculated from pH measurements reported in the original **Figure 2d** as pH allowed us to estimate the order of magnitude of this process because it is not possible to quantify it from the cell number directly.

3. Cyanobacterial growth was also assessed qualitatively by microscopy. Increases in cell number should be quantified by cell counting.

We thank the reviewer for this suggestion and agree that quantifying cyanobacteria growth would further support the conclusions around biomass accumulation, as described above. We performed individual cell counting in the hydrogels during days 0 and 1 of the incubation using autofluorescence images taken via confocal microscopy. However, the cells later grew in clusters (**Figure S12**) and accurate evaluation of their number was no longer possible after the first days of incubation. Therefore, after day 5, we quantified the cell number using chlorophyll extraction. We observed increasing cell number throughout the 30 day incubation (**Figure S29b**) with an initial rapid growth rate up to day 15 followed by a slower growth rate from day 15 to day 30. We have updated the Method section accordingly and we also added cell growth data to updated **Figure 2d**.

Supplementary **Figures S12** and **S29** appended below:

Figure S12. Representative sample maximum intensity Z-projection images at different time points (cell autofluorescence, green). Scale bar, 100 μ m.

Figure S29. a) Calibration curve fitting to relate chlorophyll absorption at OD_{663nm} to cell suspension OD_{730nm} (black) and calibration curve fitting to relate cell suspension OD_{730nm} and cell number measured via particle counting (black). **b)** Cell concentration in liquid suspension (particle counting, black) compared to average cell concentration in encapsulated samples obtained via confocal microscopy max projection (light green) and chlorophyll extraction (dark green). **c)** Comparison of equivalent OD_{730nm} of encapsulated cells (green) and actual OD_{730nm} of liquid suspension (black) from day 10.

Updated **Figure 2** (panels d,e):

Figure 2. d) An increase in cell concentration was observed throughout the 30-day incubation in the biotic samples, with an initial rapid growth rate up to day 15 followed

by a slower growth rate from day 15 to day 30 (green). Biotic samples continuously sequestered CO₂ over the 30-day incubation in the form of biomass accumulation (~0.31 mg CO₂ per g of gel, black). e) Both biomass growth and CaCO₃ precipitation contributed to the increase in dry mass of the biotic samples. The cumulative CO₂ sequestration by biotic samples was reflected by the difference in normalized dry mass between biotic and abiotic samples (black). Over the incubation period of 30 days, biotic samples accumulated 36% more dry mass than the abiotic samples. Alizarin red S staining on the biotic samples showed a total of 4.8 mg CaCO₃ precipitation per g of gel (red).

Updated Methods section (Page 17):

Biomass quantification and cell viability characterization

“<...>

Chlorophyll extraction was done on randomly chosen disc samples (n = 5) every 5 days starting from day 10. The samples were fully immersed in 0.5–1 mL of isopropyl alcohol and homogenized (IKA, T25 Ultra Turrax with S25N-8G-ST inset). The slurry was then sonicated for 85 min followed by centrifugation at 19000 rcf for 10 min. Technical triplicates of 100 µL supernatant were transferred into a 96-well plate and the absorption at 663 nm was measured by UV-vis plate reader (HIDEX Sense). Suspension samples of different optical densities were used to calculate an equivalent cell concentration from the chlorophyll amount (Figure S29a).”

Minor comment:

4. Abstract: “scalability nature” should be “scalable nature”

Thank you. We updated the text in the Abstract accordingly.

Reviewer #3 (Remarks to the Author):

In this manuscript, 3D printed hydrogel-based materials were developed and evaluated as novel tools for dual carbon sequestration. Photosynthetic living materials were engineered by embedding cyanobacteria within a bioinert hydrogel matrix, to achieve effective dual carbon sequestration through biomass accumulation and irreversible microbially-induced calcium carbonate precipitation. The authors investigated the platform performance not only in the short-term (1 month) but also conducted long-term studies to demonstrate the platform ability in forming biomass and producing mineral precipitation for over a year.

This original study brings interesting findings and useful knowledge to the field. In terms of novelty of the engineering research approach, similar photosynthetic 3D printed living materials have been recently reported in the literature (Datta, Weiss, Wangpraseurt, et al. Nat. Commun. 14, 4742, 2023). Regardless, the system novelty mainly relies in its application, as showed by the platform's potential for concomitant biomass accumulation and irreversible microbially-induced calcium carbonate precipitation over a year timepoint. However, at the current manuscript state, several questions related to the system performance still need to be addressed.

The reviewer believes that, because of lack of quantitative evaluation supporting the system performance, the manuscript is not suitable for publication in Nature Materials at its current state. The manuscript may be considered for publication in other Nature family journals after addressing the following comments.

We thank the reviewer for acknowledging the novelty and the application oriented aspects of our work and its potential in the field. As stated by the Reviewer, additive manufacturing has been applied in the field of engineering with living materials and was our processing technique of choice specifically as it allows the encapsulation of microorganisms together with the fabrication of complex geometries needed for efficient growth of photosynthetic microorganisms. Successful CO₂ sequestration requires not only suitable hydrogel material but also porous structures with large light-exposed surface areas to overcome limitations posed by bulk materials.

Technical comments

1. Hydrogel formulation – The authors identified a hybrid Pluronic F127 formulation composed by pristine F127 (13.2 wt %) and methacrylated F127 (7.3 wt %), which resulted in optimal cell viability and printability through direct ink writing and light-based additive manufacturing. However, little information is reported about how the formulation was optimized.

We appreciate the Reviewer's feedback and prepared the explanation regarding the rationale behind our formulation. We expand on the specific rationale in the sub-points below and included additional data to support these points to the Supporting Information.

a. What is the rationale behind the selection of the final F127:F127-BUM ratio?

The specific overall polymer content was selected so that we had the lowest polymer content that maintained suitable printability. We now include a comparison of the printability of F127-based hydrogels of different polymer contents (**Figure S5**) in the Supporting Information. Hydrogels with polymer content >20 wt% could be printed via direct ink writing and the printing resolution

increased with polymer concentration. Therefore, we used a total polymer content of 20.5 wt% as it had the lowest polymer content while remaining printable.

Figure S5. Printability of Pluronic F127 hydrogels of different polymer contents (22G nozzle, printing speed $v = 5\text{--}15\text{ mm s}^{-1}$, printing pressure $P = 16\text{--}45\text{ kPa}$). Scale bar, 5 mm.

In addition, we compared different F127:F127-BUM ratios to rationalize the specific ratio of F127:F127-BUM used in this work (**Figure S6**). Formulations with low ($<3\text{ wt}\%$) F127-BUM exhibited instability during photo-cross-linking, as observed by the drop in the storage modulus after the light was turned on at $t = 60\text{ s}$. By increasing the concentration of F127-BUM to 7 wt% and above the final modulus and rate of cross-linking increased, and there was no observable benefit of further increases in F127-BUM. Therefore, we selected a final formulation of F127 (13.2 wt%) and F127-BUM (7.3 wt%) to maintain a total polymer content above 20 wt% and to ensure rapid cross-linking and a stable final hydrogel.

Figure S6. Photo-cross-linking of different ratios of F127 and F127-BUM studied as an increase in storage modulus during the reaction. The 405 nm light ($I = 8\text{ mW cm}^{-2}$) was turned on 60 s after the beginning of the measurement.

b. Similarly, how was the final photo initiator (LAP) concentration and cell density determined?

We chose photoinitiator concentration of 0.1 wt% as it is within a range of commonly used concentrations for biofabrication (0.05–0.2 wt%) and resulted in high cell viability and rapid reaction rate. For our material, only a small increase in cross-linking rate was observed for higher

LAP concentrations (0.2 wt%; **Figure S31**). Further, increasing LAP to 0.2 wt% compromised bacteria viability (**Figure S32a**). We decreased the concentration of LAP to 0.05 wt% to slow down the reaction for volumetric printing as the structures were overexposed with 0.1 wt% LAP. Both of the conditions showed similar cell growth in clusters on day 5 (**Figure S32b**).

In literature, starting OD values of 0.3–0.4 have been reported for similar work¹. In our work, we used OD of 0.8 as this was the value measured in our liquid culture at the start of the exponential growth phase. The only different OD we used was in the case of volumetric printing where we decreased it to 0.3 to allow sufficient light transmittance for the fabrication of porous structures. Higher OD values are also possible and the cells show similar growth patterns in all cases (**Figure S33**), and further increasing the starting OD would increase the initial biomass required to prepare the bioink.

1. Heveran, C. M., Williams, S. L., Qiu, J., Artier, J., Hubler, M. H., Cook, S. M., ... & Srubar, W. V. (2020). Biomineralization and successive regeneration of engineered living building materials. *Matter*, 2, 481-494.

The additional Figures that support these points were included in the Supporting Information and are appended below:

Figure S31. a) Photo-cross-linking of the bioink with different photoinitiator LAP concentrations ($\lambda = 405 \text{ nm}$; $I = 8 \text{ mW cm}^{-2}$; light turned on at $t = 60 \text{ s}$). **b)** relative reaction rates of the same hydrogels, calculated based on the time derivative of the storage modulus.

Figure S32. a) Differences in cell growth represented as green color of the living samples on day 5 after photo-cross-linking the bioink at with 0.05, 0.1, and 0.2 wt% LAP. The lowest cell growth was observed with 0.2 wt% LAP in the latter case. **b)** Cell viability on day 5 after photo-cross-linking the bioink at with 0.05 and 0.1 wt% LAP (SYTOX Blue dead cell staining, cyan; cell autofluorescence, green; maximum intensity Z-projection). Scale bar, 100 μm .

Figure S33. Cell viability of samples with different starting cell densities (OD in the range from 0.4 to 1.6) on day 5 (SYTOX Blue dead cell staining, cyan; cell autofluorescence, green; maximum intensity Z-projection). Scale bar, 100 μm .

Formulation optimization and related characterizations should be added to the Supplement.

As described above, we have now included substantial additional supporting data on cell viability and material characterization to the Supporting Information to show how we reached our reported formulation. We have also commented on these choices in the Methods section of the manuscript.

Text in the original Methods section Bioink preparation subsection (page 15):

*Stock solutions of 18 wt% F127 in culturing medium, 30 wt% of F127-BUM (**synthesis described in ESI, F127-bis-urethane methacrylate synthesis**) in Milli-Q water, and 5 wt% LAP (**synthesis described in ESI, LAP synthesis**) in Milli-Q water were mixed at 3:1:0.082 ratio to reach final concentrations of 13.2 wt% F127, 7.3 wt% F127-BUM, and 0.1 wt% LAP. For the bioink preparation, cyanobacteria suspension was centrifuged at 3500 rcf for 5 mins, the supernatant was discarded, and the cell pellet was resuspended in the prepared bioink mixture to reach an equivalent optical density $OD_{730nm} = 0.8$. For volumetric printing, an equivalent optical density $OD_{730nm} = 0.3$ was used. The equivalent OD_{730nm} calculation is described in **Equation S1**.*

Text in the revised Methods section Bioink preparation subsection (page 16):

*Stock solutions of 18 wt% F127 in culturing medium, 30 wt% of F127-BUM (**synthesis described in ESI, F127-bis-urethane methacrylate synthesis**) in Milli-Q water, and 5 wt% LAP (**synthesis described in ESI, LAP synthesis**) in Milli-Q water were mixed at 3:1:0.082 ratio to reach final concentrations of 13.2 wt% F127, 7.3 wt% F127-BUM, and 0.1 wt% LAP. **This formulation was optimized to have the lowest polymer content and stable hydrogel after photo-cross-linking (Figure S6 and S10). LAP concentration was chosen to have a high cross-linking rate without compromising cell viability (Figure S31 and S32).** For the bioink preparation, cyanobacteria suspension from the early exponential phase was centrifuged at 3,500 rcf for 5 mins, the supernatant was discarded, and the cell pellet was resuspended in the prepared bioink mixture (liquefied on ice) via pipetting to reach a homogeneous solution with an equivalent optical density $OD_{730nm} = 0.8$, **as this was the value measured in our liquid culture at the start of the exponential growth phase.** For volumetric printing, an equivalent optical density $OD_{730nm} = 0.3$ was used to allow sufficient light transmittance for the fabrication. **Choice of initial OD did not impact the cell growth (Figure S33).** The equivalent OD_{730nm} calculation is described in **Equation S1**.*

2. PCC 7002 embedding – Little details are reported regarding the experimental procedures adopted to embed cyanobacteria in the hydrogel matrix. To ensure research reproducibility, please carefully describe the embedding procedure in the corresponding materials and methods section.

We thank the Reviewer for the feedback regarding the clarity of our experimental procedures and we apologize that this was not sufficiently clear in the initial version of the manuscript. We included additional information about the mixing of cyanobacteria and the hydrogel precursor solution. In brief, the bioink was prepared with the cyanobacteria prior to printing by liquefying the material on ice and mixing with a concentrated solution of bacteria.

Text in the original manuscript (page 15):

*For the bioink preparation, cyanobacteria suspension was centrifuged at 3500 rcf for 5 mins, the supernatant was discarded, and the cell pellet was resuspended in the prepared bioink mixture to reach an equivalent optical density $OD_{730nm} = 0.8$. For volumetric printing, an equivalent optical density $OD_{730nm} = 0.3$ was used. The equivalent OD_{730nm} calculation is described in **Equation S1**.*

Updated text in the revised version of the manuscript (page 16):

*For the bioink preparation, cyanobacteria suspension from the early exponential phase was centrifuged at 3,500 rcf for 5 min, the supernatant was discarded, and the cell pellet was resuspended in the prepared bioink mixture (liquefied on ice) via pipetting to reach a homogeneous solution with an equivalent optical density $OD_{730nm} = 0.8$, as this was the value measured in our liquid culture at the start of the exponential growth phase. For volumetric printing, an equivalent optical density $OD_{730nm} = 0.3$ was used to allow sufficient light transmittance for high resolution fabrication. The choice of initial OD does not impact the cell growth (**Figure S33**). The equivalent OD_{730nm} calculation is described in **Equation S1**.*

3. PCC 7002 encapsulation efficiency and retention overtime – The presented F127 formulation is characterized by a high percentage of pristine F127 (i.e., 13.2 wt%). After crosslinking, this hybrid hydrogel formulation will be composed by a highly stable chemically crosslinked network (F127-BUM) mixed with a physical network constituted by F127. Because of the absence of chemical bonds, it is expected that the physical F127 phase will progressively dissolve overtime and leak out from the crosslinked network (or removed right after as reported for volumetric printing). As such, besides an initial loss of cyanobacteria (absorbed to the hydrogel surface only and not embedded in the inner structure), it is expected that there will be a progressive leakage of PCC 7002 embedded in the F127 phase.

a. What is the system PCC 7002 encapsulation efficiency overtime?

We thank the reviewer for raising this interesting point and we have now evaluated the encapsulation efficiency over time. Our system is not closed and cell leakage can occur. We calculated the ratio between cell count within the hydrogel and surrounding media and added it to the Supplementary Information (**Figure S13**). The encapsulation efficiency was above 90% over the first 25 days and decreased to 80% by day 30. The efficiency would potentially be higher in the samples that are only partially immersed in the medium.

Updated text in Results section (page 7):

*“The pH did not change in abiotic control samples throughout the incubation (**Figure S11b**) the majority of the encapsulated bacteria remained within the hydrogel throughout the incubation in biotic samples (**Figure S13**).”*

Text added to the updated Methods section (page 17):

Biomass quantification and cell viability characterization

"<...>

Chlorophyll extraction was done on randomly chosen disc samples ($n = 5$) every 5 days starting from day 10. The samples were fully immersed in 0.5–1 mL of isopropyl alcohol and homogenized (IKA, T25 Ultra Turrax with S25N-8G-ST inset). The slurry was then sonicated for 85 min followed by centrifugation at 19000 rcf for 10 min. Technical triplicates of 100 μ L supernatant were transferred into a 96-well plate and the absorption at 663 nm was measured by UV-vis plate reader (HIDEX Sense). Suspension samples of different optical densities were used to calculate an equivalent cell concentration from the chlorophyll amount using exponential fitting. (Figure S25). A non-linear fitting was chosen as we observed an increase in encapsulated cell concentration with chlorophyll extraction due to the interference of other cyanobacterial pigments, such as phycobiliprotein⁶⁸⁻⁷⁰.

For suspension and medium samples from the well plate, samples were collected every 5 days and the cell number was measured on a Multisizer 4e (Beckman Coulter) equipped with a 30 μ m aperture tube (gain: 4, current: 600 μ A). For each solution, 25 μ L of media were diluted in 10 mL of 0.2- μ m-filtered Isoton II solution within a 25 mL Accuvette (Beckman Coulter).

PCC 7002 encapsulation efficiency was quantified using the cell count and chlorophyll extraction results. The encapsulation efficiency was calculated as the percentage of encapsulated cell number to the total number of encapsulated cells and cells in the culture medium. Leakage was defined as the percentage of cells found in the culture medium to the total cell number of encapsulated cells and cells in the culture medium."

New references added:

68. Lauceri, R., Bresciani, M., Lami, A., & Morabito, G. (2018). Chlorophyll a interference in phycocyanin and allophycocyanin spectrophotometric quantification. *Journal of limnology*, 77(1).
69. Sobiechowska-Sasim, M., Stoń-Egiert, J., & Kosakowska, A. (2014). Quantitative analysis of extracted phycobilin pigments in cyanobacteria—an assessment of spectrophotometric and spectrofluorometric methods. *Journal of Applied Phycology*, 26, 2065-2074.
70. Strieth, D., Stiefelmaier, J., Wrabl, B., Schwing, J., Schmeckebier, A., Di Nonno, S., ... & Ulber, R. (2020). A new strategy for a combined isolation of EPS and pigments from cyanobacteria. *Journal of Applied Phycology*, 32, 1729-1740.

b. Did the author evaluate cyanobacteria leakage into the surrounding media?

This should be evaluated as it can affect the platform reproducibility and overall performance.

We evaluated the bacteria leakage from the samples and found that less than 20% of the total cells in the system are present in the surrounding media, indicating that the majority of the cyanobacteria remain encapsulated in the gel volume (Figure S13). For the first 25 days, cyanobacteria leakage was less than 10%. Increase in bacteria leakage/growth on day 30 (18%) might be attributed to the space limitation within the gel as we also observed a slower increase

in the growth rate of encapsulated cells, indicating that the encapsulated cell number may reach a steady state. We addressed this point of the reviewer in the Results section.

Updated text in Results section (page 7):

“The pH did not change in abiotic control samples throughout the incubation (Figure S11b) and the majority of the encapsulated bacteria remained within the hydrogel throughout the incubation in biotic samples (Figure S13).”

Figure S13. Cell count in the printed disc sample quantified with chlorophyll extraction (green) and cell count in the surrounding culture medium (pink) measured via Coulter Counter. The right axis indicates the percentage of cells that leaked into the surrounding culture medium (2 mL) normalized to the total number of cells in 1 printed disc sample (gray), as well as the encapsulation efficiency calculated as 100% – leakage % (brown).

4. PCC 7002 proliferation and CO₂ sequestration (Fig 2) – The authors report increasing cellular growth overtime, as shown by the presence of individual cells and cluster within the hydrogel matrix. Moreover, an increasing CO₂ sequestration was observed over 30 days. Do the authors expect photosynthesis to peak?

This is an interesting point. Yes, we do assume that photosynthesis will likely peak at some point, though we anticipate the system reaches a pseudo-steady state (with cells dying and growing at similar rates) while the amount of insoluble precipitates increases. In long-term studies, beyond one year, we also observed that the system undergoes cycles of more or less rapid growth,

including periodic yellowing and growth to a robust green sample. We elaborate on these aspects in the Results that describe Figure 2 and the related Discussion.

Text in the original Results section (page 7):

“Over the first 15 days the cumulative amount of CO₂ sequestered per gram of living material was 3.3 μmol (0.15 mg) and reached a value of 7 μmol (0.31 mg) by the end of the 30-day experiment, an increase that reflects CO₂ sequestration corresponding to biomass accumulation (Figure 2d).”

Revised text (Page 7):

“Over the first 15 days the cumulative amount of CO₂ sequestered per gram of living material was 3.3 μmol (0.15 mg) and reached a value of 7 μmol (0.31 mg) by the end of the 30-day experiment, an increase that reflects CO₂ sequestration corresponding to biomass accumulation (Figure 2d). Beyond 30 days, we expect the biomass accumulation to reach a pseudo-steady state (with similar cell death and growth rates) and the component of carbon sequestration related to biomass accumulation to equilibrate within the hydrogel.”

Added sentence in the Discussion section (Page 14):

“With minimum requirements of sunlight and atmospheric CO₂, our photosynthetic living materials showed a consistent carbon sequestration efficiency throughout the entire incubation period, and demonstrated the potential of using natural carbon sequestration processes at scale⁶⁰. We expect the system to reach its peak photosynthesis rate early as the cells go through an initial exponential growth phase and later operate at a pseudo-steady state while maintaining high viability (Figure S29).”

5. Dual carbon sequestration in photosynthetic living materials (Figure 2, d; Figure 2, e) –

a. It is not clear the number of biological samples and technical replicates that were used to perform the assay, as standard deviations are missing in both graphs. This info should be included and clarified in the corresponding materials & methods section, as well as in each caption (the authors mention 5 samples in the biomass and inorganic precipitation experimental section, but it is not clear whether technical or biological).

In **Figure 2d**, the pH was measured by pooling media from 20 randomly selected samples and the graph shows results of CO₂ calculated from the pH values obtained. The experiment was performed twice. We have updated the Text in the Methods accordingly (page 17):

“The collected medium pooled from individual samples (n = 20) was used for pH measurements to evaluate total sequestered CO₂ as fully described in the Supplementary Information (CO₂ sequestration by biomass growth (pH change) in living materials).”

In **Figure 2e**, all raw data of biological replicates (n = 5) and calculations are provided in the Supplementary Information. The results are presented as a difference of means with the standard error of the difference of means. We have updated the Text in the Methods accordingly (page 17):

“On day 5, 10, 20 and 30, abiotic and biotic samples ($n=5$; biological replicates) were randomly selected and washed with the same procedure. The samples were subsequently dried under the same condition before weighing. The mass was normalized to the average mass of the abiotic or biotic discs on day 0.”

The updated **Figure 2d,e** and the updated figure caption are appended below:

Figure 2. d) An increase in cell concentration was observed throughout the 30-day incubation in the biotic samples, with an initial rapid growth rate up to day 15 followed by a slower growth rate from day 15 to day 30 (green). Biotic samples continuously sequestered CO₂ over the 30-day incubation in the form of biomass accumulation (~0.31 mg CO₂ per g of gel, black). **e)** Both biomass growth and CaCO₃ precipitation contributed to the increase in dry mass of the biotic samples. The cumulative CO₂ sequestration by biotic samples was reflected by the difference in normalized dry mass between biotic and abiotic samples (black). Over the incubation period of 30 days, biotic samples accumulated 36% more dry mass than the abiotic samples. Alizarin red S staining on the biotic samples showed a total of 4.8 mg CaCO₃ precipitation per g of gel (red).

b. Biotic positive control groups (e.g., PCC 7002 cells in suspension) should also be added in addition to the negative abiotic ones, to evaluate the effect of the hydrogel matrix on cell viability and photosynthetic ability.

To address this comment we added the results comparing cell growth in hydrogel matrix and suspension positive control (cell count) to the Supplementary Information (**Figure S29**). It shows an increase in cell number for both encapsulated samples and liquid positive control during the first 15 days of incubation. Due to the cell growth in clusters in the living material, cell concentration was much higher compared to the liquid suspension positive control in terms of both equivalent cell number and equivalent OD (**Figure S29b** and **c**, respectively).

As the cells grow in clusters starting from day 5, chlorophyll extraction was conducted to quantify the encapsulated cell number. It is important to note that the available medium volume was different for our encapsulated samples and suspension. Since each disc sample was 40 μL with concentrated cell clusters grown in 2 mL of medium, we observed the cell number within an

encapsulated state is higher than that in the liquid positive control. These data demonstrate that there was no inherent limitation to cell viability and photosynthetic activity upon encapsulation.

Figure S29. a) Calibration curve fitting to relate chlorophyll absorption at OD_{663nm} to cell suspension OD_{730nm} (black) and calibration curve fitting to relate cell suspension OD_{730nm} and cell number measured via particle counting (black). **b)** Cell concentration in liquid suspension (particle counting, black) compared to average cell concentration in encapsulated samples obtained via confocal microscopy max projection (light green) and chlorophyll extraction (dark green). **c)** Comparison of equivalent OD_{730nm} of encapsulated cells (green) and actual OD_{730nm} of liquid suspension (black) from day 10.

c. Additional quantitative cell viability evaluation overtime as well as quantifying Alizarin red S staining would complement the qualitative information reported in Figure 2 b and c.

We thank the Reviewer for the comment regarding quantification of cell viability and Alizarin Red S Staining. We calculated cell viability using chlorophyll extraction as well as confocal imaging of dead cells in the material at different time points in the Supplementary Information (**Figure S34**). After reaching a steady state, the cells were >98% viable.

The results of quantitative analysis of Alizarin Red S staining were added to the updated **Figure 2e** and mentioned in the revised Results section. By the end of the 30 day experiment, the extent of precipitation measured by this method was 4.8 ± 2 mg of equivalent CaCO₃ per mL of gel,

which was higher than the total amount of precipitates (2.2 ± 0.9 mg per mL of gel) previously found by thermal decomposition.

Figure S34. Cell viability (blue) and comparison of live (green) and dead (black) cell concentration in encapsulated disc samples obtained via confocal microscopy max projection (day 0, day 1, and all dead cell data) or chlorophyll extraction (live cells day 15–30).

The updated **Figure 2d, e** with a second axis displaying the CaCO_3 equivalents as quantified by Alizarin Red S staining:

Figure 2. d) An increase in cell concentration was observed throughout the 30-day incubation in the biotic samples, with an initial rapid growth rate up to day 15 followed by a slower growth rate from day 15 to day 30 (green). Biotic samples continuously sequestered CO_2 over the 30-day incubation in the form of biomass accumulation (~ 0.31 mg CO_2 per g of gel, black). **e)** Both biomass growth and CaCO_3 precipitation contributed to the increase in dry mass of the biotic samples. The cumulative CO_2 sequestration by biotic samples was reflected by the difference in normalized dry mass between biotic and abiotic samples (black). Over the incubation period of 30 days, biotic samples accumulated 36% more dry mass than the abiotic samples. Alizarin red

S staining on the biotic samples showed a total of 4.8 mg CaCO₃ precipitation per g of gel (red).

Sentence added to the revised version of the Results section of the manuscript (page 8):

“Biotic samples stained red by day 10, indicating Ca²⁺ accumulation in the whole volume of the hydrogel and the dark red color was observed around growing bacteria clusters over time. From the analysis of Alizarin Red staining, we observed that the amount of precipitates was equivalent to > 1 mg of CaCO₃ per mL of hydrogel already after 10 days (Figure 2e). It reached 4.8 ± 2 mg of CaCO₃ per mL of hydrogel by the end of the 30 day period. Precipitates were also observed in the bottom of the incubation wells, and all analyses were carried out in fresh wells to avoid any influence on precipitation outside of the living material on our quantification (Figure S15).”

Updated Methods section about Alizarin Red S staining quantification (page 18):

“Calcium carbonate precipitate deposits in circular disc-shaped hydrogels were stained using 40 mM Alizarin Red S solution (pH = 4.2) in Milli-Q water. Each sample was fully immersed in 2 mL of Alizarin red S solution for 20 s. The sample was then washed in 200 mL of Milli-Q water 3 times for 5 min and imaged using an optical microscope. To quantify the extent of precipitation we extracted the Alizarin Red S stain from the samples using 10% acetic acid (0.5 mL per sample, n = 3) using vortexing for at least 1 hour. The solutions were then neutralized using 1 M NaOH and 100 µL of the neutralized extraction solution was used to measure the absorbance at 405 nm using a UV-vis plate reader (HIDEX Sense). Pure CaCO₃ calibration curve was prepared using the same method to calculate an equivalent amount (Figure S15a). Alizarin Red S staining was also performed on randomly chosen recovered wells (n = 3) of the 24-well plate every 5 days. 0.5 mL of 40 mM Alizarin Red S solution was added to each well for 20 s. Each well was then washed with 0.5 mL Milli-Q water 3 times to collect the precipitates which were then recovered by centrifugation at 12,300 rcf for 10 min. The supernatant was then discarded and the remaining precipitate pellet was dissolved in 0.5 mL of 10% acetic acid by pipetting. The solution was neutralized and the absorbance was measured under the same conditions as those of the disc-shaped hydrogel samples.”

d. Statistical analysis should also be included as it would be beneficial to validate the system advantages over control groups, and strength the authors' findings.

For most of our measurements we used abiotic samples as a control and we now included the results of statistical analysis where applicable for the comparison between biotic samples and abiotic control.

We performed two sample t-tests to find time points at which sample properties change significantly compared to day 0, 1, or 2 depending on the experiment (Figure 3e,f). We observed a significant increase (p = 0.001) in biomass from day 10 of incubation. A significant change in mechanical properties compared with day 1 was observed after 10 days (p = 0.05, Figure S21). No significant change was observed in the mechanical properties of abiotic samples throughout the time course.

Coefficient of determination was also determined for calibration curves: Alizarin Red S quantification of calcium precipitates (**Figure S15**), calibration curve fitting to relate chlorophyll absorption at OD_{663nm} to cell suspension OD_{730nm} and calibration curve fitting to relate cell suspension OD_{730nm} and cell number measured via particle counting (**Figure S29**) as well as the linear regression of direct CO₂ measurement within a closed container (**Figure S30**).

Specific to **Figure 2e**, we have updated the Text in the Results section (page 8):

Over the incubation period of 30 days, the biotic samples had approximately 36% more dry mass than control abiotic samples (**Figure 2e**) with significant increase ($p = 0.001$) in biomass observed from day 10 of incubation.

Figure 2. e) Both biomass growth and CaCO₃ precipitation contributed to the increase in dry mass of the biotic samples. The cumulative CO₂ sequestration by biotic samples was reflected by the difference in normalized dry mass between biotic and abiotic samples (black). Over the incubation period of 30 days, biotic samples accumulated 36% more dry mass than the abiotic samples. Alizarin red S staining on the biotic samples showed a total of 4.8 mg CaCO₃ precipitation per g of gel (red).

Updated relevant figures:

Figure 3e,f. Statistical analysis added to the graphs.

Figure S21. The storage modulus (G'), measured by shear rheometry, of biotic (green) and abiotic (gray) samples during the 30-day incubation period. A significant change in mechanical properties compared with day 1 was also observed after 10 days ($p = 0.05$).

Figure S15. a) Calibration curve to calculate equivalent CaCO_3 amount from the absorbance of Alizarin Red S staining. b) Equivalent CaCO_3 amounts calculated as difference in Alizarin Red S staining of the precipitates in the biotic and abiotic wells normalized per milliliter of hydrogel. The printed disc samples were transferred into new well plates every 5 days and Alizarin Red S quantification was done on the recovered well plates.

Figure S29. a) Calibration curve fitting to relate chlorophyll absorption at OD_{663nm} to cell suspension OD_{730nm} (black) and calibration curve fitting to relate cell suspension OD_{730nm} and cell number measured via particle counting (black). **b)** Cell concentration in liquid suspension (particle counting, black) compared to average cell concentration in encapsulated samples obtained via confocal microscopy max projection (light green) and chlorophyll extraction (dark green). **c)** Comparison of equivalent OD_{730nm} of encapsulated cells (green) and actual OD_{730nm} of liquid suspension (black) from day 10.

Figure S30. a) Rate of change of CO₂ concentration in a 250 mL closed container over a measurement period of 10 days with a 0.25 mL sample and 10 mL culture

medium. CO₂ sequestration was observed from day 3 and plateaued beyond day 5.
b) Representative snapshot of CO₂ evolution (black) in a closed container and linear interpolation of the measurement (blue).

6. Confirmation of mineral precipitates (Figure 3a; 3b). The XRD results are not convincing; if calcite were present, sharp peaks would be present that match the fingerprint of calcite whether or not magnesium or other trace elements were incorporated into the CaCO₃ framework. If magnesium were incorporated into the CaCO₃ network, only marginal broadening would be evident in the principal calcite peaks. The other peaks may be MgCO₃ or other minerals. The authors should label all peaks in the XRD regardless whether they are attributable to calcite or another mineral precipitate.

My suspicion is that there are other crystallized salts (e.g., CaCl₂, MgCl₂, CaSO₄) that are present after the heat treatment the authors conduct. The SEM images (Fig 3a) also give some concern that these are salts not calcite precipitates, as the morphology is not consistent with calcite. Furthermore, the elemental mapping does not show that carbon is collocated with the others, further indicating that these may, in fact, be crystallized salts. The additional peaks in the EDS spectra should be labeled to substantiate the authors' conclusions regarding the identification of mineral phases. The authors are encouraged to repeat the mineral precipitate characterization experiments using thermogravimetric analysis (TGA). TGA is a more appropriate method to confirm the presence (or absence) of CaCO₃, as it would indicate the onset of thermal degradation of CaCO₃, which can exist between ~450-700C.

We thank the Reviewer for the insightful suggestions on thermogravimetric analysis. We apologize that the characterization of the precipitates in our material was not convincing from the provided data and compiled additional information. To supplement the original analyses of the formed carbonates, we performed extensive additional experiments, including TGA on the mineral precipitates (Updated **Figure 3b**), FTIR analysis of the mineral precipitates (**Figure S17**), and further SEM–EDX analysis of the mineral precipitates (**Figure S20**). We have also included additional reference patterns in the XRD analysis (**Figure S18**). All of these additions corroborate our conclusion that the formed mineral precipitates are Ca–Mg carbonates and do not contain substantial amounts of other crystallized salts. We elaborate on each of these additions below and believe that this abundance of data supports the conclusions drawn in this work.

Thermogravimetric analysis was performed on the mineral precipitates under a full nitrogen environment. The decomposition temperature agrees well with the values reported for carbonates in literature and the range mentioned by the Reviewer, thus showing that we indeed have a Ca-carbonate phase. In addition, we performed FTIR analysis of the precipitates which show characteristic peaks (at 873 and 711 cm⁻¹) of carbonate and added it to the Supplementary Information¹.

Figure 3b. Thermogravimetric analysis (TGA) on the obtained mineral precipitates under a full nitrogen environment of the remaining precipitates, following removal of the polymer content and bacterial biomass by thermal decomposition, indicated the presence of carbonates in the biotic samples^{42,50}.

Figure S17. FTIR spectra of pure CaCO_3 (black) and the precipitates after one year of incubation (green) showing indicative peaks of carbonates at 873 and 711 cm^{-1} . For inorganic carbonates v_3 , represents asymmetrical stretching while v_2 and v_4 indicate in-plane and out-of-plane bending, respectively⁶.

In our work, we aimed to obtain crystalline precipitates that are less soluble than amorphous ones and provide more stable carbon storage than biomass alone. The XRD analysis of the precipitates formed confirmed that they were not amorphous. We rephrased the Results section to emphasize the crystallinity aspect and added a reference pattern of aragonite which has a similar peak pattern for the remaining CaCO_3 related peaks in our precipitate mixture. We added additional reference patterns of calcium carbonates with increasing Mg content for comparison of the peak position of (104) plane to XRD pattern and moved it to Supplementary Information **Figure S18**. Such a shift of the peak has been observed in biogenic precipitates¹.

Figure S18. XRD pattern of the precipitates after sample thermal decomposition at 600 °C and reference XRD pattern of aragonite, magnesite, dolomite and calcite⁷.

Further, we checked the presence of other elements at the precipitate site and could not observe an increased amount of chloride compared to the rest of the sample. We added SEM–EDS elemental mapping of our sample (for Na⁺ and Cl⁻) in the Supplementary Information (**Figure S20a**) and labeled additional chloride peaks EDX spectra (updated **Figure 3c**). We added the data analysis of EDX measurements confirming that Ca to Mg atom % ratios of (number of analyzed samples n = 10) was 78 ± 5 % to 22 ± 5 %, which indicates that both elements were embedded in the precipitates. We agree that the intensity of the carbon signal appears low in the same location as oxygen and calcium in the SEM-EDS elemental mapping. However, the carbon is visible in the EDX graph of **Figure 3c** of the area with the precipitates. As carbon is part of the polymeric matrix, we expected that the carbon signal will be everywhere as shown, also as a dominant peak of the abiotic matrix in **Figure 3c**. We added the full atom percentage distribution from two areas of the sample in **Figure S20b**, which shows that carbon and oxygen are dominant elements found in both areas. We changed the text to state that the areas of precipitates are enriched with oxygen and calcium signals in the results section.

Regarding the morphology of the precipitates in the SEM images, we now refer to them as ‘crystalline carbonate precipitates’ instead of ‘calcite’ as we can confirm crystallinity from XRD and carbonates from FTIR and TGA.

Figure S20. a) EDS elemental mapping of Ca, Mg, Na, O, C, and Cl atoms in the region with the precipitates of the biotic samples. Scale bar, 20 μm . b) Atom percentage distribution in polymer- and precipitate- dominant areas of the sample.

Figure 3c. EDX update with marked chloride peaks.

Figure 3. Carbonate formation within photosynthetic living materials.

Text in the Results section of the original manuscript (page 9):

“Having demonstrated that photosynthetic living materials are capable of CO_2 sequestration via both biomass accumulation and carbonate formation, we further investigated the composition of the mineral phase produced during incubation. MICP results in the formation of calcium carbonates that vary in stability and solubility with calcite being the most stable polymorph and, therefore, desirable for carbon sinking purposes (Figure 3a)⁴¹. XRD analysis of precipitates in biotic samples revealed a highly crystalline structure with the main peaks corresponding to calcite structure with a noticeable shift of the main diffraction peak representing the (104) plane (Figure 3b). The observed shift likely indicated compositional variation due to the incorporation of magnesium in the carbonate structure as previously observed in marine environments and, in this case, due to the use of simulated seawater (ASNIII medium)⁴². The XRD was obtained after thermal decomposition ($T = 600\text{ }^\circ\text{C}$, which is below the

decomposition temperature of CaCO_3 ⁴⁰) as the signal from the F127-based matrix was much stronger than the one of the precipitates (**Figure S9**).

Precipitates distributed throughout the polymeric matrix (**Figure 3c**). Energy dispersive X-Ray analysis (EDX) indicated mainly carbon and oxygen in the abiotic matrix whereas peaks of calcium and magnesium were prominent in the areas with precipitates in biotic samples. Elemental mapping corroborated these data with calcium, magnesium, carbon, and oxygen in overlapping regions near clusters of cyanobacteria, suggesting the pericellular formation of carbonates (**Figure 3d**).”

Text in the revised version of the Results section (page 10):

“Having demonstrated that photosynthetic living materials are capable of CO_2 sequestration via both biomass accumulation and carbonate formation, we further investigated the composition of the mineral phase produced during incubation. MICP results in the formation of **crystalline** carbonates that vary in stability and solubility with calcite being the most stable polymorph and, therefore, desirable for carbon sinking purposes (**Figure 3a**)⁴¹. **TGA analysis confirmed carbonate presence in the precipitates (Figure 3b) which was additionally confirmed by FTIR (Figure S17)**. XRD analysis of precipitates in biotic samples revealed a highly crystalline structure with the main peaks corresponding to **calcium–magnesium carbonates indicated by the position of the main** diffraction peak representing the (104) plane (**Figure S18**). The observed shift likely indicated compositional variation due to the incorporation of magnesium in the carbonate structure as previously observed in marine environments and, in this case, due to the use of simulated seawater (ASNIII medium)⁴². The XRD was obtained after thermal decomposition ($T = 600\text{ }^\circ\text{C}$, which is below the decomposition temperature of CaCO_3 ⁴⁰) as the signal from the F127-based matrix was much stronger than the one of the precipitates (**Figure S19**).

Precipitates distributed throughout the polymeric matrix (**Figure 3c**). Energy dispersive X-Ray analysis (EDX) indicated mainly carbon and oxygen in the abiotic matrix whereas peaks of calcium and magnesium were prominent in the areas with precipitates in biotic samples. Elemental mapping corroborated these data with calcium, magnesium, **and oxygen enrichment** in overlapping regions near clusters of cyanobacteria, suggesting the pericellular formation of carbonates (**Figure 3d and S20**). **EDX measurements also indicated that the Ca to Mg atom % ratio was $78 \pm 5\%$ to $22 \pm 5\%$ (number of analyzed samples $n = 10$), which confirmed that both elements were embedded in the precipitates.**”

Text added to the Methods section (pages 18–19):

“The presence of carbonates in the precipitates was confirmed by thermogravimetric analysis using a TGA/DSC 3+ (Mettler Toledo). The sample was placed inside a 70 μL alumina crucible and heated from 25 to 900 $^\circ\text{C}$ in pure N_2 under a constant flow rate of 150 mL min^{-1} . Buoyancy effects were accounted for by performing measurements with an empty crucible. The sample was first heated to 400 $^\circ\text{C}$ and held at this temperature for 30 min to remove any bound water, before continuing heating to 900

°C at a heating rate of 5 °C min⁻¹. The same measurement conducted with an empty crucible was taken as the baseline and was subtracted from the measured data.

The extent of carbonate precipitate formation within abiotic and biotic samples was further characterized by thermal decomposition at atmospheric pressure. 5 dry abiotic or biotic samples were loaded into a porcelain crucible and heated to 600 °C (ramp rate = 2 °C min⁻¹) for 2 h. The samples were then cooled to room temperature and the remaining mass was measured. In addition, the remaining precipitates were collected for attenuated total reflectance Fourier transform infrared spectroscopy (ATR-FTIR, Spectrum Two, PerkinElmer; 1 cm⁻¹ resolution, 32 scans) as well as X-ray diffraction (XRD) analysis with a PANalytical Empyrean diffractometer (Cu K α radiation, 45 kV and 40 mA) with an X'Celerator Scientific ultrafast line detector and Bragg-Brentano HD incident beam optics. XRD samples were measured over the 2 θ range 10–90° for 1 h with a step increment of 0.016°. Results were compared with reference patterns from the ICDD database.”

7. Hydrogel degradation and mechanical properties – Because of progressive dissolution of pristine F127, the authors observed a significant mass loss overtime (approx. 45% of the original mass as reported for the abiotic samples in Supplementary). This should translate in a significant reduction of mechanical properties for the abiotic samples, as well as an initial decrease for the biotic ones followed by progressive increase due reinforcement because of MICP deposition overtime.

a. How do the authors explain that no differences were observed for the abiotic samples between day 2 and 30, despite the remarkable mass loss?

While the formulation with pristine F127 is a viscoelastic solid, photo-cross-linking of the F127:F127-BUM hydrogel resulted in an increase of the storage modulus from 16 kPa to 48 kPa prior to swelling (**Figure 1e** and **S9**). For improved clarity, we have updated the axes in the updated **Figure 1**. Therefore, we concluded that the main contributor to the storage modulus in the samples tested (abiotic and biotic) was the chemically cross-linked F127-BUM network. Therefore, dissolution of the weak viscoelastic F127 that was not cross-linked would not be expected to make a substantial change in the final material properties. In addition, the reported mechanical properties were measured on day 2 and day 30, as we first allowed the samples to equilibrate in the culture medium prior to the initial mechanical measurement. Partial dissolution of non-cross-linked F127 from the hydrogel matrix could have already occurred prior to the measurement on day 2. Measurements on day 0 were not included as sufficient time is needed for the samples to equilibrate their swelling state for proper mechanical analysis. Given this, any potential dissolution of pristine F127 did not result in a significant drop in modulus between day 2 and day 30 samples.

b. Adding additional intermediate timepoints would better characterize when MICP deposition becomes relevant for material reinforcement.

We thank the reviewer for the comment. We performed rheological measurements and quantitative Alizarin Red S staining at intermediate time points to further assess when the MICP deposition became relevant for material reinforcement. We observed that precipitate accumulation was > 1 mg mL⁻¹ of hydrogel already after 10 days (see answer to R3.5c). A significant change in mechanical properties compared with day 1 was also observed after 10

days ($p = 0.05$, **Figure S21**). No significant change was observed in the mechanical properties of abiotic samples throughout the time course. We added an additional comment about the onset of the change in properties in the Results section.

Figure S21. The storage modulus (G'), measured by shear rheometry, of biotic (green) and abiotic (grey) samples during the 30-day incubation period.

Text in the Results section of the original manuscript (page 9):

“The formation of precipitates in hydrogel matrices via MICP can reinforce the mechanical properties of the material^{17,43,44}. The storage moduli (G') of the printed biotic samples on day 2 of the experiment were slightly lower compared with abiotic samples (5.4 ± 2 kPa and 6.7 ± 2 kPa, respectively).”

Revised text in the Results section of the manuscript (page 10):

*“The formation of precipitates in hydrogel matrices via MICP can reinforce the mechanical properties of the material^{17,43,44}. The storage moduli (G') of the printed biotic samples on day 2 of the experiment were slightly lower compared with abiotic samples (5.4 ± 2 kPa and 6.7 ± 2 kPa, respectively). **A significant increase ($p = 0.05$) in storage moduli (G') in the case of biotic samples was observed already after 10 days (Figure S21).”***

Text added in the Methods section (page 19):

“Viscoelastic properties of the printed hydrogels at different time points ($n = 3$; biological replicates) during incubation were evaluated using the same rheometer as for the bioink characterization equipped with a Peltier stage and 8 mm sandblasted probe (gap size $h = 0.15$ – 0.35 mm, normal force $F_N = 0.1$ N) at 25 °C.”

c. Statistical analyses should be included to confirm significant differences in mechanical performance, if any.

We performed a two-sample t-test and added the p values to the graphs of mechanical analysis of **Figure 3** where significant differences were found between samples. In case of rheology results, on day 2 as well as between day 2 and day 30 of abiotic samples there was no significant

difference $p > 0.05$. There were also no significant differences between samples in the toughness measurements $p > 0.05$. We added the methods of statistical analysis to the Methods section.

Figure 3e,f. Statistical analysis added to the graphs.

Text added in the Methods section (page 20):

“Statistical analysis of the samples at different time points was performed using two-sample t-tests (OriginPro 2019 software). All values are shown as average with standard deviation or root mean square of standard deviations in cumulative cases.”

Revised text in the Results section of the manuscript (page 10):

“Over the course of 30 days, the modulus increased significantly to 10.1 ± 1 kPa ($p = 0.01$) in the biotic case while it did not change for abiotic samples (6.5 ± 1 kPa) (Figure 3e).”

In case of tensile testing, we updated the number of biological replicates in the Methods section (page 19).

“To obtain homogeneous samples for tensile testing, abiotic and biotic bioink were molded into a rectangular-shaped mold (12 mm × 50 mm × 1 mm, width × length × thickness) and cross-linked for 2 min ($\lambda = 405$ nm light, $I = 8$ mW cm⁻²). Samples were then washed and incubated under the same conditions as the disc-shaped samples. Uniaxial tensile testing was performed on the samples ($n = 5$; biological replicates) incubated for 2 and 30 days using a tensile testing machine (Stable Micro Systems) equipped with two parallel metal screw-clamps.”

8. Volumetric printing of lattice structures – Volumetric 3D printing was adopted to achieve high resolution and a defined internal porosity. However, this cannot be appreciated from brightfield images reported in Fig 4,b.

a. SEM of the external surface and internal cross-sections of the final structure would validate that the desired high resolution and internal defined porosity was achieved.

We added images of the external surface (updated **Figure 4a**) as well as cross sections of lyophilized sample (**Figure S25**) prepared by volumetric printing to confirm the porosity of the sample to the Supplementary Information.

Figure S25. *a) Optical image of a lyophilized sample prepared by volumetric 3D printing. Scale bar, 1 cm. b) SEM image of the same sample with lattice structure falsed colored in blue. Scale bar, 1 mm. c) SEM image of the cross-section of the sample prepared by volumetric 3D printing. Scale bar, 1 mm.*

Updated **Figure 4a**:

Figure 4. Digital fabrication of photosynthetic living structures for dual carbon sequestration. *a) Lattice structures were digitally designed to facilitate light exposure and medium transport via passive capillary wetting, enabling growth of the photosynthetic living materials. The structures were fabricated using volumetric printing. The porosity of the printed structure can be observed in the lyophilized sample.*

b. False color high resolution SEM images of seeded samples should also be provided to better understand localization of cells within the hydrogel network.

We added false color high resolution SEM images of cells embedded in the polymer (**Figure S1a**) to the Supplementary Information. We also added lower magnification images of the same area for context (**Figure S1b–d**).

Figure S1. SEM images of the biotic sample at different magnifications. A representative cluster of embedded cyanobacteria *Synechococcus* sp. PCC 7002 are highlighted in green false color to indicate the location within the supporting matrix. It should be noted that the sample was homogeneously populated with embedded cyanobacteria that have not been false colored, as best seen in panel b). Scale bars, a) 1 μm , b) 10 μm , c) & d) 100 μm .

9. Long-term viability evaluation of 3D printed photosynthetic living materials – The authors report extended cells viability for over a year.

a. To better characterize cell proliferation within the lattice structure overtime, quantitative cell viability assays should be performed.

To assess cell viability, we performed dead cell staining at different time points and evaluated overall biomass in the sample by chlorophyll content (**Figure S34**). After reaching a steady state, the encapsulated cells were > 98% viable throughout the time course of analysis.

Figure S34. Cell viability (blue) and comparison of live (green) and dead (black) cell concentration in encapsulated disc samples obtained via confocal microscopy max projection (day 0, day 1, and all dead cell data) or chlorophyll extraction (live cells day 15–30).

Added sentence in the Discussion section (page 14):

“With minimum requirements of sunlight and atmospheric CO₂, our photosynthetic living materials showed a consistent carbon sequestration efficiency throughout the entire incubation period, and demonstrated the potential of using natural carbon sequestration processes at scale⁶⁰. We expect the system to reach its peak photosynthesis rate early as the cells go through an initial exponential growth phase and later operate at a pseudo-steady state while maintaining high viability (Figure S29).”

b. The authors indicate that transparency of the hydrogel facilitates light penetration and cell growth within the structure. However, because of progressive mineral deposition, the structure loses its transparency overtime. Do the authors expect the reduce light penetration to affect viability of the inner layers of the structure? Evaluation of cell distribution within the material thickness overtime should also be investigated by providing representative images of material cross sections at different timepoints (both brightfield images as well as fluorescently stained live and dead cells).

We agree that precipitates would decrease the transparency of the matrix. However, from our observations, cell encapsulation and further growth played a larger role in the decrease in transparency (Figure S3). We observed a 5 mm depth to be an optimal material thickness to maintain PCC 7002 viability (Figure S27). This motivated the design parameters for the textured surfaces shown in Figure 4d. We added a comment regarding light penetration to the Results of the revised manuscript as well as images of cell growth in living material 5 x 5 x 5 mm cube cross sections (Figure S24) with cell growth viability differences at the edges and in the center of the cube (Figure S24). The cells grow in big clusters at the edges while small clusters and individual cells are more prominent in the center. We assume that the lower cell growth in the deeper regions was due to light shielding from cell clusters in the outer regions of the cube.

Figure S24. a) Optical images of the cross section of a 5 x 5 x 5 mm printed cube. Scale bar, 5 mm. Cell viability of samples from **b)** edge and **c)** center of a cross section of a 5 x 5 x 5 mm printed cube (SYTOX Blue dead cell staining, cyan; cell autofluorescence, green). Scale bar, 100 μm .

Text added to the Results (page 12):

“The transport of gases and liquid medium as well as access to light are essential for the efficient functioning of biologic processes (including carbon sinking) in the photosynthetic living materials^{9,31}. The decrease in light penetration over the course of the life cycle is expected due to increased light absorption and scattering with increased biomass, resulting in reduced proliferation in the volume fraction further from the surface (Figure S24).”

RESPONSE TO REVIEWERS: All responses are written in blue font and all changes to the main text or supplementary information have been highlighted in yellow.

Reviewer #1 (Remarks to the Author):

The authors have performed several additional experiments. My comments are mostly related to the additional data:

1. Figure S11a: The BG11-ASNIII medium was changed every 5 days, so additional carbon source is made available (potentially) each time that the medium is refreshed. How is that taken into consideration?

We agree that this is an important point and we apologize that the consideration of added carbon was not explained clearly in the original manuscript. We included sodium carbonate (Na_2CO_3) in the medium to simulate seawater; however, the amount of Na_2CO_3 present in the medium was below the level of dissolved inorganic carbon (DIC) at equilibrium under our operating conditions. Therefore, additional carbon was not made available each time the medium was refreshed. When quantifying the DIC content after each 5-day period, we re-collected all the medium and analyzed the total DIC after the 5 days of culture. Using this methodology, there was no way for sequential accumulation of DIC from the medium itself and we were able to compare the DIC in the medium after each 5-day culture period in the basal medium. The amount of DIC in the re-collected culture medium was at or above the amount of inorganic carbon initially present from the Na_2CO_3 in the medium for each culture period (former **Figure S11a** now **Figure S1a**). This means that during each 5-day culture period, additional atmospheric carbon dissolved in the culture medium to maintain the equilibrium amount of DIC, as illustrated in **Figure 1a**. As the initially available inorganic carbon from Na_2CO_3 did not affect the DIC equilibrium, the change in medium did not make additional carbon available.

To further support that the DIC equilibrium under these conditions was at or above that of the initially supplemented carbon, we conducted additional DIC analyses on Na_2CO_3 -free BG11-ASNIII medium at pH = 7 and pH = 9 (**Figure S1c**). Again, the DIC content in the medium without Na_2CO_3 was higher than 381 $\mu\text{mol/L}$, which is the amount of Na_2CO_3 in the culture medium. This indicates that the DIC equilibrium is not dependent on the presence of Na_2CO_3 in the simulated seawater medium and that atmospheric carbon did dissolve in the culture medium under all conditions.

While no additional carbon is made available to the system due to the Na_2CO_3 in the medium, some of the carbon in the final construct comes from the inorganic carbon from the Na_2CO_3 in the culture medium, which was added to simulate the mineral content of seawater. In the previous revision, we had already softened all language about atmospheric CO_2 being the sole carbon source for carbon sequestration throughout the manuscript for this reason.

In addition, we clarified the caption for **Figure S1** to avoid potential confusion for the reader:

Figure S1. **a)** Total amount of dissolved inorganic carbon (DIC) under equilibrium in the re-collected abiotic (black) and biotic (green) medium after every 5-day culture period ($n = 2$, with pooling of 20 individual samples in each case). The gray dashed line indicates the amount of DIC in the form of Na_2CO_3 that was used in simulated seawater medium (BG11–ASNIII). **b)** pH changes of pooled biotic (green) and abiotic (black) medium every 5 days ($n = 3$). **c)** DIC analyses on BG11–ASNIII medium without the addition of inorganic carbon through sodium carbonate (Na_2CO_3) at pH = 6.7 and pH = 8.9 ($n = 2$, black). The gray dashed line indicates the amount of DIC in the form of Na_2CO_3 that was used in simulated seawater medium (BG11–ASNIII).

Figure 1. a) PCC 7002 are capable of CO₂ sequestration via biomass accumulation and microbially induced carbonate precipitation (MICP) during photosynthesis using dissolved CO₂ as the major carbon source.

Updated text in the Results section (page 7)

“40.4 mg L⁻¹ (0.38 mM) of Na₂CO₃ was dissolved in the medium to simulate natural seawater. Analysis of the total DIC under equilibrium in the culture medium indicated that the DIC content was above or at the quantity of carbon initially dissolved in the form of CO₃²⁻ (**Figure S1**). This increase in the DIC confirmed the dissolution of atmospheric CO₂, which served as the primary carbon source for CO₂ sequestration.”

2. Figure S11a: Why are the values for the DIC in the abiotic medium varying like this with a peak at 20 days? Please also include the number of replicates and error bars.

The observed variation in dissolved inorganic carbon (DIC) value at day 20 was likely due to cumulative variability in the measurements. The DIC in the medium was calculated based on several parameters, including the medium pH, medium conductivity, salinity, and alkalinity. Each measurement of these parameters incurred a certain degree of variability, and the error bar in the final plot for DIC reflects the combined uncertainty from all the factors. The abiotic samples exhibited increased variability starting from day 20—this variability was also observed in the pH data (**Figure 1b**). In contrast, the variability was less pronounced in the biotic samples, especially in the later stages of the experiments.

The DIC calculation consisted of two sets of experiments, and each data point was obtained by pooling the medium from 20 individual samples, which was required for more accurate titration results.

The updated former **Figure S11** now **Figure S1** and caption appended below:

Figure S1. a) Total amount of dissolved inorganic carbon (DIC) under equilibrium in the re-collected abiotic (black) and biotic (green) medium after every 5-day culture period ($n = 2$, with pooling of 20 individual samples in each case). The gray dashed line indicates the amount of DIC in the form of Na_2CO_3 that was used in simulated seawater medium (BG11-ASNIII). **b)** pH changes of pooled biotic (green) and abiotic (black) medium every 5 days ($n = 3$). **c)** DIC analyses on BG11-ASNIII medium without the addition of inorganic carbon through sodium carbonate (Na_2CO_3) at pH = 6.7 and pH = 8.9 ($n = 2$, black). The gray dashed line indicates the amount of DIC in the form of Na_2CO_3 that was used in simulated seawater medium (BG11-ASNIII).

3. The amount of CO_2 sequestered by precipitation is less than 0.1% of the whole mass of the materials. Based on Figure S16 (biotic culture) and Supplementary text on “ CO_2 sequestration by biomass growth (pH change) in living materials”, the precipitates’ weight is fairly low, so the dual carbon sequestration does not appear really efficient.

We have removed references to this process as being “efficient” throughout the manuscript and instead focused on reporting the quantitative amounts of CO_2 sequestered per unit mass of wet hydrogel. The main text has been adapted accordingly.

Text in the original version of the Abstract:

“Digital design and fabrication of the living material ensured sufficient access to light and nutrient transport of the encapsulated cyanobacteria, which were essential for long-term viability (more than one year) as well as efficient photosynthesis and carbon sequestration.”

Updated text in the revised version of the Abstract:

*“Digital design and fabrication of the living **materials** ensured sufficient access to light and nutrient transport of the encapsulated cyanobacteria, which were essential for long-term **culture** (more than one year), **photosynthesis**, and carbon sequestration. [...] These findings highlight the potential of photosynthetic living materials for scalable carbon sequestration **with possible future use in carbon-neutral infrastructure and green building materials.**”*

Text in the original version of the Results (Page 8):

“CO₂ sequestration via MICP was approximately one order of magnitude more than that achieved via biomass growth, making the dual CO₂ sequestration a more efficient carbon sink, and indicating that the majority of the sequestered CO₂ had been stored in a more stable mineral form.”

Updated text in the revised version of the Result (Page 8):

*“**CO₂ sequestered via MICP has been stored in a more stable mineral form.**”*

We did not report the amount of CO₂ sequestered by precipitation as a percentage, as we believe that presenting the mass of carbon sequestered per unit mass is a more meaningful and representative value—both for potential future applications of this approach and for comparison with other technologies. That said, the amount of CO₂ sequestered via carbonate precipitation was quantified as 2.2 ± 0.9 mg of CO₂ per gram of wet hydrogel material over a culture period of 30 days, increasing to 26 ± 7 mg over 400 days. This corresponds to a mass percentage of ~2.6% after 400 days and ~0.22% after 30 days, based on wet (whole) mass of the materials. In addition, we prefer to report these values in quantitative terms per unit mass of wet hydrogel rather than as percentages given the continuous nature of the sequestration in our living materials.

4. Concerning the carbon sequestration, is it possible to provide a quantitative analysis of the carbon sequestration contribution from each factor? (e.g., biomass accumulation, carbonate precipitation without the source of sodium bicarbonate in ASNIII-BG11).

We thank the Reviewer for the comment and we would first like to clarify the culture medium composition. We did not add sodium bicarbonate (NaHCO₃) to BG11–ASNIII, only sodium carbonate (Na₂CO₃), which was included in the culture medium to simulate the composition and ionic strength of seawater. As the samples and culture medium were exposed to the open atmosphere, a carbon equilibrium was established according to Henry’s Law (**Figure 1a**). As seen in **Figure S1a** of the updated manuscript, that initial dissolved inorganic carbon (DIC) concentration was double that of the carbon content from Na₂CO₃ alone. Therefore, the carbon integrated into biomass and insoluble carbonates is likely coming from both the atmosphere and the inorganic carbon included in the medium. In our current experimental setup, the amount of carbon (CO₂) sequestration via carbonate precipitation was 2.2 ± 0.9 mg of CO₂ per gram of wet material after 30 days. We also observed an increase in cell number from day 0 to day 30, from an initial cell concentration of 3 × 10⁷ cells mL_{gel}⁻¹ to a final cell concentration of 5 × 10⁹

cells mL_{gel}⁻¹. While biomass increased during the same culture period, we were not able to provide a quantitative measure of the CO₂ sequestered into this biomass. Approximations based on cell counts, chlorophyll extraction, and medium pH indicated that the CO₂ sequestration by biomass accumulation was likely to be a minor fraction of the total CO₂ sequestered by our photosynthetic living materials (approximately one order of magnitude less than by carbonate precipitation). However, as the CO₂ sequestered into this biomass were approximated based on the cell counts, chlorophyll extraction and medium pH, we only report the quantitative measure of CO₂ sequestration via carbonate precipitation in the revised form of the manuscript.

In the future, we agree that more quantitative measures of the relative contributions to the carbon sequestered should be conducted. Such experiments could be carried out in closed systems, and preferably large-scale experiments. These experiments should control carbon input into the system and more accurately quantify the rate of photosynthesis via the direct measurement of O₂ evolution. In this manner, one may be able to quantitatively assess the relative contributions of biomass accumulation and carbonate precipitation as well as to provide a more precise number of the amount of atmospheric carbon sequestered per unit mass of material and per unit time. We have revised the Discussion to indicate the importance of such future studies in the field.

Revised text in the Discussion (pages 14 and 15):

[...] Future work should include a more quantitative evaluation of CO₂ sequestration through biomass accumulation to better understand the photosynthetic rate of the cyanobacteria. Direct measurements of O₂ evolution using an oxygen electrode would provide a more accurate quantification of photosynthesis. Additionally, established computational models could be applied to further explore the relationship between cytosolic pH of the cyanobacteria and CO₂ assimilation into biomass⁶⁵.

“Upscaling would provide the means to perform more robust CO₂ sequestration analyses using gas composition monitoring (Figure S30) as well as control and tracking of carbon sources via media and atmospheric supply.”

5. Figure S30: The CO₂ consumption rate data is quite noisy, and a very low R² value (0.15) is shown on panel b. Also, in panel b, is there a reason why the x-axis is from 0.5 to 1.4h? Please also include the number of replicates and error bars.

The measurement was conducted by measuring the CO₂ concentration within a closed container over 10 days. Upon closing the container, the setup was allowed to equilibrate for approximately 30 min, and therefore, the x-axis starts from 0.5 h. Each data point was obtained every 4–5 s (Figure S30b). The low R² value was due to the high sampling frequency of the CO₂ concentration within the closed container, as any external fluctuation (in temperature, vibration, or movement) caused the value to vary.

We added error bars in Figure S30a for the 10-day measurement (n = 2). We have also updated the Methods section accordingly to better explain how these data were acquired.

Updated Figure S30:

Figure S30. a) Rate of change of CO₂ concentration in a 250 mL closed container over a measurement period of 10 days with a 0.25 mL sample and 10 mL culture medium ($n = 2$). CO₂ sequestration was observed from day 3 and plateaued beyond day 5. **b)** Representative snapshot of CO₂ evolution (black) in a closed container and exemplary linear interpolation of the measurement (blue) used to calculate the rate of change of CO₂ concentration.

Method appended in the Supplementary Information (page 10):

“Direct CO₂ measurement in a closed environment

A closed CO₂ measurement system was built with a 250 mL airtight container, a CO₂ sensor (Sensirion I2C SCD4x Multiple Function Sensor), and an Arduino microcontroller board (Uno r4 minima). An SD card reader was connected to the Arduino board to record the data. Two 8.2 kΩ resistors were added into the electric circle of the sensor to amplify the voltage signal of the two data cables. A 0.25 mL sample with approximately 10 mL of culture medium was placed in each measurement container. The containers were opened every day and were allowed to equilibrate with atmospheric air to replenish the CO₂ within the closed system. The CO₂ concentrations were recorded on the SD card every 4–5 s.”

Minor points:

- The authors use expressions like carbon-negative and carbon-neutral infrastructure in the manuscript. To use such wording, it would require a quantitative cradle-to-grave CO₂ emission analysis (LCA).

We thank the reviewer for the comment and we agree that a quantitative life cycle analysis would be needed for these materials. We have softened our statement and included a comment related to the need for life cycle assessment in the revised Discussion.

Original text in the manuscript (page 15):

“Photosynthetic living materials also hold promise for future applications as surface coatings for green building materials or bioreactors in commercial-scale sequestration plants, enabling bioremediation of CO₂ emissions and supporting the realization of carbon-neutral to carbon negative infrastructure.”

Updated text in the manuscript (page 15):

“Photosynthetic living materials also hold promise for future applications as surface coatings for green building materials or bioreactors in commercial-scale sequestration plants, enabling bioremediation of CO₂ emissions and could potentially support the reduction of CO₂ emission in the built environment. However, a comprehensive life cycle assessment would be necessary to quantify the net carbon impact of such systems.”

- Caption of Figure S11a: Shouldn't that be BG11-ASNIII medium?

We thank the reviewer for pointing this out and we have corrected this accordingly.

- Typo in Figure S11: mediuim (twice)

We thank the reviewer for noticing the typo and these have been corrected.

- Typo in Figure 2a: medum

We thank the reviewer for noticing the typo and the word has been corrected.

Reviewer #2 (Remarks to the Author):

After reading the authors' Response to Reviewers, I appreciate the addition of cell counts and chlorophyll measurements to support the conclusions regarding biomass growth. However, I remain unconvinced that measuring extracellular pH is a valid way to quantify uptake of inorganic carbon into biomass in this system, which is open to the atmosphere. The standard way to quantify photosynthesis in cyanobacteria is to measure O₂ evolution using an oxygen electrode or by membrane inlet mass spectrometry.

I understand that there are several references (cited in the Response to Reviewers) that describe the release of OH⁻ ions by carbonic anhydrase located in the carboxysome as part of a model for the cyanobacterial CCM and MICP, but this is an oversimplification that is not biochemically correct. A paper on quantitative modeling of the CCM in beta-cyanobacteria (Mangan et al., 2016. PNAS 113: E5354-E5362) notes that the combined action of CA and Rubisco in the carboxysome actually produces a net H⁺, not an OH⁻ ion.

The statements on p. 4 ("Here, carbonic anhydrase (CA) within the carboxysome converts the bicarbonate species into CO₂, releasing a hydroxyl ion (OH⁻) into the surrounding media^{7,33–35}. The OH⁻ is excreted to the exterior of the bacterium....") and p. 7 ("Using HCO₃⁻ as a substrate, carbonic anhydrase generates equimolar amounts of OH⁻ ions, which are excreted into the surrounding medium, and CO₂ molecules, which are integrated into bacterial metabolism.^{17,27}") and the depiction in Fig. 1 are incorrect.

We thank the Reviewer for the additional comments on this point and we agree that complementing the initial data in the original version of the manuscript with cell counts and chlorophyll measurements were important and helpful additions to corroborate the conclusions related to biomass growth. In order to provide an estimate of the amount of CO₂ sequestered via biomass, we had initially used the increase in pH and these data were in agreement with the cell counts and chlorophyll measurements. However, to avoid any incorrect statements and to focus on the central conclusions of the manuscript, we have removed the pH data from the main text and no longer use pH changes as a metric to quantify biomass accumulation. In addition, we have softened the statements on this topic pointed out by the Reviewer to avoid confusion or incorrect statements related to these points. Further, we would like to re-emphasize that while biomass accumulation is a necessary feature of our system, our approximate values indicated that it was only a minor fraction of the CO₂ sequestered by these photosynthetic living materials.

We have now rephrased the text on both page 4 and page 7 to avoid any misstatements as pointed out by the Reviewer:

Original text on page 4:

"We engineered photosynthetic living materials for dual carbon sequestration by exploiting the carbon concentrating mechanism of cyanobacteria strain PCC 7002 (Figure 1a). Dual CO₂ sequestration proceeds via both reversible biomass accumulation and irreversible mineral precipitation. For carbon concentration to occur, CO₂ from the surrounding environment dissolves in aqueous solutions and forms bicarbonate ions (HCO₃⁻) that are then transported to the marine β-cyanobacteria 7002 carboxysome³³. Here, carbonic anhydrase (CA) within the carboxysome converts the bicarbonate species into CO₂, releasing a hydroxyl ion (OH⁻) into the surrounding media^{7,33–35}. The OH⁻ is excreted to the exterior of the bacterium and the

CO₂ is fixed by ribulose-1,5-bisphosphate carboxylase/oxygenase (RuBisCo) during photosynthesis into two molecules of 3-phosphoglycerate, which is enzymatically converted into sugars that support cell growth and biomass production. Due to the metabolic production of OH⁻, the local pH value of the local medium surrounding the bacteria increases. This increase in pH, combined with negatively charged extracellular polysaccharides on the bacterial membrane, creates a favorable environment for the nucleation and formation of insoluble carbonates³¹. In the presence of divalent cations, such as Ca²⁺ and Mg²⁺, CO₃²⁻ in the medium is consumed and fixed irreversibly into calcium or magnesium carbonates. Chemical equilibrium then favors the dissolution of additional atmospheric CO₂ into the culture medium, continuously driving the dual carbon sink.”

Revised text (Page 4):

*“We engineered photosynthetic living materials for dual carbon sequestration by exploiting the carbon concentrating mechanism of cyanobacteria strain PCC 7002 (**Figure 1a**). Dual CO₂ sequestration proceeds via both reversible biomass accumulation and irreversible mineral precipitation via MICP. For carbon concentration to occur, CO₂ from the surrounding environment dissolves in aqueous solutions and forms bicarbonate ions (HCO₃⁻) that are then transported to the marine β-cyanobacteria 7002 carboxysome³³ Within the carboxysome, carbonic anhydrase (CA) catalyzes the conversion of bicarbonate into CO₂, which is then fixed by ribulose-1,5-bisphosphate carboxylase/oxygenase (RuBisCo) during photosynthesis into two molecules of 3-phosphoglycerate. This product is enzymatically converted into sugars that support cell growth and biomass production, representing the reversible part of CO₂ sequestration.*

*Irreversible CO₂ sequestration as mineral precipitation occurs outside the cells, near the cyanobacteria. Negatively charged extracellular polysaccharides on the bacterial membrane combined with a suitable extracellular environment (alkaline pH, presence of divalent cations), facilitates the nucleation and formation of insoluble carbonates^{33–37}. Thus, by culturing photosynthetic living materials in simulated seawater that contains Ca²⁺ and Mg²⁺, an increase in medium pH, which occurred in cyanobacteria-containing samples (**Figure S1**), allows CO₃²⁻ in the medium to be fixed irreversibly into calcium or magnesium carbonates. Chemical equilibrium then favors the dissolution of additional atmospheric CO₂ into the culture medium, continuously driving the dual carbon sink.”*

Original text on page 7:

*“PCC 7002 growth within the living material was observed directly by microscopy and indirectly by measured pH changes in the surrounding medium (**Figure 2b**). After 10 days of incubation samples exhibited visible clusters of PCC 7002 cells as well as many individual cells throughout the volume of the hydrogel matrix. The number of clusters and individual cells increased with time (**Figure S12**). The change in pH of the culture medium was attributed to carbonic anhydrase activity in the carboxysome of the bacterial cells. Using HCO₃⁻ as a substrate, carbonic anhydrase generates equimolar amounts of OH⁻ ions, which are excreted into the surrounding medium, and CO₂ molecules, which are integrated into bacterial metabolism^{17,27}. As the medium was changed every 5 days, the cumulative amount of OH⁻ ions was reflected in the pH increase of the medium from the initial pH value of 6.5. The cumulative OH⁻ ion production served as a surrogate measure of carbonic anhydrase activity and CO₂ integration into biomass⁴⁰. Between days 5 and 10, the pH of the biotic samples*

increased to 9.5, corresponding to approximately three orders of magnitude increase in OH⁻ concentration. In each subsequent 5-day interval, the pH increased by 1.8–2.9 with an average of more than two orders of magnitude increase in OH⁻ concentration (**Figure S11b**). The cumulative CO₂ sequestered was calculated based on the pH change of the medium (calculations in the Supplementary Information, **CO₂ sequestration by biomass growth (pH change) in living materials**). Over the first 15 days the cumulative amount of CO₂ sequestered per gram of living material was 3.3 μmol (0.15 mg) and reached a value of 7 μmol (0.31 mg) by the end of the 30-day experiment, an increase that reflects CO₂ sequestration corresponding to biomass accumulation (**Figure 2d**). Beyond 30 days, we expect the biomass accumulation to reach a pseudo-steady state (with similar cell death and growth rates) and the component of carbon sequestration related to biomass accumulation to equilibrate within the hydrogel. The pH did not change in abiotic control samples throughout the incubation (**Figure S11b**) and the majority of the encapsulated bacteria remained within the hydrogel throughout the incubation in biotic samples (**Figure S13**).”

Revised text (Page 7):

“PCC 7002 biomass growth within the living material for CO₂ capture was evaluated by fluorescence microscopy (cell counts) and chlorophyll extraction (**Figure 2b**). After 10 days of incubation samples exhibited visible clusters of PCC 7002 cells as well as many individual cells throughout the volume of the hydrogel matrix. The number of clusters and individual cells increased over time with cell concentrations reaching 5×10^9 cells mL_{gel}⁻¹ after 30 days (**Figure 2d, S12**). Further, the choice of initial OD_{730nm}, within the range of 0.3–1.6, did not impact cell growth (**Figure S13a**). Moreover, the majority of the encapsulated bacteria remained within the hydrogel throughout the incubation in biotic samples (**Figure S13b**). Beyond 30 days, we expect the biomass accumulation to reach a pseudo-steady state (with similar cell death and growth rates) based on the plateau observed in the cell concentration beyond day 25.”

In addition, we have removed the arrow indicating the secretion of OH⁻ ions by the cyanobacteria into the extracellular environment in **Figure 1a** to avoid confusion or incorrect statements and updated **Figure 2d** to focus on the increase in cell concentration over time as measured by cell counts and chlorophyll extraction. The corresponding sentence in the Abstract was also changed accordingly.

Original text in the Abstract:

“Additionally, the metabolic production of OH⁻ ions in the surrounding medium created an environment for the formation of insoluble carbonates via microbially-induced calcium carbonate precipitation (MICP). Digital design and fabrication of the living material ensured sufficient access to light and nutrient transport of the encapsulated cyanobacteria, which were essential for long-term viability (more than one year) as well as efficient photosynthesis and carbon sequestration. The photosynthetic living materials sequestered approximately 2.5 mg of CO₂ per gram of hydrogel material over 30 days via dual carbon sequestration, with 2.2 ± 0.9 mg stored as insoluble carbonates.”

Revised text in the Abstract:

“Additionally, the living materials created a favorable environment for the formation of insoluble carbonates via microbially-induced calcium carbonate precipitation (MICP). Digital design and fabrication of the living materials ensured sufficient access to light and nutrient transport of the encapsulated cyanobacteria, which were essential for long-term culture (more than one year), photosynthesis, and carbon sequestration. The photosynthetic living materials sequestered 2.2 ± 0.9 mg of CO_2 per gram of hydrogel material in the form of insoluble carbonates over 30 days, with continuous increase in cyanobacteria cell number, enabling further CO_2 sequestration in the form of biomass.”

Revised figures:

Figure 1. a) PCC 7002 are capable of CO_2 sequestration via biomass accumulation and microbially induced carbonate precipitation (MICP) during photosynthesis using dissolved CO_2 as the major carbon source.

Figure 2. d) An increase in cell concentration was observed throughout the 30-day incubation in the **encapsulated samples** ($n = 3$), with an initial rapid growth rate up to day 15 followed by a slower growth rate from day 15 to day 30 (green). **Liquid suspension with the same starting cell concentration was used as a control of cell growth** (black).

Finally, we agree that more sophisticated methods can be used to quantify photosynthesis in cyanobacteria, such as direct measurements of O_2 evolution using an oxygen electrode or using membrane inlet mass spectrometry. We have included a comment on this in the Discussion as an important focus for future work, from our group and others, that aim to quantitatively measure the photosynthetic rate of encapsulated cyanobacteria in photosynthetic living materials.

The updated text in the Discussion now reads (page 13):

*[...] We expect the system to reach its peak photosynthesis rate early as the cells go through an initial exponential growth phase and later operate at a pseudo-steady state while maintaining high viability (Figure S29). **Future work should include a more quantitative evaluation of CO_2 sequestration through biomass accumulation to better understand the photosynthetic rate of the cyanobacteria. Direct measurements of O_2 evolution using an oxygen electrode would provide a more accurate quantification of photosynthesis. Additionally, established computational models could be applied to further explore the relationship between cytosolic pH of the cyanobacteria and CO_2 assimilation into biomass⁶⁵.***

Reviewer #3 (Remarks to the Author):

General Comments:

The manuscript describes 3D-printed hydrogel-based materials as novel tools for dual carbon sequestration. Photosynthetic living materials were engineered by embedding cyanobacteria within a bioinert hydrogel matrix, to achieve effective dual carbon sequestration through biomass accumulation and irreversible microbially-induced calcium carbonate precipitation. The authors investigated the platform's performance not only in the short term (1 month) but also conducted long-term studies to demonstrate the platform's ability in forming biomass and producing mineral precipitation for over a year.

While the authors have made commendable efforts to address the reviewer's questions, some questions were not adequately addressed. The reviewer acknowledges these efforts but believes that several questions need further clarification. Therefore, the manuscript is not recommended for publication in its current state.

Technical Comments:

1. Hydrogel formulation

a. The authors have adequately addressed the printability of Pluronic F127 hydrogels of different polymer contents. However, the images in Figure S5 are not consistent in lighting, resulting in difficulty to access the resolution of prints. Higher-quality images with proper and consistent lighting are recommended.

To improve the quality of the transparent hydrogel images, we took additional pictures of the printed samples using a camera (alpha 7R, Sony) equipped with a macro lens (FE 2.8/90 MACRO G OSS, Sony) with consistent lighting.

We have updated the former **Figure S5**, current **Figure S6** with these new images.

Figure S6. Printability of Pluronic F127 hydrogels of different polymer contents (22G nozzle, printing speed $v = 5\text{--}15\text{ mm s}^{-1}$, printing pressure $P = 16\text{--}45\text{ kPa}$). **Top and planar view.** Scale bar, 10 mm.

b. How was the stability in Figure S6 evaluated? What causes the instability?

As seen in **Figure S6**, low (<3 wt%) F127-BUM concentrations exhibited instability in the rheology during photo-cross-linking, as observed in the decrease in the storage modulus upon irradiation ($\lambda = 405\text{nm}$; $I = 8 \text{ mW cm}^{-2}$). We suppose that the low concentration led to phase separation and micelle rearrangement rather than network formation, rendering these formulations unsuitable for use in printed structures.

c. The authors mentioned that “The choice of initial OD does not impact the cell growth (Figure S33)”. This statement should be moved to Results and Discussion to enhance clarity.

We thank the Reviewer for the suggestion and we have moved the statement to Results section accordingly. Previous **Figure S33** is named **Figure 13a** in the revised manuscript.

Revised text in Results (page 7):

“The number of clusters and individual cells increased over time with cell concentrations reaching $5 \times 10^9 \text{ cells mL}_{\text{gel}}^{-1}$ after 30 days (Figure 2d, S12). Further, the choice of initial $OD_{730\text{nm}}$, within the range of 0.3–1.6, did not impact cell growth (Figure S13b).”

2. The authors have adequately addressed the comment.

3. Cyanobacteria encapsulation

a. The authors are encouraged to expand the time points for biomass quantification and cell viability measurements to include day 0 and day 5, in addition to the current intervals starting from day 10, due to swelling and dissolution of the un-crosslinked F127 during this period. The newly added Figure S21 suggests that most changes predominantly occur within the day 1-10 timeframe.

We thank the Reviewer for the suggestion and we added viability measurements on day 0 and day 5 to **Figure 2d**.

The updated **Figure 2d** is appended below:

Figure 2. d) An increase in cell concentration was observed throughout the 30-day incubation in the **encapsulated** samples ($n = 3$), with an initial rapid growth rate up to day 15 followed by a slower growth rate from day 15 to day 30 (green). **Liquid suspension with the same starting cell concentration was used as a control of cell growth (black).**

Updated text in the Methods section:

“Cell viability and growth within the hydrogels were evaluated using confocal microscopy for **day 1 to 5** and chlorophyll extraction **for day 10 to 30 (Figure S33).**”

b. The same comment applies to the evaluation of cyanobacteria leakage.

We thank the Reviewer for the suggestion and we have also included data for day 1 and day 5 to the leakage data (**Figure S13b**).

Figure S13. b) Cell count in the printed disc sample quantified with chlorophyll extraction (green) and cell count in the surrounding culture medium (pink) measured via Coulter Counter ($n = 3$). The right axis indicates the percentage of cells that leaked into the surrounding culture medium normalized to the total number of cells in the printed disc and the culture medium combined (gray), as well as the encapsulation efficiency calculated as 100% – leakage % (brown).

4. The authors have adequately addressed the comment.

5. Dual carbon sequestration

a. In Figure 2E, the variability in the abiotic samples on day 1, day 5, and day 30 is very high. Please provide an explanation. In addition, why is there a drop from day 0 to day 1? Discussion of the results should be added to the main text.

We would like to clarify the data presented in **Figure 2e**. The left axis of **Figure 2e** (black) represents the difference in dry mass between 5 randomly selected biotic samples and 5 randomly selected abiotic samples. The variability on days 1 and 5 was high as the error bars reflected the cumulative error of these 5 randomly selected biotic samples and the 5 randomly

selected abiotic samples (**Figure S-I**, see **Equivalent CO₂ sequestration calculation in living materials** in the Supplementary Information, page 11). Due to the differences in the diffusion rate of unfunctionalized F127 polymer from the printed discs, there was sample to sample variation in dry mass on days 1 and 5 (**Figure S-I**). This resulted in a relatively high error when calculating the mass difference between the 5 biotic and 5 abiotic samples. We also attribute the drop in mass from day 0 to day 1 to the diffusion of unfunctionalized F127 out of the samples. We have updated the caption of **Figure SX** to explain these points.

The right axis of **Figure 2e** (right axis, red) represents the CaCO₃ concentration quantified by Alizarin Red S staining and the absorbance measurement at 405 nm. For each time point, we measured the absorbance of 3 randomly selected biotic samples. The average absorbance of 3 randomly selected abiotic samples (measured at each time point) was subtracted as background to obtain the final values. The error bars, therefore, reflected the cumulative variability between the biotic and abiotic samples. We then calculated the corresponding CaCO₃ concentration based on these absorbance values using the calibration curve (**Figure S15 a**).

Text in the original Results section (page 8):

*“To assess the amount of biomass and precipitate formation, the mass of abiotic and biotic samples was measured and compared. Over the incubation period of 30 days, the biotic samples had approximately 36% more dry mass than control abiotic samples (**Figure 2e**) with significant increase ($p = 0.001$) in biomass observed from day 10 of incubation.”*

Revised text in the Results section (page 7):

*“To assess the amount of biomass and precipitate formation, the mass of abiotic and biotic samples was measured and compared (**Figure 2e**). The initial decrease in sample mass was attributed to the diffusion of unfunctionalized F127 polymer out of the printed discs, which also caused high variability in the data of day 1 and day 5. Over the incubation period of 30 days, the biotic samples had approximately 36% more dry mass than control abiotic samples with significant increase ($p = 0.001$) in **dry mass** observed from day 10 of incubation.”*

Figure 2e. Both biomass growth and CaCO_3 precipitation contributed to the increase in dry mass of the biotic samples. The cumulative CO_2 sequestration by biotic samples was reflected by the difference in normalized dry mass between biotic and abiotic samples (black). Over the incubation period of 30 days, biotic samples accumulated 36% more dry mass than the abiotic samples. Alizarin Red S staining on the biotic samples indicated a total of 4.8 mg CaCO_3 precipitation per gram of gel (red).

Figure S-I: Normalized dry mass of abiotic (black) and biotic (green) samples over the incubation period of 30 days. The dry mass of each sample was normalized to the average initial mass of day 0. The non-cross-linked F127 diffused out over time and caused higher sample to sample variation in dry mass on day 0 to 5.

b. The authors stated that “It is important to note that the available medium volume was different for our encapsulated samples and suspension. Since each disc sample was 40 μL with concentrated cell clusters grown in 2 mL of medium, we observed the cell number within an encapsulated state is higher than that in the liquid positive control.” In order to compare the two experimental groups, the experiment should be carried out at same concentration or volume, otherwise there is an overestimation of the performance of the encapsulated samples.

We apologize for any confusion regarding the control experiment setup in this case. We indeed started the encapsulated samples and the liquid suspension control at the same cell concentration per volume. Both the liquid suspension control and the encapsulated samples were prepared at the same initial concentration (equivalent optical density of $\text{OD}_{730\text{nm}} = 0.8$) to ensure that the cells are in the same growth phase in both cases. For the encapsulated sample, an equivalent starting $\text{OD}_{730\text{nm}} = 0.8$ was reached within a gel volume of 40 μL , which we considered as the primary cell growth volume. The encapsulated samples were immersed in 2 mL of culture medium, but cell growth occurred in the gel volume of 40 μL and very limited cell leakage was observed (**Figure S13b**). The cells grew in immobilized clusters causing their high final concentration (**Figure S12**). For the suspension sample, the starting cell concentration was also $\text{OD}_{730\text{nm}} = 0.8$ with the individual cyanobacteria cells dispersed in the culture medium without forming clusters. This led to a lower final concentration compared with that observed in the encapsulated state after the 30-day culture period. We would like to note that the

comparative value between encapsulated and liquid suspension culture is not a conclusion of the manuscript. We include these data to show that the cyanobacteria grow efficiently in both the liquid suspension control and within our photosynthetic living materials, indicated that encapsulation does not inhibit their growth.

Figure S12. Representative sample maximum intensity Z-projection images at different time points (cell autofluorescence, green). Scale bar, 100 μm .

Figure S13. b) Cell count in the printed disc sample quantified with chlorophyll extraction (green) and cell count in the surrounding culture medium (pink) measured via Coulter Counter ($n = 3$). The right axis indicates the percentage of cells that leaked into the surrounding culture medium normalized to the total number of cells in the printed disc and the culture medium combined (gray), as well as the encapsulation efficiency calculated as $100\% - \text{leakage } \%$ (brown).

c. Where are the standard deviations in Figure 2D?

We have added the standard deviation accordingly.

Figure 2. d) An increase in cell concentration was observed throughout the 30-day incubation in the **encapsulated samples (n = 3)**, with an initial rapid growth rate up to day 15 followed by a slower growth rate from day 15 to day 30 (green). **Liquid suspension with the same starting cell concentration was used as a control of cell growth (black).**

6. The authors have adequately addressed the comment.

7. The authors have adequately addressed the comment.

8. Volumetric lattice structure

a. Please provide the information on time points for the SEM cross-section. Which cross-section of the sample was imaged?

The samples were printed and incubated for 5 days before lyophilization and mounting for SEM imaging. To showcase the porosity on the sample surface, a sample (**Figure S25a, b**) was dissected along the vertical axis and flattened on the SEM holder. Another sample (**Figure S25c**) was dissected horizontally (transverse cross-section) to showcase the internal structure radially.

Figure S25. a) Optical image of a lyophilized sample prepared by volumetric 3D printing and incubated for 5 days. Scale bar, 1 cm. b) SEM image of the sample surface with lattice structure false colored in blue. Scale bar, 1 mm. c) SEM image of the transverse cross-section of the sample prepared by volumetric 3D printing. Scale bar, 1 mm.

b. The authors stated that the material “exhibited robust viability up to 365 days” in Figure 4. However, cell viability was evaluated for up to 30 days only. Further evaluation at longterm time points (days 60, 120, and 365) should be carried out to corroborate the authors’ statement.

We thank the Reviewer for this comment and as we did not quantify cell viability using chlorophyll extraction and confocal microscopy beyond day 30, we have softened these statements in the manuscript. Fortunately, cyanobacteria, owing to their continual production of chlorophyll when metabolic active, provide a visual representation of their active state. We observed a vibrant green color in the samples, indicative of continuous chlorophyll production throughout the lifecycle reported in this manuscript. When samples die, they yellow, in our hands, within 48 h. We elaborate on this point below and have included additional images of our samples that were cultured long-term as well as representative images of samples that died, showing yellowing.

The updated text in the revised manuscript now reads (Figure 4 caption, page 12):

[...] maintained a vibrant green color up to 365 days, indicative of continuous chlorophyll production.”

We collected images throughout the life cycle of the photosynthetic living materials cultured over 1 year that were presented in Figure 4b. The green shade in the optical images comes from the *chlorophyll a* content of the encapsulated cyanobacteria¹. When *chlorophyll a* in cyanobacteria degrades upon or after cell death, it will be converted to phaeopigments, which gives a brownish-yellow color. Throughout the sample incubation time of one year, there was no yellowing observed, which would be indicative of cyanobacteria death and associated chlorophyll oxidation. We included additional images in the series in the Supporting Information (Figure S26a), appended below. In addition, we included an image of a typical unhealthy sample where the embedded cyanobacteria show the phaeopigment-like brownish-yellow color² within 48 h of cell death (Figure S26b).

The updated text in the revised manuscript now reads (page 11):

“The 3D-printed photosynthetic living structure was cultured for over a year, during which it continuously produced chlorophyll and performed dual carbon sequestration (Figure S26).”

In addition, we updated the text in the original manuscript (page 11, page 13):

“This prolonged viability of the photosynthetic living materials showed that dual carbon sequestration can take place beyond 30 days, especially within rationally designed 3D structures.”

And

“Tailoring construct geometry via additive manufacturing enabled our photosynthetic living structures to survive beyond one year while continuously undergoing dual carbon sinking.”

Which now reads (page 11, page 13):

“This prolonged culture of the photosynthetic living materials showed that dual carbon sequestration can take place beyond 30 days, especially within rationally designed 3D structures.”

And

“Tailoring construct geometry via additive manufacturing enabled our photosynthetic living structures to continuously perform dual carbon sequestration beyond one year.”

Figure S26. a) Optical images of 3D printed photosynthetic living material at different time points up to 365 days. Scale bar, 1 cm. **b)** 3D-printed photosynthetic living material yellowed within 48 h upon nutrient depletion, indicative of cyanobacteria death and chlorophyll a oxidation. Scale bar, 1 cm.

As stated above, the samples reached a pseudo-steady state of cell number prior to day 30 (**Figure 2d**). Thus, given the visual indication of continuous chlorophyll production throughout the life cycle of the photosynthetic living materials, we do not see added value in further measures of individual cell viability. In order to be precise, we have therefore softened all language related to ‘robust viability’ or ‘viability’ in the long-term samples and rephrase this as continuous chlorophyll production.

References:

1. Alvey, R. M., Biswas, A., Schluchter, W. M. & Bryant, D. A. Effects of modified Phycobilin biosynthesis in the Cyanobacterium *Synechococcus* sp. Strain PCC 7002. *J. Bacteriol.* 193, 1663–1671 (2011).

2. Sanmartín, P., Villa, F., Silva, B., Cappitelli, F. & Prieto, B. Color measurements as a reliable method for estimating chlorophyll degradation to phaeopigments. *Biodegradation* 22, 763–771 (2011).

Figure 2. d) An increase in cell concentration was observed throughout the 30-day incubation in the **encapsulated** samples ($n = 3$), with an initial rapid growth rate up to day 15 followed by a slower growth rate from day 15 to day 30 (green). **Liquid suspension with the same starting cell concentration was used as a control of cell growth (black).**

9. The authors reported cell viability up to day 30. The viability test should be evaluated at the latest time points (days 60, 120, and 365).

We refer to the response to comment 8b from the same Reviewer directly above.

10. The authors claimed that “Beyond 30 days, we expect the biomass accumulation to reach a pseudo steady state (with similar cell death and growth rates) and the component of carbon sequestration related to biomass accumulation to equilibrate within the hydrogel” However, further explanation is needed to clarify why this outcome is expected. Could the authors provide the rationale or reference supporting this expected outcome?

We drew this conclusion as we observed that cell number reached a plateau beyond 25 days within the encapsulated samples, as shown in **Figure 2d**. This suggests that the cell growth and cell death rates likely balanced out, leading to a plateau in biomass accumulation. Even though the cell concentration reached a plateau, we still observed increased pH (**Figure S11b**), indicating that the cyanobacteria within the encapsulated samples remain metabolically active.

Figure 2. d) An increase in cell concentration was observed throughout the 30-day incubation in the **encapsulated samples** ($n = 3$), with an initial rapid growth rate up to day 15 followed by a slower growth rate from day 15 to day 30 (green). **Liquid suspension with the same starting cell concentration was used as a control of cell growth** (black).

Figure S1. b) pH changes of pooled biotic (green) and abiotic (black) medium every 5 days ($n = 3$).

11. The authors mentioned that “The collected medium pooled from individual samples ($n = 20$) was used for pH measurements to evaluate total sequestered CO₂ as fully described in the Supplementary Information”. However, it is unclear why the measurements were pooled rather than taken from individual samples, which would allow reporting of the averages and standard deviations. Additionally, while the authors stated that the experiment was performed twice, no information was provided on the averages or standard deviations for these repeated trials.

We pooled media from 20 individual samples as each sample was incubated with 2 mL of culture medium. Titration on 2 mL of culture medium is challenging and, in our hands, resulted in larger error. Therefore, we pooled the medium from 20 samples together and titrated on the pooled medium for more accurate and reliable results. The error bars have now been added to **Figure S11**. The DIC calculation also consisted of two sets of independent experiments, and each data point was obtained by sampling the pooled medium from 20 individual samples.

Figure S1. a) Total amount of dissolved inorganic carbon (DIC) under equilibrium in the re-collected abiotic (black) and biotic (green) medium after every 5 day culture period ($n = 2$, with pooling of 20 individual samples in each case). The gray dashed line indicates the amount of DIC in the form of Na₂CO₃ that was used in simulated seawater medium (BG11-ASNIII). b) pH changes of pooled biotic (green) and abiotic (black) medium every 5 days ($n = 3$).

Minor Comments:

12. Page 8, Line 9 – There is a spelling error in “asses”. This should be corrected.

We thank the reviewer for pointing out the typo and it has been corrected accordingly.

13. The reference to “Biomass quantification and cell viability characterization” (currently referred to Figure S25) appears to be incorrect. Please verify and correct this figure reference.

We thank the reviewer for pointing this out, the reference has been corrected.

Response to Reviewers

Reviewer #1 (Remarks to the Author):

The authors have made improvements in the manuscript. However, the authors have still not addressed two critical comments.

Major comments:

1. It's still questionable whether atmospheric CO₂ is truly the primary carbon source for sequestration (lines 183-184). It's not sure whether we can regard these materials as meaningful sequestration system if the authors take initial input [CO₃²⁻] into consideration as "cost" carbon sources.

Maybe I miss something, but when I look at Figure S1a, I find that the amount of carbonate refreshed every 5 days is not an insignificant amount. For example, let's consider the decrease in DIC from day 5 to day 10: Right after the medium refreshment at day 5, [DIC] should be similar to the one at day 0 (i.e. ~900umol/L with ~380umol/L coming from Na₂CO₃ in the medium). At day 10, [DIC] have reduced to ~600umol/L, corresponding to a reduction of 300umol/L used for biomass and carbonate precipitates. Why could these 300umol/L not come (in significant part) from Na₂CO₃ in the medium? And then from day 15 to 20, it goes from ~900umol/L to ~400umol/L, and why could these 500umol/L "used" not come in large part from Na₂CO₃ in the medium?

The authors should make two things clear in the manuscript or supplementary information: 1] How much CO₃ of carbonate precipitates or biomass originally come from the atmospheric CO₂, at least with clear simulation or calculations. The [CO₃²⁻] supplied every 5 days (refreshed) can be used for both clearly. 2] The contents of Figure S1a should include more details to make it clearer. It would help a lot if this graph also included the datapoints for [DIC] right after the refreshing of the medium.

These are critical points, which are essential to convincingly demonstrate the claims of "dual sequestration". So, once this has been demonstrated, I recommend the data to be included as a main Figure or Table rather than in the S.I.

To ensure there is no confusion in the manuscript about the initial dissolved inorganic carbon (DIC) present in the medium, we have modified the statement of the primary source of carbon in the manuscript accordingly.

Updated text in the revised version of the Results (page 7):

"This increase in the DIC confirmed the dissolution of atmospheric CO₂, which together with dissolved Na₂CO₃ in the simulated seawater medium, served as the carbon sources for CO₂ sequestration."

Updated text in the revised version of the Discussion (page 14):

"Our photosynthetic living materials function at ambient conditions with atmospheric CO₂ and simulated seawater as the carbon sources, [...]"

Updated text in the revised version of the Conclusion (page 14):

“The photosynthetic living materials performed dual carbon sequestration over an extended life of up to one year with light as its energy source; atmospheric carbon and simulated seawater as its carbon sources.”

However, we would like to note that the precise source of inorganic carbon—whether originating from the initial sodium carbonate or dissolved from the atmosphere to maintain the carbon equilibrium in the medium—has no bearing on the claim of “dual carbon sequestration” in the manuscript.

In the prior revision, we included the DIC calculations for the biotic and abiotic samples taking measurements at the end of each 5-day cycle. These data demonstrated that the DIC remained above the level given by the addition of Na₂CO₃ at all times, independent of medium pH, temperature, and bacterial consumption of CO₃²⁻ ions, all of which affect the DIC content. As suggested by the reviewer, we have now updated **Figure S1** to include the amount of DIC in the fresh medium at the start of each 5 day cycle (**Figure S1a**). As the living materials undergo carbon sequestration, they decrease the amount of DIC in the culture medium, yet atmospheric carbon dissolves in the culture medium to restore the carbon equilibrium (which depends on temperature and pH). Notably, **Figure S1c** shows that even without Na₂CO₃, the amount of DIC exceeded 381 μmol L⁻¹, confirming that atmospheric carbon dissolved in the culture medium in all conditions.

In this context, even if the initial Na₂CO₃ is considered a distinct source of inorganic carbon, the maximum percentage of inorganic carbon from Na₂CO₃ that is used by the living materials is $381\mu\text{mol L}^{-1}/892\mu\text{mol L}^{-1} \times 100\% = 42.7\%$. Therefore, atmospheric carbon remains the primary source of carbon for the living materials (> 50%). Nonetheless, we have now explicitly acknowledged in the manuscript that the simulated seawater contributed to the overall carbon pool, as stated above.

We would like to emphasize that the use of sodium carbonate (Na₂CO₃) in the medium was to simulate natural seawater conditions in a controlled lab environment. In natural seawater, carbonate ions result from the dissolution and equilibration of atmospheric CO₂, where the equilibrium is dictated by pH, temperature, and biological activity. We used Na₂CO₃ to replicate this carbonate buffering system, which is a standard approach in the field. The photosynthetic living materials then performed dual carbon sequestration in this simulated seawater environment by consuming dissolved inorganic carbon (DIC), which was in dynamic equilibrium with atmospheric CO₂.

As demonstrated in the previous revision, the equilibrium DIC was always at or above the amount introduced in the medium by Na₂CO₃, independent of pH and temperature. In an open system, if a CO₃²⁻ ion is consumed it will be replaced with atmospheric carbon at equilibrium, regardless of whether the CO₃²⁻ ion came from Na₂CO₃ or the atmosphere. This was also observed experimentally in closed-container experiments, which indicated a decrease in the concentration of atmospheric CO₂ during the day cycle (**Figure S30 b,c,d**) as elaborated in response to point 2.

Figure S1. a) Total amount of dissolved inorganic carbon (DIC) under equilibrium in the re-collected abiotic (black) and biotic (green) medium after every 5-day culture period ($n = 2$, with pooling of 20 individual samples in each case). The gray dashed line indicates the amount of DIC in the form of Na_2CO_3 that was used in simulated seawater medium (BG11–ASNIII). **b)** pH changes of pooled biotic (green) and abiotic (black) medium every 5 days ($n = 3$). **c)** DIC analysis of BG11–ASNIII medium without the addition of inorganic carbon through sodium carbonate (Na_2CO_3) at pH = 6.7 and pH = 8.9 ($n = 2$, black). The gray dashed line indicates the amount of DIC in the form of Na_2CO_3 that was used in simulated seawater medium (BG11–ASNIII).

2. Figure S30: The rate of CO_2 change in a closed chamber is indeed a way to directly demonstrate CO_2 sequestration. However, the uncertainty on this measurement is very high, with quite low R2. The addition of 2 controls is needed to validate this result: A control in dark conditions where CO_2 should increase because of respiration of the cells (this data should already be available to the authors since the materials were monitored for 10 days, so I guess also while in dark condition). A control without cells (but in the same chamber and with an initial CO_2 concentration higher than atmospheric) to check whether the sensor reports a constant amount of CO_2 in such conditions, without any leakage, etc. It would be nice to show these controls and calibration data together. The authors can refer to the following paper for further information:t

Banerjee, S., Siemianowski, O., Liu, M., Lind, K. R., Tian, X., Nettleton, D., & Cademartiri, L. (2019). Stress response to CO_2 deprivation by *Arabidopsis thaliana* in plant cultures. *PLoS One*, 14(3), e0212462.

Also, the authors should display the full data from their CO₂ measurements including the 30min equilibration time and also the data after 1.4h, and explain clearly why the period between 0.5h-1.4h was specifically used to calculate the rate in change of [CO₂]. If these are not possible, I would recommend to just remove figure S30 from the Supplementary data.

We thank the Reviewer for the careful look at these additional data. In order to build confidence in our setup to monitor CO₂ concentration within a closed chamber containing our photosynthetic living materials, we had carried out the 2 controls requested and we have now included these data in the Supporting Information.

First, we loaded the chamber with an excess concentration of CO₂ (~40'000 ppm) and monitored the concentration over time. The CO₂ concentration remained constant throughout the measurement period, after an initial stabilization duration of ~0.5 h in two independent experiments, suggesting no leakage from our system (**Figure S30 e,f**). Based on these leakage tests, we also identified 0.5 h (30 min) as the initial equilibration time for all subsequent measurements.

Figure S30. e, f CO₂ leakage tests in 250 mL closed containers without cells loaded with an excess concentration of CO₂.

The change in CO₂ concentrations during light(day)/dark(night) cycles was observed during the experiment period and **Figure S30 b,c,d** show representative snapshots of the CO₂ evolution within the closed chambers. In dark conditions (night; areas shaded in grey), an increase (or plateau) in CO₂ concentration was observed and this can be attributed mainly to cell respiration, as indicated by the Reviewer. During the light cycles (day; areas shaded in green), CO₂ concentration decreased within the closed chamber as expected for active photosynthesis.

Figure S30. **a** Rate of change of CO₂ concentration in a 250 mL closed container over a measurement period of 10 days with a 0.25 mL sample and 10 mL culture medium ($n = 2$). CO₂ sequestration was observed from day 3 and plateaued beyond day 5. **b,c,d** Representative snapshots of CO₂ evolution in a closed container over 1 day-night cycle and 3 day-night cycles. Black scatters: CO₂ concentration; green shading: light (day) conditions; grey shading: dark (night) conditions.

We would like to clarify that the quantity of dual CO₂ sequestration and conclusions reported in the main text of the manuscript were not based on any closed-container measurements. These experiments were included to corroborate that the systems consume atmospheric carbon via photosynthesis during their life cycle and are therefore included in the Supporting Information. We extracted rate information by interpolating stable sections of our data from the CO₂ sensor, as previously shown in **Figure S30b**. Such interpolations were taken over many sections of the data and we agree that it is better to show whole sections of the sensor data, which are now present in the light(day)/dark(night) cycles presented in **Figure S30 b,c,d**. We have kept the extracted rate information as it nicely shows that the system takes some days to reach a steady state of CO₂ consumption during the day cycles. Again, we thank the Reviewer for this comment and agree that with the additional controls and explanation, the data more clearly demonstrate that our systems convincingly consume atmospheric CO₂ during their life cycle.

Minor comment:

1. In the revised manuscript (Line 173), the authors mentioned “uniform circular samples ($V_{\text{sample}} = 40 \mu\text{L}$, $d = 10 \text{ mm}$)”. Is the volume (40uL) correct here?

Yes, the volume of the printed samples is correct here, $V_{\text{sample}} = 40 \mu\text{L} = 40 \text{ mm}^3$, $d = 10 \text{ mm}$, thickness $l = 0.5 \text{ mm}$. The samples were printed using 22G conical nozzles ($\varnothing = 0.41 \text{ mm}$). We clarified the information of the final printed sample height in the revised manuscript.

Title: Dual carbon sequestration with photosynthetic living materials

General Comments:

The manuscript describes 3D-printed hydrogel-based materials as novel tools for dual carbon sequestration. Photosynthetic living materials were engineered by embedding cyanobacteria within a bioinert hydrogel matrix, to achieve effective dual carbon sequestration through biomass accumulation and irreversible microbially-induced calcium carbonate precipitation. The authors investigated the platform's performance not only in the short term (1 month) but also conducted long-term studies to demonstrate the platform's ability in forming biomass and producing mineral precipitation for over a year.

While the authors have made commendable efforts to address the reviewer's questions, some questions were not adequately addressed. The reviewer acknowledges these efforts but believes that several questions need further clarification. Therefore, the manuscript is not recommended for publication in its current state.

Technical Comments:

1. Hydrogel formulation
 - a. The authors have adequately addressed the printability of Pluronic F127 hydrogels of different polymer contents. However, the images in Figure S5 are not consistent in lighting, resulting in difficulty to access the resolution of prints. Higher-quality images with proper and consistent lighting are recommended.
 - b. How was the stability in Figure S6 evaluated? What causes the instability?
 - c. The authors mentioned that "The choice of initial OD does not impact the cell growth (Figure S33)". This statement should be moved to Results and Discussion to enhance clarity.
2. The authors have adequately addressed the comment.
3. Cyanobacteria encapsulation
 - a. The authors are encouraged to expand the time points for biomass quantification and cell viability measurements to include day 0 and day 5, in addition to the current intervals starting from day 10, due to swelling and dissolution of the un-crosslinked F127 during this period. The newly added Figure S21 suggests that most changes predominantly occur within the day 1-10 timeframe.
 - b. The same comment applies to the evaluation of cyanobacteria leakage.
4. The authors have adequately addressed the comment.
5. Dual carbon sequestration
 - a. In Figure 2E, the variability in the abiotic samples on day 1, day 5, and day 30 is very high. Please provide an explanation. In addition, why is there a drop from day 0 to day 1? Discussion of the results should be added to the main text.
 - b. The authors stated that "It is important to note that the available medium volume was different for our encapsulated samples and suspension. Since each disc sample was 40 μ L with concentrated cell clusters grown in 2 mL of medium, we observed the cell number within an encapsulated state is higher than that in the liquid positive control." In order to

compare the two experimental groups, the experiment should be carried out at same concentration or volume, otherwise there is an overestimation of the performance of the encapsulated samples.

- c. Where are the standard deviations in Figure 2D?
6. The authors have adequately addressed the comment.
7. The authors have adequately addressed the comment.
8. Volumetric lattice structure
 - a. Please provide the information on time points for the SEM cross-section. Which cross-section of the sample was imaged?
 - b. The authors stated that the material “exhibited robust viability up to 365 days” in Figure 4. However, cell viability was evaluated for up to 30 days only. Further evaluation at long-term time points (days 60, 120, and 365) should be carried out to corroborate the authors’ statement.
9. The authors reported cell viability up to day 30. The viability test should be evaluated at the latest time points (days 60, 120, and 365).
10. The authors claimed that “Beyond 30 days, we expect the biomass accumulation to reach a pseudo-steady state (with similar cell death and growth rates) and the component of carbon sequestration related to biomass accumulation to equilibrate within the hydrogel” However, further explanation is needed to clarify why this outcome is expected. Could the authors provide the rationale or reference supporting this expected outcome?
11. The authors mentioned that “The collected medium pooled from individual samples (n = 20) was used for pH measurements to evaluate total sequestered CO₂ as fully described in the Supplementary Information”. However, it is unclear why the measurements were pooled rather than taken from individual samples, which would allow reporting of the averages and standard deviations. Additionally, while the authors stated that the experiment was performed twice, no information was provided on the averages or standard deviations for these repeated trials.

Minor Comments:

12. Page 8, Line 9 – There is a spelling error in “asses”. This should be corrected.
13. The reference to “Biomass quantification and cell viability characterization” (currently referred to Figure S25) appears to be incorrect. Please verify and correct this figure reference.